# Latent Laplace Diffusion for Irregular Multivariate Time Series

**Zinuo You** [1]  **Jin Zheng** [2]  **John Cartlidge** [2]

## Abstract

Irregular multivariate time series impose a trade-off for long-horizon forecasting: discrete methods can distort temporal structure via re-gridding, while continuous-time models often require sequential solvers prone to drift. To bridge this gap, we present Latent Laplace Diffusion (LLapDiff), a generative framework that models the target as a low-dimensional latent trajectory, enabling horizon-wide generation without step-by-step integration over physical time. We guide the reverse process utilizing a stable modal parameterization motivated by stochastic port-Hamiltonian dynamics, and parameterize its mean evolution in the Laplace domain via learnable complex-conjugate poles, enabling direct evaluation over irregular timestamps. We also link continuous dynamics to irregular observations through renewal-averaging analysis, which maps sampling gaps to effective event-domain poles and motivates a gap-aware history summarizer. Extensive experiments show that LLapDiff improves over baselines in long-horizon forecasting, and its continuous-time generative nature supports missing-value imputation by querying the same model at historical timestamps. Code is available at https://github.com/pixelhero98/LLapDiffusion.

## 1. Introduction

Real-world multivariate time series spanning healthcare, climate science, and finance are rarely observed on a clean and synchronous grid. Instead, measurements arrive at nonuniform timestamps and exhibit variable-wise missingness that is often informative (Weerakody et al., 2021; Schirmer et al., 2022). Modeling in this regime requires respecting physical time and sampling gaps without reducing irregularity to a mere preprocessing artifact, while producing stable

---

[1]School of Computer Science, University of Bristol. [2]School of Engineering Mathematics and Technology, University of Bristol. Correspondence to: Zinuo You <zinuo.you@bristol.ac.uk>.

*Proceedings of the 43$^{rd}$ International Conference on Machine Learning*, Seoul, South Korea. PMLR 306, 2026. Copyright 2026 by the author(s).

long-horizon trajectories. Conversely, unconstrained rollouts easily drift or diverge, particularly as gaps become larger and available observations become sparse.

Most existing methods for irregular multivariate time series (IMTS) commonly fall into three families. First, discrete-time pipelines interpolate or impute onto a regular grid (Cao et al., 2018; Shukla & Marlin, 2019), or discretize events into tokens/patches to apply strong sequence models (Zhang et al., 2023; 2024). While efficient, aggressive re-gridding beyond native-bin indexing can distort timing information under severe irregularity and blur the semantics of missingness (Peng et al., 2025). Second, continuous-time models, such as Neural ODE/CDE families (Chen et al., 2018; Rubanova et al., 2019; Kidger et al., 2020), continuous-time RNNs (Schirmer et al., 2022), and continuous-time Transformers (Chen et al., 2023), incorporate timestamps naturally. But, these methods typically require sequential numerical integration, increasing cost and accumulating error over long horizons (Biloš et al., 2021; Westny et al., 2024). Third, diffusion/score-based generative models (Ho et al., 2020; Song et al., 2021b) provide strong uncertainty quantification and sample quality, and have been adapted to forecasting and imputation by conditioning denoising on historical context (Tashiro et al., 2021; Rasul et al., 2021). However, most time-series diffusion methods denoise directly in the observation space and handle irregularity mainly through masks and time embeddings. Consequently, the denoisers lack explicit dynamical structure or stability control, making long-horizon generation fragile under irregular sampling (Alcaraz & Strodthoff, 2023; Kollovieh et al., 2023).

In the face of these challenges, we present Latent Laplace Diffusion (LLapDiff), a conditional generative model that combines continuous-time inductive bias without ODE/SDE solver calls. Here, solver-free refers to avoiding numerical integration over the physical time; the reverse process is iterative only over diffusion noise levels. First, LLapDiff represents irregular target horizons as low-dimensional latent trajectories and performs diffusion in this latent space, avoiding direct denoising over sparse masked observations. Then, to guide the reverse process toward stable dynamics without numerical solvers, we draw inspiration from stochastic port-Hamiltonian systems (Satoh & Fujimoto, 2008; van der Schaft & Maschke, 2002) and parameterize the latent mean evolution in the Laplace domain (Antoulas, 2005) via sta-

ble complex-conjugate poles (Gustavsen & Semlyen, 2002), enabling horizon-wide generation (sampling still relies on reverse steps) over irregular grids. Finally, we analyze irregular sampling via a renewal-averaging perspective (Cox, 1962; Antunes et al., 2009), showing how random gaps map continuous-time poles to effective event-domain poles. This further motivates a gap-aware history summarizer that aggregates observations and time gaps to adapt learned dynamics to the sampling pattern. More broadly, LLapDiff formulates forecasting as continuous latent trajectory generation, which offers missing-value imputation as a byproduct by simply generating the trajectory at historical query timestamps.

Our contributions are threefold. First, we present LLapDiff, a latent diffusion framework for irregular multivariate time series that models its evolution as low-dimensional latent trajectories while preserving timestamp fidelity. Second, we derive a Laplace-domain modal parameterization grounded on stochastic port-Hamiltonian dynamics, allowing horizon-wide generation with explicit stability control through constrained pole damping. Third, we establish a theoretical link between continuous-time poles and random sampling gaps, which motivates a gap-aware history conditioning that improves long-horizon modeling under varying irregularities.

## 2. Related Work

**Irregular Time Series Modeling**. Discrete-time pipelines often align observations to a regular grid via interpolation or recurrence (Cao et al., 2018; Che et al., 2018; Das et al., 2024) and apply strong sequence backbones on tokenized representations (Zeng et al., 2023; Challu et al., 2023). Although efficient, aggressive re-gridding can distort temporal structure under severe irregularity and entangle missingness with preprocessing artifacts. Meanwhile, other discrete approaches avoid gridding by incorporating time gaps via attention (Shukla & Marlin, 2021; Li et al., 2023) or irregularity-aware encoding (Tipirneni & Reddy, 2022; Chowdhury et al., 2023). In parallel, graph-based forecasters (Zhang et al., 2024; Yalavarthi et al., 2024) construct temporal graphs but become fragile under severe sparsity due to weak connectivity. While continuous-time approaches, including Neural ODE/CDE (Rubanova et al., 2019; Kidger et al., 2020; Oh et al., 2024) and continuous Transformers (Chen et al., 2023), handle timestamps naturally but rely on costly sequential integration. Structured dynamical parameterizations, like SSMs (Gu et al., 2022; Gu & Dao, 2024), offer efficient long-context inference and have been adapted in time-series (Alcaraz & Strodthoff, 2023).

**Diffusion for Time Series**. Diffusion-based methods learn conditional generation by denoising with historical context. Forecasting models such as TimeGrad (Rasul et al., 2021) and TSDiff (Kollovieh et al., 2023) have been improved via non-autoregressive (Shen & Kwok, 2023) and multi-

resolution architectures (Shen et al., 2024), while imputation methods such as CSDI (Tashiro et al., 2021) perform well under structured missingness. However, many existing approaches denoise directly in observation space, treating irregularity primarily as a masking problem. This leaves dynamical structure implicit (Rühling Cachay et al., 2023), which can lead to unstable long-horizon trajectories as models must relearn consistency without physical constraints.

**Physics-informed Priors for Deep Learning**. Physics-inspired architectures encode conservation and dissipation structure (e.g., Hamiltonian formulations (Greydanus et al., 2019)) to provide stability-oriented inductive biases, with recent variants enforcing global Lyapunov stability by construction (Roth et al., 2025). Complementary works connect generative modeling to PDEs via first-principles loss functions that encourage samples to satisfy physical constraints (Bastek et al., 2025). Furthermore, neural Laplace models (Holt et al., 2022) enable solver-free evaluation for differential equations. LLapDiff unifies these threads by applying diffusion to latent trajectories and utilizing a stable modal parameterization to achieve horizon-wide generation over irregular grids, enabling forecasting while also supporting imputation via trajectory queries at queried timestamps.

## 3. Problem Formulation

**Preliminary**. Consider $N$ entities observed at some physical timestamps $\{t_j\}_{j=1}^{\mathcal{T}}$. At each observation time $t_j$, we have inputs $\boldsymbol{X}_{t_j} \in \mathbb{R}^{N \times d_x}$, targets (often a subset of inputs) $\boldsymbol{Y}_{t_j} \in \mathbb{R}^{N \times d_y}$, and their masks $\boldsymbol{M}_{t_j}^x \in \{0,1\}^{N \times d_x}$, $\boldsymbol{M}_{t_j}^y \in \{0,1\}^{N \times d_y}$. For a timestamp $t_i$, we define the observed history $\mathcal{H}_{t_i} = \{(t_j, \boldsymbol{X}_{t_j}, \boldsymbol{M}_{t_j}^x)\}_{j=i-\ell+1}^{i}$ and targets $\mathcal{Y}_{t_i} = \{(t_r^q, \boldsymbol{Y}_{t_r^q}, \boldsymbol{M}_{t_r^q}^y)\}_{r=1}^{h}$ corresponding to $h$ queries $\{t_r^q\}_{r=1}^{h}$. These queries may be future points (forecasting) or historical timestamps with missing entries (imputation).

**Latent target representation**. We represent the target as a low-dimensional latent trajectory $\boldsymbol{z} \in \mathbb{R}^{h \times d_z}$ on a latent manifold. During training, we encode ground-truth targets $\boldsymbol{z} = \text{VAE}_{\text{enc}}(\mathcal{Y}_{t_i})$ with a pretrained VAE, and learn a diffusion model $p_\theta(\boldsymbol{z}|\boldsymbol{E}_{t_i})$ conditioned on history summary $\boldsymbol{E}_{t_i} = \mathcal{S}_\phi(\mathcal{H}_{t_i})$. At inference, we sample $\hat{\boldsymbol{z}}$ and decode $\hat{\mathcal{Y}}_{t_i} = \text{VAE}_{\text{dec}}(\hat{\boldsymbol{z}})$, interpreting length-$h$ latent sequence at query times $\{t_r^q\}_{r=1}^{h}$ aligned to the native resolution via offsets $\tilde{t}_r := t_r^q - t_1^q$. As generation conditions only on context $\mathcal{H}_{t_i}$ ending at $t_i$ (no observations $> t_i$), we can also impute queries with $t_r^q \leq t_i$. Pretraining details are in Appendix E.

## 4. Stability Bias: Stochastic Port-Hamiltonian

We use the stochastic port-Hamiltonian formulation to motivate a stability-inducing inductive bias for denoising. Its local linearization yields a Laplace-domain characterization of

mean dynamics, parameterized by stable complex-conjugate poles to avoid time stepping and dense exponentials.

## 4.1. Stochastic Port-Hamiltonian Formulation

To encode a passivity-based inductive bias and discourage energy growth, we thus impose a stability-oriented inductive bias by modeling the latent evolution via a stochastic port-Hamiltonian structure. Let $\boldsymbol{x}_t \in \mathbb{R}^{d_z}$ be an auxiliary Hamiltonian state and $H(\boldsymbol{x}_t; \psi_t)$ a context-conditioned energy function, where $\psi_t$ is a history-dependent context. We consider the following port-Hamiltonian SDE (Itô form),

$$d\boldsymbol{x}_t = \underbrace{\left[(\boldsymbol{J} - \boldsymbol{R})\nabla_{\boldsymbol{x}}H(\boldsymbol{x}_t; \psi_t) + \boldsymbol{G}(\psi_t)\boldsymbol{u}_t\right]}_{\text{Drift } f} dt + \boldsymbol{\Sigma}_t d\boldsymbol{W}_t,$$
(1)

where $\boldsymbol{J}^\top = -\boldsymbol{J}$ is the interconnection matrix, $\boldsymbol{R} \succ 0$ the dissipation matrix, $\boldsymbol{G}$ the input matrix, and $\boldsymbol{\Sigma}_t d\boldsymbol{W}_t$ the stochastic variations. The port-collocated output is defined by,

$$\tilde{\boldsymbol{y}}_t = \boldsymbol{G}^\top(\psi_t)\nabla_{\boldsymbol{x}}H(\boldsymbol{x}_t; \psi_t).$$
(2)

Applying Itô's lemma (Oksendal, 2013) to $H(\boldsymbol{x}_t; \psi_t)$ and taking expectations, this gives the average energy balance,

$$\frac{d}{dt}\mathbb{E}[H] = \mathbb{E}\left[(\nabla_\psi H)^\top \dot{\psi}_t\right] - \mathbb{E}[\nabla_{\boldsymbol{x}}H^\top \boldsymbol{R}\nabla_{\boldsymbol{x}}H]$$
$$+ \mathbb{E}[\tilde{\boldsymbol{y}}_t^\top \boldsymbol{u}_t] + \frac{1}{2}\mathbb{E}\left[\text{tr}\left(\boldsymbol{\Sigma}_t \boldsymbol{\Sigma}_t^\top \nabla_{\boldsymbol{x}}^2 H\right)\right]. \quad (3)$$

Here, Eq. (3) is the instantaneous form of the energy balance induced by Eq. (1) and holds for almost every $t$, independent of the sampling grid. We treat $\psi_t$ as time-varying and piecewise-constant, thus the context term vanishes between observation updates (updates contribute only discrete energy jumps). Accordingly, in the absence of external inputs and noise, the expected system energy is non-increasing between context updates. This passivity-based structure isolates forecastable dynamics from stochastic volatility, which justifies a stable and dissipative nature for the learned mean dynamics. Derivations are provided in Appendix A.

**Link to latent diffusion variables.** We leverage stochastic port-Hamiltonian prior (mean energy balance) for latent diffusion: the Hamiltonian state corresponds to the clean latent trajectory evaluated at query times, $\boldsymbol{x}_t \equiv \boldsymbol{z}_0(t)$, while diffusion variables $\boldsymbol{z}_\tau(t)$ are noisy versions denoised toward $\boldsymbol{z}_0(t)$. The energy balance in Eq. (3) thus motivates mean latent dynamics that discourage energy growth between context updates. In forecasting, future $\boldsymbol{u}_t$ is unobservable; instead, history summary (Sec. 5.1) provides a latent port context that sets initial residues (stored energy) and conditions the poles governing subsequent autonomous transients.

## 4.2. Laplace-Domain Mean Dynamics

Although Eq. (1) offers a stability-inducing reference, it still requires time-domain integration on irregular grids. To by-

pass this, we derive a local Laplace-domain characterization of its mean dynamics. At reference time $t_0$, we consider a nominal operating point $(\bar{\boldsymbol{x}}_{t_0}, \bar{\boldsymbol{u}}_{t_0})$ with negligible residual drift, $f(\bar{\boldsymbol{x}}_{t_0}; \psi_{t_0}, \bar{\boldsymbol{u}}_{t_0}) \approx 0$, and freeze coefficients locally $\psi_t \approx \psi_{t_0}$ and $\boldsymbol{\Sigma}_t \approx \boldsymbol{\Sigma}_{t_0}$. For a physical interpretation, we analyze the unforced case ($\delta\boldsymbol{u} \equiv 0$) under strict dissipation (i.e., $\boldsymbol{R} \succ 0$), where any zero-drift equilibrium is an energy critical point such that $\nabla_{\boldsymbol{x}}H = 0$. This justifies focusing on stable equilibria (energy wells) with locally dissipative mean dynamics, where the modal poles correspond to eigenvalues of the local Jacobian. Here, we define deviations as $\delta\boldsymbol{x}_t := \boldsymbol{x}_t - \bar{\boldsymbol{x}}_{t_0}$ and $\delta\boldsymbol{u}_t := \boldsymbol{u}_t - \bar{\boldsymbol{u}}_{t_0}$, which yields,

$$\mathrm{d}\delta\boldsymbol{x}_t = f(\bar{\boldsymbol{x}}_{t_0} + \delta\boldsymbol{x}_t; \psi_{t_0}, \bar{\boldsymbol{u}}_{t_0} + \delta\boldsymbol{u}_t)dt + \boldsymbol{\Sigma}_{t_0}d\boldsymbol{W}_t. \quad (4)$$

This locally linearizes to,

$$d\delta\boldsymbol{x}_t = \boldsymbol{A}\delta\boldsymbol{x}_t dt + \boldsymbol{B}\delta\boldsymbol{u}_t dt + \boldsymbol{\Sigma}_{t_0}d\boldsymbol{W}_t. \quad (5)$$

Here, $\boldsymbol{A} = (\boldsymbol{J} - \boldsymbol{R})\nabla_{\boldsymbol{x}}^2 H(\bar{\boldsymbol{x}}_{t_0}; \psi_{t_0})$ denotes the local Jacobian and $\boldsymbol{B} \approx \boldsymbol{G}(\psi_{t_0})$ the linearized input matrix. Accordingly, the mild solution decomposes into,

$$\delta\boldsymbol{x}_t = \mathrm{e}^{\boldsymbol{A}(t-t_0)}\delta\boldsymbol{x}_{t_0} + \int_{t_0}^t \mathrm{e}^{\boldsymbol{A}(t-r)}\boldsymbol{B}\delta\boldsymbol{u}_r dr$$
$$+ \int_{t_0}^t \mathrm{e}^{\boldsymbol{A}(t-r)}\boldsymbol{\Sigma}_{t_0}d\boldsymbol{W}_r. \quad (6)$$

Taking expectations over the Brownian noise eliminates the martingale term, yielding the mean dynamics,

$$\mathbb{E}[\delta\boldsymbol{x}_t] = \mathrm{e}^{\boldsymbol{A}(t-t_0)}\mathbb{E}[\delta\boldsymbol{x}_{t_0}] + \int_{t_0}^t \mathrm{e}^{\boldsymbol{A}(t-r)}\boldsymbol{B}\,\delta\boldsymbol{u}_r dr. \quad (7)$$

Instead of restricting the readout to the port-collocated output in Eq. (2), we use a generic latent linearized readout,

$$\delta\boldsymbol{y}_t \approx \boldsymbol{C}\,\delta\boldsymbol{x}_t, \ \boldsymbol{C} \in \mathbb{R}^{d_z \times d_z}, \quad (8)$$

where $\boldsymbol{C}$ is a projection to the latent space. Consequently, the mean output decomposes into the following terms,

$$\mathbb{E}[\delta\boldsymbol{y}_t] = \boldsymbol{C}\mathrm{e}^{\boldsymbol{A}(t-t_0)}\mathbb{E}[\delta\boldsymbol{x}_{t_0}] + (\boldsymbol{g} * \delta\boldsymbol{u})(t), \quad (9)$$

where $\boldsymbol{g}(t) = \boldsymbol{C}\mathrm{e}^{\boldsymbol{A}t}\boldsymbol{B}$ denotes the Green's function. With no future exogenous inputs ($\delta\boldsymbol{u}(t) \approx 0$ for $t > t_0$), generation reduces to an autonomous evolution from the history-conditioned state at $t_0$, so the forcing term $(\boldsymbol{g} * \delta\boldsymbol{u})$ vanishes for $t > t_0$ and past inputs are already absorbed into $\mathbb{E}[\delta\boldsymbol{x}_{t_0}]$. To parameterize poles governing this decay, we work with the Laplace-domain resolvent,

$$\mathcal{G}(s) = \text{Lap}\{\boldsymbol{g}\}(s) = \boldsymbol{C}(s\boldsymbol{I} - \boldsymbol{A})^{-1}\boldsymbol{B}. \quad (10)$$

This provides a Laplace-domain characterization that captures transients and long-range decay beyond stationary Fourier analysis. Rather than time stepping in physical time, we condition poles $(\rho, \omega)$ on the history summary $\boldsymbol{E}_{t_i}$ to obtain a window-specific Jacobian-equivalent representation that adapts to non-stationarity while enforcing stable poles. Derivations are provided in Appendix B.

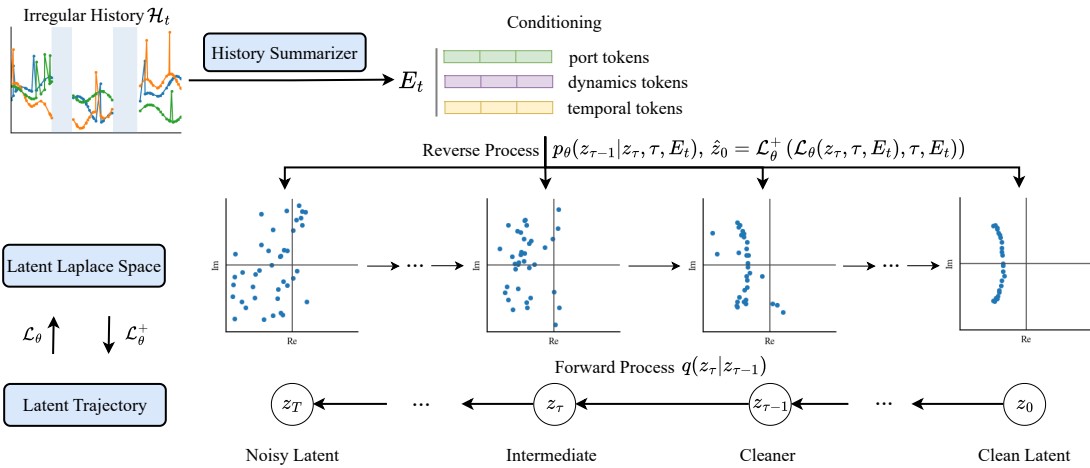

*Figure 1.* Overview of the proposed Latent Laplace Diffusion. The observed history $\mathcal{H}_{t_i}$ is summarized into a conditioning token sequence $\boldsymbol{E}_{t_i}$. The forward diffusion corrupts the clean latent trajectory from $\boldsymbol{z}_0$ to $\boldsymbol{z}_T$. In the reverse process, modal predictor $\mathcal{L}_\theta$ estimates modal parameters and modal synthesizer $\mathcal{L}_\theta^+$ generates the denoised latent estimate $\hat{\boldsymbol{z}}_0$ to update $\boldsymbol{z}_{\tau-1}$ based on the predicted modal parameters.

## 4.3. Stable Modal Parameterization

For implementation, we do not directly compute the local Jacobian/Hessian in Sec. 4.2. Instead, we learn its equivalent modal realization inspired by partial-fraction forms of the resolvent (Gustavsen & Semlyen, 2002). Since local stochastic port-Hamiltonian dynamics often exhibit damped oscillations, we represent $\mathcal{G}(s)$ using finite $K$ complex-conjugate pole pairs,

$$\mathcal{G}(s) = \sum_{k=1}^{K} \frac{\omega_k \, \boldsymbol{c}_k \boldsymbol{b}_k^\top}{s^2 + 2\rho_k s + (\rho_k^2 + \omega_k^2)}, \quad (11)$$

$$\boldsymbol{c}_k, \boldsymbol{b}_k \in \mathbb{R}^{d_z}, \ \rho_k, \omega_k > 0.$$

Here, $\rho_k, \omega_k$ denote decay and frequency rates. In our diffusion instantiation, $\boldsymbol{b}_k, \boldsymbol{c}_k$ serve as latent residue vectors (sine/cosine components). These vectors encode the projection of the history-determined state (including past forcing effects) onto the modal basis, of which the denoiser progressively refines them in modal space and utilizes them to synthesize the trajectory. This admits a canonical real state-space realization with a $2 \times 2$ block $\boldsymbol{A}_k$ having eigenvalues $-\rho_k \pm i\omega_k$,

$$\boldsymbol{A}_k = \begin{bmatrix} -\rho_k & -\omega_k \\ \omega_k & -\rho_k \end{bmatrix}, \boldsymbol{B}_k = \begin{bmatrix} 0 \\ 1 \end{bmatrix} \boldsymbol{b}_k^\top, \boldsymbol{C}_k = \boldsymbol{c}_k \begin{bmatrix} -1 & 0 \end{bmatrix}.$$

Therefore, we have the following forms for $\boldsymbol{A}, \boldsymbol{B}, \boldsymbol{C}$,

$$\boldsymbol{A} = \begin{bmatrix} \boldsymbol{A}_1 & \cdots & 0 \\ \vdots & \ddots & \vdots \\ 0 & \cdots & \boldsymbol{A}_K \end{bmatrix}, \boldsymbol{B} = \begin{bmatrix} \boldsymbol{B}_1 \\ \vdots \\ \boldsymbol{B}_K \end{bmatrix}, \boldsymbol{C} = [\boldsymbol{C}_1 \cdots \boldsymbol{C}_K],$$

which recovers Eq. (11). Under $\rho_k > 0$, $\boldsymbol{A}$ is Hurwitz, yielding exponentially decaying impulse responses and stable mean dynamics. Derivations are provided in Appendix B.

**Physics view**. This parameterization links the latent mean dynamics to damped oscillatory modes governed by local energy curvature $\nabla_{\boldsymbol{x}}^2 H$. The poles encode the equivalent interplay between interconnection and dissipation: the imaginary part $\omega_k$ corresponds to conservative oscillation frequencies (via $\boldsymbol{J}$ acting on the curvature), while the real part $\rho_k$ captures dissipative decay rates (effect of dissipation $\boldsymbol{R}$). Although we do not explicitly learn the full Hamiltonian structure, the theoretical derivation in Eq. (3) serves as the theoretical justification for enforcing strictly positive damping ($\rho_k > 0$). This guarantees exponential stability of the parameterized mean dynamics, providing stability bias for generation when the underlying dynamics are nonlinear.

## 4.4. Renewal Averaging View for Irregular Gaps

Bridging continuous-time latent dynamics with discrete observations requires accounting for stochastic sampling gaps $\Delta_j := t_j - t_{j-1} \geq 0$. We first analyze this via a stationary renewal process with i.i.d. intervals $\{\Delta_j\}$ (Cox, 1962), which offers a closed-form mapping between continuous-time poles and their effective event-domain counterparts.

Consider a single latent mode with continuous-time pole $s_k = -\rho_k + i\omega_k$ and $\zeta_j^{(k)} \in \mathbb{C}$ complex eigen-coordinate at event $j$. Under random sampling, $\zeta_{j+1}^{(k)} = e^{s_k \Delta_{j+1}} \zeta_j^{(k)}$, hence the mean evolution satisfies,

$$\mathbb{E}[\zeta_j^{(k)}] = \left(\mathbb{E}[e^{s_k \Delta}]\right)^j \zeta_0^{(k)} = \lambda_k^j \zeta_0^{(k)} = e^{\bar{s}_k j} \zeta_0^{(k)}, \quad (12)$$

where the effective discrete pole (mean multiplier) is $\lambda_k = \mathbb{E}[e^{s_k \Delta}]$. To differentiate from the continuous-time pole, we define the effective log-pole $\bar{s}_k := \log(\lambda_k) = -\bar{\rho}_k + i\bar{\omega}_k$, where $\log(\cdot)$ denotes the principal complex logarithm. Note $\bar{\omega}_k$ is a per-event phase increment, not an angular frequency

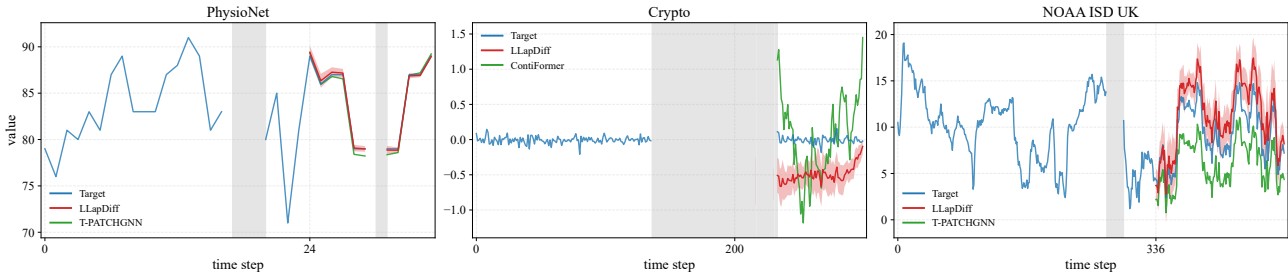

*Figure 2.* Qualitative forecast plots (one slice). **Red**: LLapDiff median results with 10%–90% predictive interval. **Green**: representative baseline averaged results. **Blue**: ground truth (target) lines. Gray bands mark timestamps where multiple missing entries are present.

per unit time. For nonstationary or history-dependent gaps, the same expression holds when conditioning on the available history, $\lambda_k(\cdot) = \mathbb{E}[e^{s_k \Delta} \mid \cdot]$. Equivalently, for the real $2 \times 2$ block $\boldsymbol{A}_k$, we represent the same mode in real coordinates by $\boldsymbol{\xi}_j^{(k)} = [\Re(\zeta_j^{(k)}), \Im(\zeta_j^{(k)})]^\top \in \mathbb{R}^2$. Renewal averaging yields,

$$\mathbb{E}[\boldsymbol{\xi}_j^{(k)}] = \Phi_k^j \, \boldsymbol{\xi}_0^{(k)}, \ \Phi_k := \mathbb{E}[e^{\boldsymbol{A}_k \Delta}]. \tag{13}$$

When $\Phi_k$ has a principal matrix logarithm (e.g., no eigenvalues on $\mathbb{R}_{\leq 0}$), we can write $\Phi_k = e^{\bar{\boldsymbol{A}}_k}$ and $\mathbb{E}[\boldsymbol{\xi}_j^{(k)}] = e^{\bar{\boldsymbol{A}}_k j} \boldsymbol{\xi}_0^{(k)}$ with $\bar{\boldsymbol{A}}_k := \log(\Phi_k)$. Since $\Delta \geq 0$ and $\Re(s_k) < 0$, by Jensen's inequality (equivalently, $|\mathbb{E}(\cdot)| \leq \mathbb{E}(|\cdot|)$),

$$|\lambda_k| = |\mathbb{E}[e^{s_k \Delta}]| \leq \mathbb{E}[|e^{s_k \Delta}|] \leq \mathbb{E}\left[e^{\Re(s_k)\Delta}\right] \leq 1, \tag{14}$$

so $\Re(\bar{s}_k) = \log|\lambda_k| \leq 0$ (and $\Re(\bar{s}_k) < 0$ when $\mathbb{P}(\Delta > 0) > 0$). Thus, renewal averaging maintains mean stability. Here, $\bar{s}_k = \log \lambda_k$ is an event-index log-multiplier. For oscillatory modes, random gaps can be more contractive due to phase cancellation, via an exponentially weighted characteristic factor with magnitude at most one. A second-order Taylor expansion provides the intuition,

$$\bar{s}_k = \log \mathbb{E}[e^{s_k \Delta}] \approx s_k \, \mathbb{E}[\Delta] + \tfrac{1}{2} s_k^2 \, \mathrm{Var}(\Delta), \tag{15}$$

which shows that gap variability modulates damping and oscillation. Since $\Re(s_k^2) = \rho_k^2 - \omega_k^2$, gap variance increases effective decay only in the underdamped regime $\omega_k > \rho_k$; in the overdamped regime $\omega_k < \rho_k$, the first-order correction flips sign. Derivations are provided in Appendix C.

**Renewal averaging view.** Analytically, Eqs. (12)-(15) offer a mapping from continuous-time poles to discrete ones under an i.i.d. assumption for tractability. In practice, sampling gaps are often unknown and non-i.i.d. (e.g., state-dependent). Critically, Eq. (15) shows that the effective event-domain log-pole $\bar{s}_k$ entangles the intrinsic pole $s_k$ with gap statistics. To infer intrinsic dynamics, the model must disentangle these factors. We treat the renewal formulation as a guiding architectural inductive bias: by conditioning on a gap-aware history summary, the model learns gap-robust continuous-time parameters $(\hat{\rho}_k, \hat{\omega}_k, \hat{\boldsymbol{c}}_k, \hat{\boldsymbol{b}}_k)$ that remain consistent even under complex and non-renewal sampling patterns.

## 5. Latent Laplace Diffusion

### 5.1. History Summary Conditioning

We instantiate the context $\psi_{t_i}$ via a history summarizer $\mathcal{S}_\phi$ that maps the observed history to a summary token sequence $\boldsymbol{E}_{t_i}$. This token sequence conditions the reverse diffusion process by encoding three complementary signals: (i) observed values acting as latent port signals, (ii) native-step finite-difference features capturing local state deviations, and (iii) sampling irregularity (gap size and variability) via timestamp / $\Delta t$ encodings and masks, which the renewal view links to effective damping. Concretely, at each timestep $j$ we form three tokens: port tokens $\boldsymbol{p}_j^{\text{tok}}$ from observed channels, dynamics tokens $\boldsymbol{v}_j^{\text{tok}}$ from native-step signal proxies, and temporal tokens $\boldsymbol{o}_j^{\text{tok}}$ from timestamp/$\Delta t$ encodings and masks, each produced by a lightweight projection head. These features are fused to form sequence $\{[\boldsymbol{p}_j^{\text{tok}} \| \boldsymbol{v}_j^{\text{tok}} \| \boldsymbol{o}_j^{\text{tok}}]\}_{j=i-\ell+1}^i$, which is fed into $\mathcal{S}_\phi$ to obtain $\boldsymbol{E}_{t_i}$. Implementations and pretraining are in Appendix E.2.

### 5.2. Denoising with Modal Dynamics

The core of LLapDiff is parameterizing the denoiser via two operations: a modal predictor $\mathcal{L}_\theta$ and a modal synthesizer $\mathcal{L}_\theta^+$. Implementations are provided in Appendix F.

**Modal predictor** $\mathcal{L}_\theta$ predicts continuous-time modal parameters that summarize the latent dynamics. Given noisy latent $\boldsymbol{z}_\tau$ and history summary $\boldsymbol{E}_{t_i}$ at diffusion step $\tau$,

$$\hat{\vartheta} := \mathcal{L}_\theta(\boldsymbol{z}_\tau, \tau, \boldsymbol{E}_{t_i}), \ \hat{\vartheta} = \{(\hat{\rho}_k, \hat{\omega}_k, \hat{\boldsymbol{c}}_k, \hat{\boldsymbol{b}}_k)\}_{k=1}^K.$$

These parameters (predicted per diffusion step $\tau$) are conditioned on $\boldsymbol{E}_{t_i}$, which encodes gaps and missingness motivated by renewal-averaging analysis of effective poles.

**Modal synthesizer** $\mathcal{L}_\theta^+$ generates denoised latent from $\hat{\vartheta}$ conditioned on $\boldsymbol{E}_{t_i}$ at $\tau$. For relative query times $\tilde{t}_r$ in the query window, directly evaluating analytical reconstruction,

$$\hat{\boldsymbol{z}}_0(\tilde{t}_r) = \left[\mathcal{L}_\theta^+(\hat{\vartheta}, \tau, \boldsymbol{E}_{t_i})\right]_r$$
$$= \sum_{k=1}^K e^{-\hat{\rho}_k \tilde{t}_r}\left(\hat{\boldsymbol{c}}_k \cos(\hat{\omega}_k \tilde{t}_r) + \hat{\boldsymbol{b}}_k \sin(\hat{\omega}_k \tilde{t}_r)\right).$$

*Table 1.* Forecasting results for four horizons: we report CRPS for probabilistic models and MAE for deterministic models (equivalent to CRPS of a Dirac), and MSE. All numbers are means over 10 runs. For probabilistic models, CRPS uses 25 samples; MSE uses the sample mean. The best scores are in **bold**, second-best ones are double underlined, and standard deviations are reported in Appendix I.

| Dataset | h | DLinear MAE | DLinear MSE | PatchTST MAE | PatchTST MSE | mr-Diff CRPS | mr-Diff MSE | TimeGrad CRPS | TimeGrad MSE | mTAN CRPS | mTAN MSE | T-PATCHGNN MAE | T-PATCHGNN MSE | ContiFormer MAE | ContiFormer MSE | NeuralCDE MAE | NeuralCDE MSE | LLapDiff CRPS | LLapDiff MSE |
|---|---|---|---|---|---|---|---|---|---|---|---|---|---|---|---|---|---|---|---|
| BMS Air | 24 | 1.123 | 2.700 | 0.936 | 1.306 | 0.579 | 1.382 | **0.458** | **0.884** | 0.481 | 1.000 | 0.781 | 1.140 | 0.923 | 1.765 | 0.883 | 1.630 | 0.491 | 1.180 |
| | 48 | 1.308 | 3.580 | 0.937 | 1.303 | 0.577 | 1.382 | **0.462** | **0.915** | 0.525 | 1.192 | 0.730 | 1.032 | 0.891 | 1.700 | 0.895 | 1.720 | 0.500 | 1.183 |
| | 96 | 1.432 | 4.130 | 0.936 | 1.304 | 0.580 | 1.382 | 0.506 | 1.083 | 0.539 | 1.286 | 0.683 | **0.977** | 0.949 | 1.904 | 1.008 | 2.216 | **0.504** | 1.195 |
| | 168 | 1.448 | 4.207 | 0.929 | 1.300 | 0.555 | 1.332 | 0.537 | 1.172 | 0.547 | 1.298 | 0.759 | **1.100** | 0.984 | 2.030 | 1.019 | 2.204 | **0.516** | 1.250 |
| UCI Air | 24 | 2.677 | 9.118 | 1.104 | **1.919** | 1.086 | 3.364 | 1.222 | 3.706 | **0.910** | 2.550 | 1.109 | 1.987 | 2.081 | 7.057 | 1.861 | 5.670 | 0.935 | 2.480 |
| | 48 | 2.643 | 9.001 | 1.092 | **1.984** | 0.992 | 2.812 | 1.238 | 3.811 | **0.834** | 2.166 | 1.133 | 2.051 | 2.121 | 7.340 | 1.816 | 5.419 | 0.936 | 2.378 |
| | 96 | 2.693 | 9.508 | 1.058 | **1.787** | 1.097 | 3.373 | 1.200 | 3.585 | 0.967 | 2.593 | 1.228 | 2.226 | 2.091 | 7.138 | 2.046 | 6.808 | **0.938** | 2.326 |
| | 168 | 2.751 | 9.904 | 1.149 | 2.198 | 1.174 | 3.813 | 1.122 | 3.117 | **0.836** | 2.392 | 1.117 | **1.997** | 2.143 | 7.453 | 1.991 | 6.326 | 1.003 | 2.865 |
| PhysioNet | 4 | 0.463 | 0.691 | 0.472 | 0.718 | 0.421 | 0.809 | 0.433 | 0.848 | 0.439 | 0.844 | 0.373 | **0.564** | 0.407 | 0.633 | 0.418 | 0.638 | **0.338** | 0.650 |
| | 8 | 0.468 | 0.700 | 0.477 | 0.726 | 0.426 | 0.817 | 0.437 | 0.857 | 0.444 | 0.853 | 0.397 | **0.583** | 0.400 | 0.629 | 0.411 | 0.640 | **0.337** | 0.642 |
| | 10 | 0.472 | 0.709 | 0.481 | 0.733 | 0.388 | 0.776 | 0.441 | 0.866 | 0.448 | 0.863 | 0.401 | **0.589** | 0.416 | 0.646 | 0.448 | 0.675 | **0.330** | 0.670 |
| | 12 | 0.476 | 0.718 | 0.486 | 0.741 | 0.434 | 0.835 | 0.446 | 0.875 | 0.452 | 0.872 | 0.385 | **0.579** | 0.420 | 0.652 | 0.431 | 0.657 | **0.318** | 0.639 |
| NOAA US | 24 | 0.352 | 0.209 | 0.325 | **0.197** | 0.370 | 0.361 | 0.451 | 0.538 | **0.254** | 0.242 | 0.357 | 0.229 | 0.484 | 0.349 | 0.455 | 0.317 | 0.432 | 0.463 |
| | 48 | 0.354 | 0.210 | 0.314 | **0.186** | 0.369 | 0.360 | 0.452 | 0.541 | **0.253** | 0.241 | 0.363 | 0.235 | 0.497 | 0.368 | 0.471 | 0.335 | 0.433 | 0.465 |
| | 96 | 0.355 | 0.212 | 0.334 | **0.207** | 0.370 | 0.362 | 0.453 | 0.543 | **0.239** | 0.224 | 0.363 | 0.231 | 0.489 | 0.357 | 0.509 | 0.384 | 0.433 | 0.462 |
| | 168 | 0.355 | 0.213 | 0.333 | 0.206 | 0.372 | 0.364 | 0.454 | 0.545 | **0.255** | 0.244 | 0.328 | **0.198** | 0.468 | 0.346 | 0.511 | 0.386 | 0.440 | 0.455 |
| NOAA UK | 24 | 1.457 | 3.219 | 0.735 | **0.714** | 0.859 | 1.796 | 0.645 | 0.946 | 0.900 | 1.887 | 0.778 | 0.789 | 1.326 | 2.541 | 1.084 | 1.812 | **0.559** | 1.062 |
| | 48 | 1.650 | 3.985 | 0.746 | **0.738** | 0.836 | 1.708 | 0.633 | 0.915 | 0.936 | 2.014 | 0.817 | 0.881 | 1.283 | 2.402 | 1.100 | 1.855 | **0.559** | 1.082 |
| | 96 | 1.576 | 3.646 | 0.751 | **0.745** | 0.862 | 1.802 | 0.636 | 0.921 | 0.935 | 2.018 | 0.844 | 0.822 | 1.327 | 2.586 | 1.091 | 1.829 | **0.560** | 1.098 |
| | 168 | 1.546 | 3.506 | 0.750 | **0.740** | 0.879 | 1.864 | 0.639 | 0.929 | 0.869 | 1.800 | 0.823 | 0.908 | 1.354 | 2.660 | 1.114 | 1.891 | **0.557** | 1.086 |
| US Equity | 5 | 0.567 | 0.673 | 0.572 | 0.658 | 0.421 | 0.768 | 0.425 | 0.817 | 0.418 | 0.758 | 0.575 | **0.649** | 0.566 | 0.658 | 0.573 | 0.655 | **0.417** | 0.708 |
| | 20 | 0.575 | 0.670 | 0.571 | 0.655 | 0.419 | 0.762 | 0.424 | 0.813 | 0.418 | 0.756 | 0.586 | 0.670 | 0.563 | **0.651** | 0.568 | 0.657 | **0.415** | 0.706 |
| | 60 | 0.571 | 0.655 | 0.566 | 0.639 | 0.416 | 0.744 | 0.422 | 0.794 | 0.413 | 0.733 | 0.564 | 0.645 | 0.563 | 0.640 | 0.560 | 0.662 | **0.401** | 0.679 |
| | 100 | 0.572 | 0.655 | 0.565 | 0.643 | 0.417 | 0.752 | 0.423 | 0.810 | 0.417 | 0.742 | 0.562 | **0.638** | 0.563 | 0.646 | 0.561 | 0.666 | **0.406** | 0.695 |
| Cryptos | 5 | 0.496 | 0.602 | 0.485 | 0.569 | 0.367 | 0.664 | **0.353** | 0.654 | 0.382 | 0.698 | 0.421 | **0.489** | 0.461 | 0.537 | 0.461 | 0.537 | 0.367 | 0.636 |
| | 20 | 0.489 | 0.579 | 0.473 | 0.540 | 0.366 | 0.635 | **0.343** | 0.620 | 0.359 | 0.658 | 0.443 | 0.508 | 0.449 | **0.508** | 0.449 | 0.508 | 0.357 | 0.624 |
| | 60 | 0.475 | 0.540 | 0.469 | 0.508 | 0.381 | 0.711 | **0.334** | 0.579 | 0.353 | 0.604 | 0.393 | **0.410** | 0.435 | 0.474 | 0.462 | 0.473 | 0.351 | 0.581 |
| | 100 | 0.472 | 0.532 | 0.465 | 0.524 | 0.358 | 0.610 | **0.356** | 0.623 | 0.364 | 0.627 | 0.461 | 0.520 | 0.437 | 0.473 | 0.452 | 0.485 | 0.350 | 0.563 |
| Avg. rank | | 7.9±2.0 | 6.2±2.5 | 5.8±2.5 | 3.0±1.9 | 4.1±1.1 | 6.5±1.1 | 3.8±2.2 | 6.3±2.9 | 3.3±2.1 | 5.9±2.1 | 4.7±1.8 | 1.9±1.0 | 6.6±1.8 | 5.0±2.7 | 6.8±1.3 | 5.2±2.0 | 2.1±1.7 | 4.9±1.8 |

The learned poles impose dynamical consistency for long-range behaviors and history-conditioned tokens modulate residues to capture nonlinear transients; optionally, $\hat{z}_0 \leftarrow \hat{z}_0 + \mathrm{MLP}(\hat{z}_0)$ corrects local deviations beyond modal basis to yield coherent trajectories. Unlike latent ODE-style models requiring sequential integration, $\hat{z}_0(\tilde{t}_r)_{r=1}^{h}$ is computed in one parallelizable pass: once modal parameters are predicted, we evaluate the closed-form sum at all query times. For imputation, we add missing within-window timestamps to the query set and synthesize them using the same summary as forecasting (i.e., content up to $t_i$), enabling causal (filtering-style) imputation rather than bidirectional smoothing of CSDI-style methods conditioning on the full window.

### 5.3. Training and Inference

**Training**: We adopt the DDPM forward process (Ho et al., 2020) on the latent trajectory, indexed by diffusion step $\tau \in \{0, \ldots, T\}$. Let $z_0 := z$ denote the clean latent trajectory at $\tau = 0$. The forward process is,

$$q(z_\tau|z_0) = \mathcal{N}(\sqrt{\bar{\alpha}_\tau}z_0, (1 - \bar{\alpha}_\tau)I), \ \epsilon \sim \mathcal{N}(0, I)$$
$$z_\tau = \sqrt{\bar{\alpha}_\tau}z_0 + \sqrt{1 - \bar{\alpha}_\tau}\epsilon. \tag{16}$$

where $\bar{\alpha}_\tau = \prod_{m=1}^{\tau} \alpha_m$ and $\alpha_m = 1 - \beta_m$. We apply classifier-free guidance by dropping conditioning with probability $p_{\mathrm{uncond}}$ during training and applying guidance weight $w$ at inference. We adopt the standard $x_0$-parameterization

and train by minimizing mean-squared reconstruction error,

$$\mathcal{J}(\theta) = \mathbb{E}\Big[\|z_0 - \hat{z}_0(z_\tau, \tau, E_{t_i})\|_2^2\Big]. \tag{17}$$

where $\hat{z}_0(z_\tau, \tau, E_{t_i}) = \mathcal{L}_\theta^+(\mathcal{L}_\theta(z_\tau, \tau, E_{t_i}), \tau, E_{t_i})$.

**Inference**: We sample $z_T \sim \mathcal{N}(0, I)$ and iteratively denoise to obtain $\hat{z}_0$ using a deterministic sampler (DDIM-style (Song et al., 2021a)) under $x_0$-parameterization. The training/inference algorithms, hyperparameters, and inference complexity analysis are provided in Appendix G, Appendix H, and Appendix I.

## 6. Experiments

### 6.1. Experimental Setting

**Datasets & Settings**: We evaluate the models over seven real-world datasets (one regular and six irregular) spanning air quality, healthcare, meteorology, and finance: UCI Air (De Vito et al., 2008), BMS Air (Zhang et al., 2017), PhysioNet (Goldberger et al., 2000), NOAA (UK/US), Crypto, and US Equity. The dataset collection, preprocessing, and statistics are provided in Appendix D. We focus on long-horizon forecasting as the primary evaluation (chronological split for train:val:test is 0.7:0.1:0.2), where imputation examples are shown as a by-product in Fig. 3. We pretrain VAE and history summarizer on train split (val split for model selection) and keep them frozen during diffusion training.

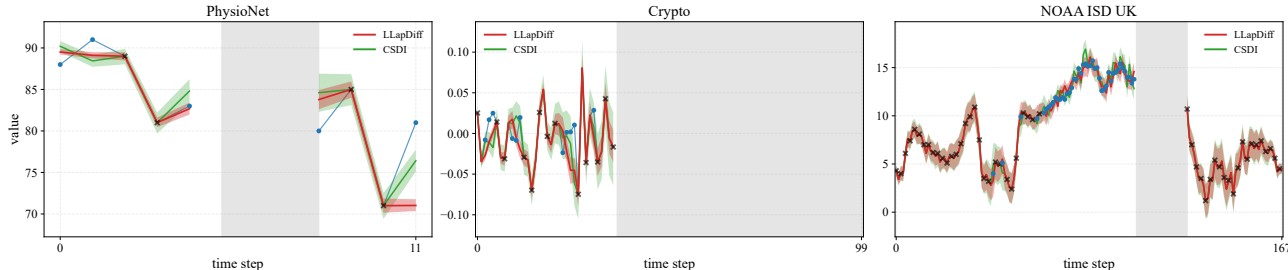

*Figure 3.* Probabilistic imputation results over historical queried timestamps. The dark crosses are observed values, and blue dots are artificially masked (30% masked) ground-truth targets. Red lines and green lines represent LLapDiff / CSDI median results (CRPS: $0.321_{\pm 0.04}/0.469_{\pm 0.03}$, $0.339_{\pm 0.01}/0.282_{\pm 0.01}$, $0.502_{\pm 0.07}/0.294_{\pm 0.04}$) over 10 runs with 10%–90% predictive interval.

*Table 2.* Ablations on training methods, where reported CRPS results are averaged over longest horizons ($\downarrow$). Values in parentheses are deltas relative to Full.

| Method | BMS Air | NOAA US | US Equity |
|---|---|---|---|
| Full | **0.516**(+0.00) | **0.440**(+0.00) | **0.406**(+0.00) |
| w/o conditioning | 0.816(+0.30) | 1.450(+1.01) | 0.466(+0.06) |
| w/o learned poles | 0.696(+0.18) | 1.310(+0.87) | 0.476(+0.07) |
| w/o latent space | 0.666(+0.15) | 1.030(+0.59) | 0.446(+0.04) |
| joint-trained summarizer | 0.806(+0.29) | 1.360(+0.92) | 0.476(+0.07) |

*Table 3.* Ablations on tokens forming history summary token sequence, where reported CRPS results are averaged over longest horizons ($\downarrow$). Values in parentheses are deltas relative to Full.

| Method | PhysioNet | NOAA UK | Crypto |
|---|---|---|---|
| Full | **0.318**(+0.00) | **0.557**(+0.00) | **0.350**(+0.00) |
| w/o temporal token | 0.498(+0.18) | 1.837(+1.28) | 0.400(+0.05) |
| w/o dynamics token | 0.478(+0.16) | 1.037(+0.48) | 0.420(+0.07) |
| w/o port token | 0.448(+0.13) | 1.247(+0.69) | 0.380(+0.03) |

*Table 4.* Stress test under manually induced missingness on the longest forecast horizons (CRPS). The performance deltas are relative to 0% Min Coverage (minimum per-timestamp coverage).

| Min Cov. | PhysioNet | NOAA UK | NOAA US | Crypto |
|---|---|---|---|---|
| 0% | **0.318**(+0.00) | **0.557**(+0.00) | **0.440**(+0.00) | **0.350**(+0.00) |
| 20% | 0.388(+0.07) | 0.577(+0.02) | 0.440(+0.00) | 0.350(+0.00) |
| 40% | 0.458(+0.14) | 0.627(+0.07) | 0.470(+0.03) | 0.350(+0.00) |
| 60% | 0.478(+0.16) | 0.807(+0.25) | 0.590(+0.15) | 0.360(+0.01) |
| 80% | 0.488(+0.17) | 0.877(+0.32) | 0.900(+0.46) | 0.380(+0.03) |

training, LLapDiff reconstructs artificially masked historical targets by querying the same model at missing timestamps, indicating that the learned modal dynamics transfer naturally from forecasting to entry-level imputation. While the relative gains shrink in dense, near-regular regimes (e.g., BMS Air at $h = 24$, UCI Air at $h = 168$), where high coverage (see Appendix Tab. 9) reduces the benefit of gap-aware conditioning and continuous-time structure.

### 6.3. Ablation Study

Tab. 2 supports our piecewise-constant context design: $\psi_t$ is updated only at observation times and held fixed between them, so performance hinges on learning a reliable update rule. As removing conditioning causes severe degradation (e.g., +1.01 CRPS on NOAA US), and jointly training the summarizer performs markedly worse than pretraining (+0.92), indicating the difficulty of simultaneously learning meaningful contexts and dynamics under high sparsity. Removing learned poles (+0.87) or the latent trajectory (+0.59) also degrades performance, confirming that the continuous-time spectral bias and smooth latent manifold are critical for preventing error accumulation over long horizons. While Tab. 2 identifies conditioning as essential, it does not reveal which aspects drive the gains; Tab. 3 ablates the summarizer tokens and shows that gap information is the dominant factor. In particular, removing the temporal token that encodes sampling irregularity yields the larger error increases (e.g., +1.28 on NOAA UK), whereas dynamics and port tokens provide consistent improvements under irregular sampling.

**Baselines**: We compare against strong deterministic models (PatchTST (Nie et al., 2023), DLinear (Zeng et al., 2023)), diffusion-based models (CSDI (Tashiro et al., 2021) only for imputation comparison, TimeGrad (Rasul et al., 2021), mr-Diff (Shen et al., 2024)) and IMTS models (Neural-CDE (Chen et al., 2018), mTAN (Shukla & Marlin, 2021), ContiFormer (Chen et al., 2023), T-PATCHGNN (Zhang et al., 2024)). The baseline irregularity handling details are provided in Appendix D.

### 6.2. Main results

Tab. 1 shows that LLapDiff achieves the best distributional metric and competitive point metric, with gains most reliable at longer horizons and on more irregular datasets (e.g., NOAA UK at $h=168$, PhysioNet at $h=12$). Fig. 2 further supports this trend, showing coherent trajectories and well-calibrated uncertainty that remain stable through gray bands (multiple missingness occurred), where most baselines more often drift or oversmooth. Furthermore, Fig. 3 highlights the broader utility of the trajectory-generation view: without re-

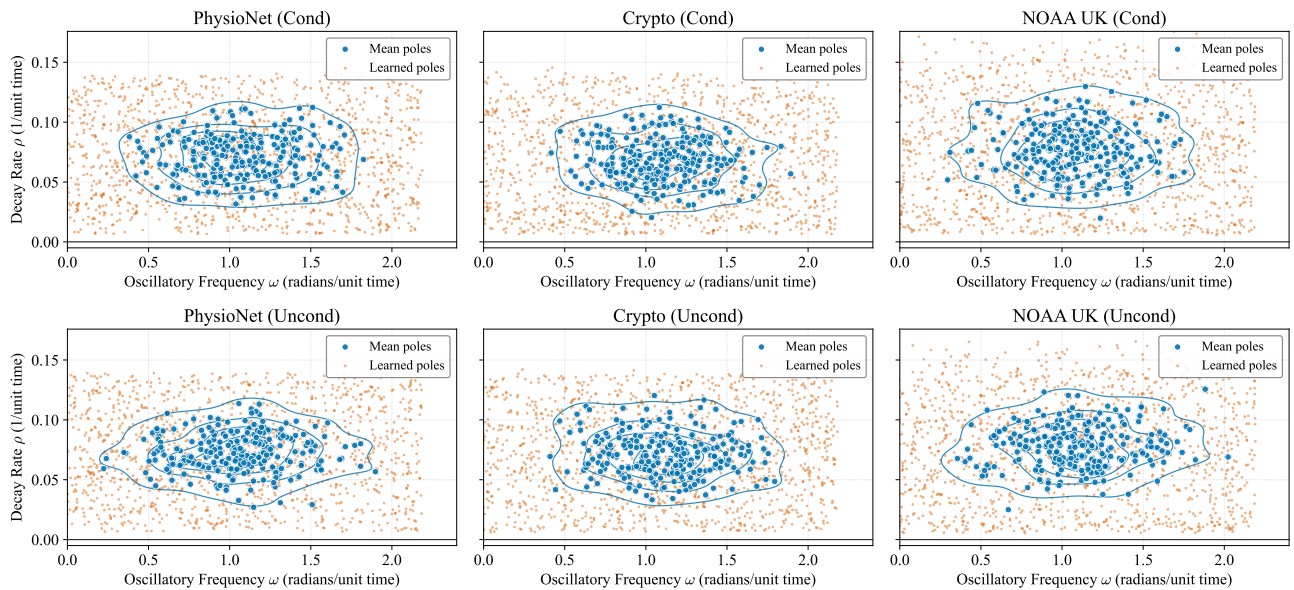

*Figure 4.* Illustrations of learned poles (mean poles are averaged over learned poles). **Pole stats (mean, std)**: PhysioNet cond $(\mu_\omega, \sigma_\omega) = (1.068, 0.299)$, $(\mu_\rho, \sigma_\rho) = (0.071, 0.017)$; uncond $(1.054, 0.301)$, $(0.073, 0.016)$; Crypto cond $(1.095, 0.264)$, $(0.068, 0.017)$; uncond $(1.113, 0.286)$, $(0.071, 0.017)$; NOAA-UK cond $(1.076, 0.278)$, $(0.076, 0.020)$; uncond $(1.093, 0.301)$, $(0.077, 0.017)$.

*Table 5.* Inference wall-clock time on NOAA-US/UK (ms) across horizons. Lower is better.

| Method | 24 | 48 | 96 | 168 | Avg. | slow ↓ |
|---|---|---|---|---|---|---|
| LLapDiff | **449** | **449** | **451** | **451** | **450** | **1.00×** |
| NeuralCDE | 553 | 560 | 560 | 554 | 557 | 1.24× |
| MRDiff | 890 | 888 | 886 | 886 | 888 | 1.97× |
| TimeGrad | 5107 | 10226 | 20452 | 35774 | 17890 | 39.8× |
| ContiFormer | 21000 | 40000 | 80000 | 130000 | 67750 | 150.60× |

*Table 6.* Reverse-step sensitivity at longest horizons: CRPS / median time (ms). Lower is better.

| DDIM steps | NOAA-US | BMS Air |
|---|---|---|
| 16 | 0.4554 / 151.69 | 0.5338 / 156.86 |
| 32 | 0.4480 / 257.54 | 0.5260 / 261.86 |
| 64 | 0.4400 / 456.32 | 0.5160 / 446.21 |
| 128 | 0.4412 / 802.21 | 0.5173 / 793.88 |

## 6.4. Visualization of Learned Poles

To validate the stable modal parameterization, Fig. 4 shows learned continuous-time poles in the quasi-frequency/decay plane $(\omega, \rho)$ under irregular sampling. **Stability**: all learned poles satisfy $\rho > 0$, guaranteeing exponential decay of each modal basis via constrained damping. The decay rates concentrate around $\mu_\rho \approx 0.07$ with $\sigma_\rho \approx 0.017$, placing most mass several $\sigma$ away from the instability boundary ($\rho = 0$) and suggesting a margin from the instability boundary in our learned prior. **Gap adaptation**: conditional (top) and unconditional (bottom) distributions shift noticeably; conditioning increases decay variability (e.g., NOAA UK $\sigma_\rho : 0.017 \rightarrow 0.020$), which suggests the summarizer modulates the learned damping/decay rates to match observed gap patterns, consistent with the renewal-averaging motivation. **Spectral diversity**: while mean poles remain anchored in a stable region, the learned poles span a broad frequency range $\omega \in [0, 2.5]$, indicating that LLapDiff decouples a stable structural prior from the spectral richness needed to capture high-frequency dynamics.

## 6.5. Stress Test under Induced Missingness

We stress-test the robustness to amplified gaps by dropping timestamps (during training and testing) whose coverage falls below a threshold. As the coverage threshold increases, Tab. 4 shows elegant degradation: the largest sensitivity is on NOAA US (CRPS: $0.440 \rightarrow 0.900$, $+0.46$), while NOAA UK and PhysioNet increase moderately ($+0.32$ and $+0.17$). Crypto remains nearly unchanged up to $60\%$ coverage ($+0.01$) and remains stable at $80\%$ ($+0.03$), consistent with its higher base coverage. This indicates failures are likely driven more by loss of uniquely informative channels than by gaps or missingness alone. Overall, results support our continuous-time latent dynamics as a robust backbone.

## 6.6. Inference Complexity

LLapDiff is not one-shot: it denoises over diffusion noise levels (see Tab. 6). Its efficiency comes from avoiding numerical integration over physical time. After the continuous modal parameters are predicted, all query timestamps are evaluated by a closed-form modal sum in parallel, making

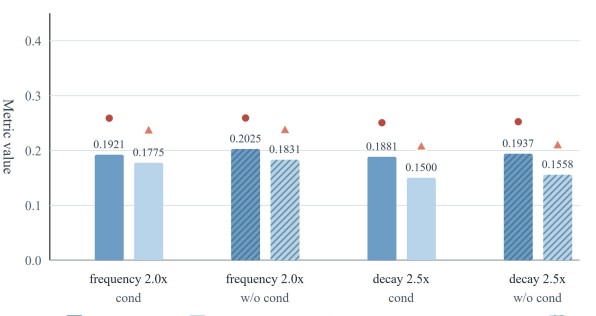
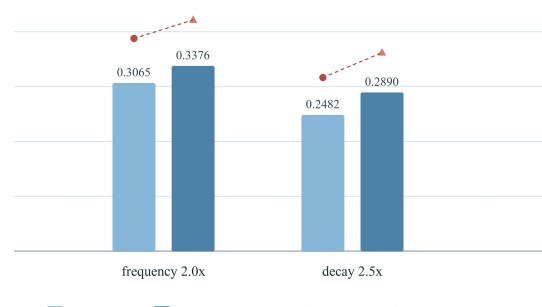

*Figure 5.* Controlled regime shifts tests with synthetic datasets. **Left**: Stricter unseen-regime split comparing adaptive and fixed poles. **Right**: Boundary-crossing robustness under severe frequency/decay shifts.

*Table 7.* Inference time (ms) vs. number of poles. Mean inference time over horizons 24/48/96/168; parentheses show min–max.

| $K$ | NOAA-US / NOAA-UK | | BMS Air / UCI Air | |
|---|---|---|---|---|
| 16 | 454.14 | (453.31–454.49) | 452.59 | (452.16–452.82) |
| 32 | 452.56 | (451.85–454.04) | 453.33 | (452.49–454.11) |
| 64 | 454.73 | (453.14–456.59) | 455.52 | (454.74–456.85) |
| 128 | 455.38 | (454.88–456.10) | 455.30 | (454.82–455.83) |
| 256 | 454.29 | (453.80–454.74) | 454.58 | (454.08–455.14) |
| 512 | 455.00 | (454.18–456.55) | 455.66 | (454.41–456.21) |
| Avg. | 454.35 | (451.85–456.59) | 454.50 | (452.16–456.85) |

*Table 8.* Latent-width sensitivity. CRPS at $h = 168, K = 256$. Lower is better.

| Dataset | ch=8 | ch=16 | ch=24 | ch=32 | ch=64 |
|---|---|---|---|---|---|
| NOAA-US | 0.4697 | 0.4406 | 0.4400 | 0.4554 | 0.4460 |
| BMS Air | 0.5301 | 0.5180 | 0.5160 | 0.5396 | 0.5761 |

the cost dominated by the fixed DDIM loop rather than a sequential horizon rollout. Tab. 5 shows nearly flat latency across horizons, while NeuralCDE, MRDiff, TimeGrad, and ContiFormer are $1.24\times$, $1.97\times$, $39.8\times$, and $150.6\times$ slower.

### 6.7. Sensitivity Analysis

Tabs. 7–8 test whether modal capacity or latent width creates a bottleneck. Increasing $K$ from 16 to 512 changes average runtime only from 454.14 to 455.00 ms and from 452.59 to 455.66 ms, since $K$ affects only batched modal synthesis, not the number of denoising steps. For latent channels, 24 channels perform best on both NOAA-US and BMS Air, with 16 channels close; larger widths bring no benefit and can degrade performance.

### 6.8. Extrapolation with Controlled Regime Shifts

Fig. 5 isolates boundary-crossing robustness from stricter unseen-regime extrapolation. For boundary-crossing, severe

shifts cause only mild degradation: frequency $2.0\times$ changes CRPS/MAE from $0.3065/0.3879$ to $0.3376/0.4217$, and decay $2.5\times$ from $0.2482/0.3166$ to $0.2890/0.3621$. Then, we adopt a stricter split where train/validation windows stay pre-shift, and test includes boundary-crossing and post-shift-context windows; the ablation disables only history-conditioned pole perturbations. Cond poles outperform the poles without, except for a near tie in crossing CRPS under decay shift. The effect is modest but consistent, supporting that LLapDiff remains stable under controlled spectral shifts and that realized poles adapt at inference through bounded history-conditioned perturbations.

## 7. Conclusion

In this paper, we present Latent Laplace Diffusion, a conditional generative framework for irregular multivariate time series that integrates latent diffusion with a stable continuous-time inductive bias. LLapDiff avoids sequential numerical integration over physical time by modeling local mean evolution in the Laplace domain and synthesizing latent trajectories through a stable modal parameterization motivated by stochastic port-Hamiltonian dynamics. This enables horizon-wide generation without physical-time stepping, while the same queried-trajectory nature supports missing-value imputation. Our renewal-averaging analysis links sampling gaps to the effective event-domain poles, motivating gap-aware history conditioning that adapts continuous-time modal parameters to irregular observation patterns. Across seven real-world datasets, LLapDiff consistently improves over strong baselines, with the clearest gains at longer horizons and under stronger irregularity. Ablations, pole visualizations, runtime analyses, and controlled stress tests further support the roles of gap-aware conditioning, stable learned poles, and parallel modal synthesis. Limitations include reliance on pretrained latent representations and a locally linear modal approximation; future work will explore richer nonlinear dynamics and faster reverse sampling.

## Acknowledgments

This work was supported by the UK Research and Innovation (UKRI) Engineering and Physical Sciences Research Council (EPSRC), grant number EP/Y028392/1: AI for Collective Intelligence (AI4CI).

## Impact Statement

This paper presents Latent Laplace Diffusion, a conditional generative framework for forecasting and imputing irregular multivariate time series, with potential applications in healthcare, climate science, and finance. Like other generative approaches, it can produce plausible but incorrect values in unobserved regions. In safety-critical scenarios, we recommend using its uncertainty estimates and validating outputs against domain constraints rather than relying on point estimations.

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

## A. Port-Hamiltonian SDE Derivations

This appendix derives the expected energy balance in Eq. (3) from the stochastic port-Hamiltonian prior. We present the derivation in Itô form, then give the equivalent Stratonovich form and the Itô–Stratonovich conversion for completeness.

### A.1. Notation and Regularity Assumptions

Let $\boldsymbol{x}_t \in \mathbb{R}^{d_z}$ be the state, $\boldsymbol{u}_t \in \mathbb{R}^{d_u}$ an exogenous input, and $\boldsymbol{W}_t \in \mathbb{R}^{d_w}$ a standard Wiener process on a filtered probability space $(\Omega, \mathcal{F}, \{\mathcal{F}_t\}_{t \geq 0}, \mathbb{P})$. Let the Hamiltonian $H(\boldsymbol{x}; \psi)$ be $C^2$ in $\boldsymbol{x}$ and $C^1$ in $\psi$. The context process $\psi_t$ is assumed $\mathcal{F}_t$-adapted and of finite variation between observation updates (i.e., piecewise constant in our method; see Appendix A.6). We treat $\psi_t$ as càdlàg (i.e., right-continuous with left limits); if it has jumps, their contribution to the energy balance is made explicit in Appendix A.6.

The structural matrices satisfy $\boldsymbol{J}^\top = -\boldsymbol{J}$ (skew-symmetric interconnection) and $\boldsymbol{R} \succ 0$ (dissipation). Define the port-collocated output

$$\tilde{\boldsymbol{y}}_t := \boldsymbol{G}(\psi_t)^\top \nabla_{\boldsymbol{x}} H(\boldsymbol{x}_t; \psi_t), \tag{18}$$

which matches Eq. (2) in the main text. For the diffusion modeling setting, the diffusion trajectory typically corresponds to a generic linear readout $\boldsymbol{y}_t = \boldsymbol{C}\boldsymbol{x}_t$; so $\tilde{\boldsymbol{y}}_t$ is used only for the energy-balance derivation as a special case (see Appendix B.4).

We assume standard conditions ensuring the existence and uniqueness of a strong solution to the SDE below, and the integrability condition

$$\mathbb{E}\left[ \int_0^{T_{\text{fin}}} \left\| \boldsymbol{\Sigma}_t^\top \nabla_{\boldsymbol{x}} H(\boldsymbol{x}_t; \psi_t) \right\|_2^2 dt \right] < \infty, \tag{19}$$

for some finite horizon $T_{\text{fin}} > 0$, so the associated stochastic integral is a martingale.

### A.2. Dynamics in Itô Form

We start from the Itô SDE and allow explicit time dependence in the diffusion:

$$d\boldsymbol{x}_t = \left( (\boldsymbol{J} - \boldsymbol{R}) \nabla_{\boldsymbol{x}} H(\boldsymbol{x}_t; \psi_t) + \boldsymbol{G}(\psi_t)\boldsymbol{u}_t \right) dt + \boldsymbol{\Sigma}_t \, d\boldsymbol{W}_t. \tag{20}$$

Define the co-energy variable and drift,

$$\boldsymbol{v}_t := \nabla_{\boldsymbol{x}} H(\boldsymbol{x}_t; \psi_t), \qquad \boldsymbol{f}_t := (\boldsymbol{J} - \boldsymbol{R})\boldsymbol{v}_t + \boldsymbol{G}(\psi_t)\boldsymbol{u}_t, \tag{21}$$

so that $d\boldsymbol{x}_t = \boldsymbol{f}_t \, dt + \boldsymbol{\Sigma}_t \, d\boldsymbol{W}_t$.

### A.3. Itô Differential of $H(\boldsymbol{x}_t; \psi_t)$

We apply the semimartingale chain rule to the pair $(\boldsymbol{x}_t, \psi_t)$: on intervals where $\psi_t$ is continuous finite variation, the dependence on $\psi_t$ contributes through a pathwise (Lebesgue–Stieltjes) differential $d\psi_t$; if $\psi_t$ has jumps, the corresponding energy jump is handled explicitly in Appendix A.6.

Applying multivariate Itô's formula to $t \mapsto H(\boldsymbol{x}_t; \psi_t)$ gives

$$dH(\boldsymbol{x}_t; \psi_t) = \boldsymbol{v}_t^\top d\boldsymbol{x}_t + \left[ \nabla_\psi H(\boldsymbol{x}_t; \psi_t) \right]^\top d\psi_t + \frac{1}{2}\text{tr}\left( \boldsymbol{\Sigma}_t \boldsymbol{\Sigma}_t^\top \nabla_{\boldsymbol{x}}^2 H(\boldsymbol{x}_t; \psi_t) \right) dt. \tag{22}$$

Substituting $d\boldsymbol{x}_t = \boldsymbol{f}_t \, dt + \boldsymbol{\Sigma}_t \, d\boldsymbol{W}_t$ yields

$$dH(\boldsymbol{x}_t; \psi_t) = \boldsymbol{v}_t^\top \boldsymbol{f}_t \, dt + \boldsymbol{v}_t^\top \boldsymbol{\Sigma}_t \, d\boldsymbol{W}_t + (\nabla_\psi H(\boldsymbol{x}_t; \psi_t))^\top d\psi_t + \frac{1}{2}\text{tr}\left( \boldsymbol{\Sigma}_t \boldsymbol{\Sigma}_t^\top \nabla_{\boldsymbol{x}}^2 H(\boldsymbol{x}_t; \psi_t) \right) dt$$

$$= \boldsymbol{v}_t^\top \left( (\boldsymbol{J} - \boldsymbol{R})\boldsymbol{v}_t + \boldsymbol{G}(\psi_t)\boldsymbol{u}_t \right) dt + \boldsymbol{v}_t^\top \boldsymbol{\Sigma}_t \, d\boldsymbol{W}_t + (\nabla_\psi H)^\top d\psi_t + \frac{1}{2}\text{tr}\left( \boldsymbol{\Sigma}_t \boldsymbol{\Sigma}_t^\top \nabla_{\boldsymbol{x}}^2 H \right) dt. \tag{23}$$

### A.4. Energy decomposition

**(i) Interconnection term.** Since $\boldsymbol{J}^\top = -\boldsymbol{J}$,

$$\boldsymbol{v}_t^\top \boldsymbol{J} \boldsymbol{v}_t = (\boldsymbol{v}_t^\top \boldsymbol{J} \boldsymbol{v}_t)^\top = \boldsymbol{v}_t^\top \boldsymbol{J}^\top \boldsymbol{v}_t = -\boldsymbol{v}_t^\top \boldsymbol{J} \boldsymbol{v}_t \; \Rightarrow \; \boldsymbol{v}_t^\top \boldsymbol{J} \boldsymbol{v}_t = 0. \tag{24}$$

**(ii) Dissipation term.** Since $\boldsymbol{R} \succ 0$,

$$\boldsymbol{v}_t^\top(-\boldsymbol{R})\boldsymbol{v}_t = -\boldsymbol{v}_t^\top \boldsymbol{R}\boldsymbol{v}_t \leq 0. \tag{25}$$

**(iii) Input power.** By $\tilde{\boldsymbol{y}}_t = \boldsymbol{G}(\psi_t)^\top \boldsymbol{v}_t$,

$$\boldsymbol{v}_t^\top \boldsymbol{G}(\psi_t)\boldsymbol{u}_t = (\boldsymbol{G}(\psi_t)^\top \boldsymbol{v}_t)^\top \boldsymbol{u}_t = \tilde{\boldsymbol{y}}_t^\top \boldsymbol{u}_t. \tag{26}$$

Substituting Eqs. (24)–(26) into Eq. (23) gives

$$dH(\boldsymbol{x}_t; \psi_t) = (\nabla_\psi H(\boldsymbol{x}_t; \psi_t))^\top d\psi_t - \boldsymbol{v}_t^\top \boldsymbol{R}\boldsymbol{v}_t\, dt + \tilde{\boldsymbol{y}}_t^\top \boldsymbol{u}_t\, dt \\ + \boldsymbol{v}_t^\top \boldsymbol{\Sigma}_t\, d\boldsymbol{W}_t + \frac{1}{2}\text{tr}\Big(\boldsymbol{\Sigma}_t \boldsymbol{\Sigma}_t^\top \nabla_{\boldsymbol{x}}^2 H(\boldsymbol{x}_t; \psi_t)\Big)\, dt. \tag{27}$$

The terms correspond to context variation, dissipation, supplied power, stochastic injection (martingale), and the Itô correction, respectively.

## A.5. Expected Energy Balance

Under Eq. (19), the stochastic integral is a martingale and has zero mean:

$$\mathbb{E}\left[\int_{t_a}^{t_b} \boldsymbol{v}_t^\top \boldsymbol{\Sigma}_t\, d\boldsymbol{W}_t\right] = 0. \tag{28}$$

Integrating Eq. (27) over $[t_a, t_b]$ and taking expectations yields

$$\mathbb{E}[H(\boldsymbol{x}_{t_b}; \psi_{t_b})] - \mathbb{E}[H(\boldsymbol{x}_{t_a}; \psi_{t_a})] = \mathbb{E}\left[\int_{t_a}^{t_b} (\nabla_\psi H(\boldsymbol{x}_t; \psi_t))^\top d\psi_t\right] - \mathbb{E}\left[\int_{t_a}^{t_b} \boldsymbol{v}_t^\top \boldsymbol{R}\boldsymbol{v}_t\, dt\right] \\ + \mathbb{E}\left[\int_{t_a}^{t_b} \tilde{\boldsymbol{y}}_t^\top \boldsymbol{u}_t\, dt\right] + \frac{1}{2}\mathbb{E}\left[\int_{t_a}^{t_b} \text{tr}\Big(\boldsymbol{\Sigma}_t \boldsymbol{\Sigma}_t^\top \nabla_{\boldsymbol{x}}^2 H(\boldsymbol{x}_t; \psi_t)\Big)\, dt\right]. \tag{29}$$

In our method, $\psi_t$ is piecewise constant with jumps (Appendix A.6); the differentiable-$\psi_t$ case below is included only to connect with the standard continuous-time statement. If $\psi_t$ is differentiable on an interval (so $d\psi_t = \dot{\psi}_t\, dt$), dividing by $dt$ gives the instantaneous form

$$\frac{d}{dt}\mathbb{E}[H(\boldsymbol{x}_t; \psi_t)] = \mathbb{E}\big[(\nabla_\psi H(\boldsymbol{x}_t; \psi_t))^\top \dot{\psi}_t\big] - \mathbb{E}[\boldsymbol{v}_t^\top \boldsymbol{R}\boldsymbol{v}_t] + \mathbb{E}[\tilde{\boldsymbol{y}}_t^\top \boldsymbol{u}_t] + \frac{1}{2}\mathbb{E}\Big[\text{tr}\Big(\boldsymbol{\Sigma}_t \boldsymbol{\Sigma}_t^\top \nabla_{\boldsymbol{x}}^2 H(\boldsymbol{x}_t; \psi_t)\Big)\Big]. \tag{30}$$

This matches Eq. (3) in the main text.

**Remark (passivity).** If $\boldsymbol{\Sigma}_t \equiv 0$, $\boldsymbol{u}_t \equiv 0$, and $\psi_t$ is constant, then $\frac{d}{dt}H(\boldsymbol{x}_t; \psi) = -\boldsymbol{v}_t^\top \boldsymbol{R}\boldsymbol{v}_t \leq 0$, i.e., energy is non-increasing.

## A.6. Piecewise-constant Context $\psi_t$ (observation-driven updates)

In our setting, $\psi_t$ is piecewise constant between observation times $\{t_j\}$: $\psi_t = \psi_{t_j}$ for $t \in [t_j, t_{j+1})$ (right-continuous). Hence $d\psi_t = 0$ on open intervals $(t_j, t_{j+1})$, and the context term in Eq. (27) vanishes between updates.

At an update time $t = t_j$, the context may jump from $\psi_{t_j^-}$ to $\psi_{t_j^+}$, producing an instantaneous energy jump

$$\Delta H\big|_{t=t_j} := H(\boldsymbol{x}_{t_j}; \psi_{t_j^+}) - H(\boldsymbol{x}_{t_j}; \psi_{t_j^-}). \tag{31}$$

Here, $\boldsymbol{x}_t$ is continuous almost surely, so $\boldsymbol{x}_{t_j^-} = \boldsymbol{x}_{t_j^+} = \boldsymbol{x}_{t_j}$. Integrating Eq. (27) over $[t_a, t_b]$ and accounting for jumps yields

$$H(\boldsymbol{x}_{t_b}; \psi_{t_b}) = H(\boldsymbol{x}_{t_a}; \psi_{t_a}) + \int_{t_a}^{t_b} \left(-\boldsymbol{v}_t^\top \boldsymbol{R}\boldsymbol{v}_t + \tilde{\boldsymbol{y}}_t^\top \boldsymbol{u}_t + \frac{1}{2}\text{tr}\Big(\boldsymbol{\Sigma}_t \boldsymbol{\Sigma}_t^\top \nabla_{\boldsymbol{x}}^2 H(\boldsymbol{x}_t; \psi_t)\Big)\right) dt \\ + \int_{t_a}^{t_b} \boldsymbol{v}_t^\top \boldsymbol{\Sigma}_t\, d\boldsymbol{W}_t + \sum_{j:\, t_j \in (t_a, t_b]} \Delta H\big|_{t=t_j}. \tag{32}$$

Thus, between observation updates dissipation/input/noise govern energy flow, while context affects energy through discrete jumps.

### A.7. Stratonovich Form and Itô–Stratonovich Conversions

For completeness, we present the general state-dependent Itô–Stratonovich conversion; the main derivation above uses additive noise $\boldsymbol{\Sigma}_t$.

#### A.7.1. STRATONOVICH DYNAMICS

An equivalent Stratonovich representation is

$$d\boldsymbol{x}_t = \boldsymbol{f}^S(\boldsymbol{x}_t, \psi_t, \boldsymbol{u}_t)\, dt + \boldsymbol{\Sigma}(\boldsymbol{x}_t, t) \circ d\boldsymbol{W}_t, \tag{33}$$

where $\circ$ denotes the Stratonovich integral. When $\boldsymbol{\Sigma}$ is additive, $\boldsymbol{f}^S$ coincides with the Itô drift $\boldsymbol{f}_t$ in Eq. (21). In Stratonovich calculus, the chain rule takes the classical form

$$dH(\boldsymbol{x}_t; \psi_t) = (\nabla_{\boldsymbol{x}} H(\boldsymbol{x}_t; \psi_t))^\top d\boldsymbol{x}_t + (\nabla_\psi H(\boldsymbol{x}_t; \psi_t))^\top d\psi_t. \tag{34}$$

Substituting Eq. (33) into Eq. (34) gives the Stratonovich energy balance

$$dH = (\nabla_\psi H)^\top d\psi_t + (\nabla_{\boldsymbol{x}} H)^\top \boldsymbol{f}^S\, dt + (\nabla_{\boldsymbol{x}} H)^\top \boldsymbol{\Sigma} \circ d\boldsymbol{W}_t. \tag{35}$$

#### A.7.2. CONVERTING STRATONOVICH $\to$ ITÔ (STATE-DEPENDENT DIFFUSION)

For general $\boldsymbol{\Sigma}(\boldsymbol{x}, t)$, the equivalent Itô SDE is

$$d\boldsymbol{x}_t = \left( \boldsymbol{f}^S(\boldsymbol{x}_t, \psi_t, \boldsymbol{u}_t) + \frac{1}{2} \sum_{\alpha=1}^{d_w} (\boldsymbol{D}_{\boldsymbol{x}} \boldsymbol{\Sigma}_{\cdot\alpha})(\boldsymbol{x}_t, t)\, \boldsymbol{\Sigma}_{\cdot\alpha}(\boldsymbol{x}_t, t) \right) dt + \boldsymbol{\Sigma}(\boldsymbol{x}_t, t)\, d\boldsymbol{W}_t, \tag{36}$$

where $\boldsymbol{\Sigma}_{\cdot\alpha}$ is the $\alpha$-th column of $\boldsymbol{\Sigma}$ and $\boldsymbol{D}_{\boldsymbol{x}} \boldsymbol{\Sigma}_{\cdot\alpha}$ is its Jacobian w.r.t. $\boldsymbol{x}$. If the noise is additive (i.e., $\boldsymbol{\Sigma}$ does not depend on $\boldsymbol{x}$), then $\boldsymbol{D}_{\boldsymbol{x}} \boldsymbol{\Sigma}_{\cdot\alpha} = 0$ and the drift correction vanishes, so $\boldsymbol{f}^I = \boldsymbol{f}^S$.

#### A.7.3. CONVERTING THE STRATONOVICH ENERGY BALANCE TO ITÔ

The Stratonovich integral satisfies

$$\int (\nabla_{\boldsymbol{x}} H)^\top \boldsymbol{\Sigma} \circ d\boldsymbol{W}_t = \int (\nabla_{\boldsymbol{x}} H)^\top \boldsymbol{\Sigma}\, d\boldsymbol{W}_t + \frac{1}{2} \sum_{\alpha=1}^{d_w} d \left\langle (\nabla_{\boldsymbol{x}} H)^\top \boldsymbol{\Sigma}_{\cdot\alpha}, W^\alpha \right\rangle_t. \tag{37}$$

Expanding the quadratic covariation term yields

$$\frac{1}{2} \sum_{\alpha=1}^{d_w} d \left\langle (\nabla_{\boldsymbol{x}} H)^\top \boldsymbol{\Sigma}_{\cdot\alpha}, W^\alpha \right\rangle_t = \frac{1}{2} \mathrm{tr}\!\left( \boldsymbol{\Sigma}\boldsymbol{\Sigma}^\top \nabla_{\boldsymbol{x}}^2 H \right) dt + \frac{1}{2} (\nabla_{\boldsymbol{x}} H)^\top \left( \sum_{\alpha=1}^{d_w} (\boldsymbol{D}_{\boldsymbol{x}} \boldsymbol{\Sigma}_{\cdot\alpha})\, \boldsymbol{\Sigma}_{\cdot\alpha} \right) dt. \tag{38}$$

Thus, starting from Eq. (35) and applying Eqs. (37)–(38) recovers the Itô correction $\frac{1}{2} \mathrm{tr}(\boldsymbol{\Sigma}\boldsymbol{\Sigma}^\top \nabla_{\boldsymbol{x}}^2 H)\, dt$ plus the same drift adjustment as in Eq. (36). This is why the Itô form is the most direct statement of the expected energy balance used in the main text.

## B. Laplace-Domain Mean Dynamics and Stable Modal Parameterization

This appendix provides derivations for Sec. 4.2 and Sec. 4.3. We (i) define a locally frozen operating point and motivate an equilibrium/stationarity approximation, (ii) linearize the drift and derive mean dynamics via the mild solution, (iii) derive the unilateral Laplace-domain characterization, and (iv) present the stable modal parameterization, a real state-space realization, and stability. We inherit the notation and standing assumptions from Appendix A.

## B.1. Local Operating Point and Equilibrium Justification

We start from the Itô port-Hamiltonian SDE as defined in Appendix A:

$$d\boldsymbol{x}_t = f(\boldsymbol{x}_t; \psi_t, \boldsymbol{u}_t)\, dt + \boldsymbol{\Sigma}_t\, d\boldsymbol{W}_t, \qquad f(\boldsymbol{x}; \psi, \boldsymbol{u}) := (\boldsymbol{J} - \boldsymbol{R})\nabla_{\boldsymbol{x}} H(\boldsymbol{x}; \psi) + \boldsymbol{G}(\psi)\boldsymbol{u}. \tag{39}$$

**Local freezing.** Fix a reference time $t_0$ and consider a short window $t \in [t_0, t_0 + \Delta_{\mathrm{loc}}]$ over which coefficients are approximately constant:

$$\psi_t \approx \psi_{t_0}, \qquad \boldsymbol{\Sigma}_t \approx \boldsymbol{\Sigma}_{t_0}. \tag{40}$$

We use the constant $\boldsymbol{\Sigma}_{t_0}$ only to write the closed-form mild solution; the drift linearization does not require additive noise.

**Operating point / approximate equilibrium.** Let $(\bar{\boldsymbol{x}}_{t_0}, \bar{\boldsymbol{u}}_{t_0})$ be a nominal operating point under context $\psi_{t_0}$ with small residual drift:

$$f(\bar{\boldsymbol{x}}_{t_0}; \psi_{t_0}, \bar{\boldsymbol{u}}_{t_0}) \approx \boldsymbol{0}. \tag{41}$$

A nonzero residual contributes an affine term in the local linear model; it affects the mean but not the local poles (eigenvalues of the Jacobian).

**Energy dissipation (unforced deterministic prior).** Under frozen context $\psi = \psi_{t_0}$, the unforced deterministic prior $(\boldsymbol{u} \equiv \boldsymbol{0}, \boldsymbol{\Sigma} \equiv \boldsymbol{0})$ is

$$\dot{\boldsymbol{x}} = (\boldsymbol{J} - \boldsymbol{R})\nabla_{\boldsymbol{x}} H(\boldsymbol{x}; \psi_{t_0}). \tag{42}$$

Using $\boldsymbol{J}^\top = -\boldsymbol{J}$,

$$\frac{d}{dt} H(\boldsymbol{x}_t; \psi_{t_0}) = \nabla_{\boldsymbol{x}} H(\boldsymbol{x}_t; \psi_{t_0})^\top (\boldsymbol{J} - \boldsymbol{R})\nabla_{\boldsymbol{x}} H(\boldsymbol{x}_t; \psi_{t_0}) = -\nabla_{\boldsymbol{x}} H(\boldsymbol{x}_t; \psi_{t_0})^\top \boldsymbol{R} \nabla_{\boldsymbol{x}} H(\boldsymbol{x}_t; \psi_{t_0}) \leq 0, \tag{43}$$

i.e., $\boldsymbol{J}$ is power-preserving and $\boldsymbol{R}$ dissipates energy.

**Lemma B.1** (Unforced equilibrium implies $\nabla_{\boldsymbol{x}} H = 0$ under strict dissipation). *With $\boldsymbol{R} \succ \boldsymbol{0}$ and consider the unforced dynamics Eq. (42) under fixed $\psi$. If $\bar{\boldsymbol{x}}$ satisfies $(\boldsymbol{J} - \boldsymbol{R})\nabla_{\boldsymbol{x}} H(\bar{\boldsymbol{x}}; \psi) = \boldsymbol{0}$, then $\nabla_{\boldsymbol{x}} H(\bar{\boldsymbol{x}}; \psi) = \boldsymbol{0}$.*

Proof. Recall $\boldsymbol{v} := \nabla_{\boldsymbol{x}} H(\bar{\boldsymbol{x}}; \psi)$. Left-multiplying by $\boldsymbol{v}^\top$ gives $0 = \boldsymbol{v}^\top (\boldsymbol{J} - \boldsymbol{R})\boldsymbol{v} = \boldsymbol{v}^\top \boldsymbol{J} \boldsymbol{v} - \boldsymbol{v}^\top \boldsymbol{R} \boldsymbol{v}$. Since $\boldsymbol{v}^\top \boldsymbol{J} \boldsymbol{v} = 0$ for skew-symmetric $\boldsymbol{J}$, we have $\boldsymbol{v}^\top \boldsymbol{R} \boldsymbol{v} = 0$. Because $\boldsymbol{R} \succ \boldsymbol{0}$, this implies $\boldsymbol{v} = \boldsymbol{0}$.

**Corollary B.2** (Strict local minimum $\Rightarrow$ local asymptotic stability). *With $\boldsymbol{R} \succ \boldsymbol{0}$ and $\nabla_{\boldsymbol{x}} H(\bar{\boldsymbol{x}}; \psi) = \boldsymbol{0}$. If*

$$\nabla_{\boldsymbol{x}}^2 H(\bar{\boldsymbol{x}}; \psi) \succ \boldsymbol{0}, \tag{44}$$

*then $\bar{\boldsymbol{x}}$ is a strict local minimum of $H(\cdot; \psi)$ and is locally asymptotically stable for Eq. (42).*

Justification. Let $V(\boldsymbol{x}) := H(\boldsymbol{x}; \psi) - H(\bar{\boldsymbol{x}}; \psi)$. Condition Eq. (44) makes $V$ locally positive definite. By Eq. (43), $\dot{V}(\boldsymbol{x}) = -\nabla_{\boldsymbol{x}} H(\boldsymbol{x}; \psi)^\top \boldsymbol{R} \nabla_{\boldsymbol{x}} H(\boldsymbol{x}; \psi) < 0$ whenever $\nabla_{\boldsymbol{x}} H(\boldsymbol{x}; \psi) \neq \boldsymbol{0}$.

**Linearized stability certificate.** Let $\kappa_{t_0} := \nabla_{\boldsymbol{x}}^2 H(\bar{\boldsymbol{x}}_{t_0}; \psi_{t_0})$ (symmetric) and define $\boldsymbol{A} := (\boldsymbol{J} - \boldsymbol{R})\kappa_{t_0}$. If $\boldsymbol{R} \succ \boldsymbol{0}$ and $\kappa_{t_0} \succ \boldsymbol{0}$, then

$$\boldsymbol{A}^\top \kappa_{t_0} + \kappa_{t_0} \boldsymbol{A} = -\kappa_{t_0}(\boldsymbol{R} + \boldsymbol{R}^\top)\kappa_{t_0} \prec \boldsymbol{0}, \tag{45}$$

so $\boldsymbol{A}$ is Hurwitz. This motivates enforcing stable poles in Sec. 4.3.

## B.2. First-order Drift Linearization

**Deviations.** Define deviations around $(\bar{\boldsymbol{x}}_{t_0}, \bar{\boldsymbol{u}}_{t_0})$:

$$\delta\boldsymbol{x}_t := \boldsymbol{x}_t - \bar{\boldsymbol{x}}_{t_0}, \qquad \delta\boldsymbol{u}_t := \boldsymbol{u}_t - \bar{\boldsymbol{u}}_{t_0}. \tag{46}$$

Under local freezing Eq. (40), substituting $\boldsymbol{x}_t = \bar{\boldsymbol{x}}_{t_0} + \delta\boldsymbol{x}_t$ and $\boldsymbol{u}_t = \bar{\boldsymbol{u}}_{t_0} + \delta\boldsymbol{u}_t$ into Eq. (39) gives

$$d(\delta\boldsymbol{x}_t) = f(\bar{\boldsymbol{x}}_{t_0} + \delta\boldsymbol{x}_t; \psi_{t_0}, \bar{\boldsymbol{u}}_{t_0} + \delta\boldsymbol{u}_t)\, dt + \boldsymbol{\Sigma}_{t_0}\, d\boldsymbol{W}_t. \tag{47}$$

**First-order expansion.** Using a first-order Taylor expansion of $\nabla_{\boldsymbol{x}} H$ around $\bar{\boldsymbol{x}}_{t_0}$ (with frozen $\psi_{t_0}$),

$$\nabla_{\boldsymbol{x}} H(\bar{\boldsymbol{x}}_{t_0} + \delta\boldsymbol{x}; \psi_{t_0}) = \nabla_{\boldsymbol{x}} H(\bar{\boldsymbol{x}}_{t_0}; \psi_{t_0}) + \kappa_{t_0}\, \delta\boldsymbol{x} + o(\|\delta\boldsymbol{x}\|), \tag{48}$$

and noting $\boldsymbol{G}(\psi_{t_0})$ is constant under freezing, we obtain

$$f(\bar{\boldsymbol{x}}_{t_0} + \delta\boldsymbol{x}; \psi_{t_0}, \bar{\boldsymbol{u}}_{t_0} + \delta\boldsymbol{u}) = f(\bar{\boldsymbol{x}}_{t_0}; \psi_{t_0}, \bar{\boldsymbol{u}}_{t_0}) + (\boldsymbol{J} - \boldsymbol{R})\kappa_{t_0}\delta\boldsymbol{x} + \boldsymbol{G}(\psi_{t_0})\delta\boldsymbol{u} + o(\|\delta\boldsymbol{x}\|). \tag{49}$$

Using Eq. (41), we absorb the residual constant term into an affine forcing (or drop it when focusing on poles) and ignore higher-order terms, yielding the local linear SDE (main text Eq. (5)):

$$d(\delta\boldsymbol{x}_t) = \left(\boldsymbol{A}\delta\boldsymbol{x}_t + \boldsymbol{B}\delta\boldsymbol{u}_t\right)dt + \boldsymbol{\Sigma}_{t_0}\,d\boldsymbol{W}_t, \tag{50}$$

with

$$\boldsymbol{A} := (\boldsymbol{J} - \boldsymbol{R})\kappa_{t_0} = (\boldsymbol{J} - \boldsymbol{R})\nabla_{\boldsymbol{x}}^2 H(\bar{\boldsymbol{x}}_{t_0}; \psi_{t_0}), \qquad \boldsymbol{B} := \boldsymbol{G}(\psi_{t_0}). \tag{51}$$

**Remark (affine residual).** If $f(\bar{\boldsymbol{x}}_{t_0}; \psi_{t_0}, \bar{\boldsymbol{u}}_{t_0}) \neq \boldsymbol{0}$, then Eq. (50) includes an additional constant drift $\boldsymbol{f}_0\,dt$. This shifts the mean but does not change the eigenvalues of the local Jacobian $\boldsymbol{A}$ (hence does not change the modal poles).

### B.3. Mild Solution and Mean Dynamics

**Mild solution.** For $t \geq t_0$, the mild solution of Eq. (50) is

$$\delta\boldsymbol{x}_t = \mathrm{e}^{\boldsymbol{A}(t-t_0)}\delta\boldsymbol{x}_{t_0} + \int_{t_0}^t \mathrm{e}^{\boldsymbol{A}(t-r)}\boldsymbol{B}\,\delta\boldsymbol{u}_r\,dr + \int_{t_0}^t \mathrm{e}^{\boldsymbol{A}(t-r)}\boldsymbol{\Sigma}_{t_0}\,d\boldsymbol{W}_r. \tag{52}$$

**Mean dynamics.** Taking expectations and using the martingale property of the stochastic integral (Appendix A) yields

$$\mathbb{E}[\delta\boldsymbol{x}_t] = \mathrm{e}^{\boldsymbol{A}(t-t_0)}\mathbb{E}[\delta\boldsymbol{x}_{t_0}] + \int_{t_0}^t \mathrm{e}^{\boldsymbol{A}(t-r)}\boldsymbol{B}\,\mathbb{E}[\delta\boldsymbol{u}_r]\,dr. \tag{53}$$

In our conditioning setup, $\delta\boldsymbol{u}_r$ is typically treated as deterministic given the history summary token sequence $\boldsymbol{E}_{t_i}$ (fixed at the reference time), so $\mathbb{E}[\delta\boldsymbol{u}_r] = \delta\boldsymbol{u}_r$; otherwise, one may interpret $\mathbb{E}[\delta\boldsymbol{u}_r]$ as a conditional expectation.

### B.4. Linearized Readout Map and Green's Function

We use a generic linear readout $\boldsymbol{y}_t := \boldsymbol{C}\boldsymbol{x}_t$ to map the Hamiltonian state to the latent prediction space. Under frozen context and linearization around $\bar{\boldsymbol{x}}_{t_0}$, this yields $\delta\boldsymbol{y}_t \approx \boldsymbol{C}\,\delta\boldsymbol{x}_t$ (main text Eq. (8)).

The port-collocated output from Appendix A is

$$\tilde{\boldsymbol{y}}_t = \boldsymbol{G}(\psi_t)^\top \nabla_{\boldsymbol{x}} H(\boldsymbol{x}_t; \psi_t). \tag{54}$$

Under frozen context and linearization, the corresponding deviation satisfies $\delta\tilde{\boldsymbol{y}}_t \approx \boldsymbol{C}\,\delta\boldsymbol{x}_t$ with $\boldsymbol{C} := \boldsymbol{G}(\psi_{t_0})^\top \kappa_{t_0}$. Combining $\delta\boldsymbol{y}_t \approx \boldsymbol{C}\,\delta\boldsymbol{x}_t$ with Eq. (53) yields the mean output decomposition

$$\mathbb{E}[\delta\boldsymbol{y}_t] = \boldsymbol{C}\mathrm{e}^{\boldsymbol{A}(t-t_0)}\mathbb{E}[\delta\boldsymbol{x}_{t_0}] + \int_{t_0}^t \boldsymbol{g}(t-r)\,\mathbb{E}[\delta\boldsymbol{u}_r]\,dr, \tag{55}$$

where the unilateral Green's function is

$$\boldsymbol{g}(t) := \boldsymbol{C}\mathrm{e}^{\boldsymbol{A}t}\boldsymbol{B}, \qquad t \geq 0. \tag{56}$$

### B.5. Unilateral Laplace Transform and Resolvent

Shift time so that $t_0 \mapsto 0$ and define the unilateral Laplace transform $\mathrm{Lap}\{h\}(s) := \int_0^\infty \mathrm{e}^{-st}h(t)\,dt$, for $\Re(s)$ sufficiently large. Let

$$\boldsymbol{m}_x(t) := \mathbb{E}[\delta\boldsymbol{x}_{t_0+t}], \qquad \boldsymbol{m}_u(t) := \mathbb{E}[\delta\boldsymbol{u}_{t_0+t}], \qquad \boldsymbol{m}_x(0) = \mathbb{E}[\delta\boldsymbol{x}_{t_0}].$$

Taking expectations in Eq. (50) yields the deterministic LTI mean equation

$$\dot{\boldsymbol{m}}_x(t) = \boldsymbol{A}\boldsymbol{m}_x(t) + \boldsymbol{B}\boldsymbol{m}_u(t). \tag{57}$$

Applying Lap and using $\text{Lap}\{\dot{\boldsymbol{m}}_x\}(s) = s\boldsymbol{\Omega}_x(s) - \boldsymbol{\Omega}_x(0)$ gives

$$s\boldsymbol{\Omega}_x(s) - \boldsymbol{m}_x(0) = \boldsymbol{A}\boldsymbol{\Omega}_x(s) + \boldsymbol{B}\boldsymbol{\Omega}_u(s),$$
$$\boldsymbol{\Omega}_x(s) = (s\boldsymbol{I} - \boldsymbol{A})^{-1}\boldsymbol{m}_x(0) + (s\boldsymbol{I} - \boldsymbol{A})^{-1}\boldsymbol{B}\,\boldsymbol{\Omega}_u(s). \tag{58}$$

With $\boldsymbol{\Omega}_y(s) = \boldsymbol{C}\boldsymbol{\Omega}_x(s)$, we obtain

$$\boldsymbol{\Omega}_y(s) = \boldsymbol{C}(s\boldsymbol{I} - \boldsymbol{A})^{-1}\boldsymbol{m}_x(0) + \underbrace{\boldsymbol{C}(s\boldsymbol{I} - \boldsymbol{A})^{-1}\boldsymbol{B}}_{\mathcal{G}(s)}\,\boldsymbol{\Omega}_u(s). \tag{59}$$

Hence, the Laplace-domain resolvent associated with the linearized mean dynamics is

$$\mathcal{G}(s) = \boldsymbol{C}(s\boldsymbol{I} - \boldsymbol{A})^{-1}\boldsymbol{B}, \tag{60}$$

which matches main text Eq. (10). The first term in Eq. (59) is the homogeneous (initial-state) contribution; in our setting, it is absorbed into conditioning rather than assuming $\mathbb{E}[\delta\boldsymbol{x}_{t_0}] = \boldsymbol{0}$.

### B.6. Stable Modal Parameterization

The resolvent $\mathcal{G}(s)$ is a proper rational matrix function whose poles coincide with the eigenvalues of $\boldsymbol{A}$. To avoid explicit dense exponentials on irregular grids, we parameterize $\mathcal{G}(s)$ directly using a sum of $K$ stable complex-conjugate modes (main text Eq. (11)).

#### B.6.1. COMPLEX-CONJUGATE POLES YIELD A REAL SECOND-ORDER FACTOR

Consider one conjugate pole pair at $s = -\rho_k \pm i\omega_k$ with $\rho_k > 0$ and $\omega_k > 0$. The corresponding real quadratic factor is

$$(s + \rho_k - i\omega_k)(s + \rho_k + i\omega_k) = (s + \rho_k)^2 + \omega_k^2 = s^2 + 2\rho_k s + (\rho_k^2 + \omega_k^2). \tag{61}$$

Using low-rank factors $\boldsymbol{b}_k \in \mathbb{R}^{d_u}$ and $\boldsymbol{c}_k \in \mathbb{R}^{d_z}$, define the $k$-th mode

$$\mathcal{G}_k(s) := \frac{\omega_k\,\boldsymbol{c}_k\boldsymbol{b}_k^\top}{s^2 + 2\rho_k s + (\rho_k^2 + \omega_k^2)}. \tag{62}$$

Summing over $k$ yields $\mathcal{G}(s) = \sum_{k=1}^{K}\mathcal{G}_k(s)$, i.e., Eq. (11). (In our latent diffusion model, we typically take $d_u = d_z$ and interpret $\boldsymbol{b}_k, \boldsymbol{c}_k$ as mode coefficients / low-rank factors predicted by the denoiser (used for synthesis).)

#### B.6.2. CANONICAL REAL STATE-SPACE REALIZATION

Define the $2 \times 2$ real block realization

$$\boldsymbol{A}_k = \begin{bmatrix} -\rho_k & -\omega_k \\ \omega_k & -\rho_k \end{bmatrix}, \qquad \boldsymbol{B}_k = \begin{bmatrix} 0 \\ 1 \end{bmatrix}\boldsymbol{b}_k^\top \in \mathbb{R}^{2 \times d_u}, \qquad \boldsymbol{C}_k = \boldsymbol{c}_k\begin{bmatrix} -1 & 0 \end{bmatrix} \in \mathbb{R}^{d_z \times 2}. \tag{63}$$

Then

$$(s\boldsymbol{I} - \boldsymbol{A}_k)^{-1} = \frac{1}{(s + \rho_k)^2 + \omega_k^2}\begin{bmatrix} s + \rho_k & -\omega_k \\ \omega_k & s + \rho_k \end{bmatrix}.$$

Multiplying,

$$\boldsymbol{C}_k(s\boldsymbol{I} - \boldsymbol{A}_k)^{-1}\boldsymbol{B}_k = \frac{\omega_k\,\boldsymbol{c}_k\boldsymbol{b}_k^\top}{(s + \rho_k)^2 + \omega_k^2} = \frac{\omega_k\,\boldsymbol{c}_k\boldsymbol{b}_k^\top}{s^2 + 2\rho_k s + (\rho_k^2 + \omega_k^2)} = \mathcal{G}_k(s), \tag{64}$$

recovering Eq. (62). Stacking blocks with

$$\boldsymbol{A} := \text{blkdiag}(\boldsymbol{A}_1, \ldots, \boldsymbol{A}_K), \quad \boldsymbol{B} := \begin{bmatrix} \boldsymbol{B}_1^\top & \cdots & \boldsymbol{B}_K^\top \end{bmatrix}^\top, \quad \boldsymbol{C} := \begin{bmatrix} \boldsymbol{C}_1 & \cdots & \boldsymbol{C}_K \end{bmatrix},$$

yields $\mathcal{G}(s) = \boldsymbol{C}(s\boldsymbol{I} - \boldsymbol{A})^{-1}\boldsymbol{B}$.

B.6.3. STABILITY UNDER $\rho_k > 0$ AND AN OPTIONAL TIME-DOMAIN BASIS

Each $\boldsymbol{A}_k$ has eigenvalues $-\rho_k \pm i\omega_k$. If $\rho_k > 0$, then both eigenvalues have negative real part, so each $\boldsymbol{A}_k$ is Hurwitz. Since $\boldsymbol{A}$ is Hurwitz under $\rho_k > 0$, the system is strictly stable. The general homogeneous solution (natural response) corresponding to these eigenvalues is spanned by the basis functions $\{\mathrm{e}^{-\rho_k t}\cos(\omega_k t), \mathrm{e}^{-\rho_k t}\sin(\omega_k t)\}$. In Appendix F.2, we utilize this basis to synthesize the latent trajectory $\hat{\boldsymbol{z}}_0$ from the history-conditioned modal residues.

## C. Renewal-Averaged View

This appendix provides detailed derivations for Sec. 4.4, mapping a continuous-time damped oscillatory mode $s_k = -\rho_k + i\omega_k$ under random sampling gaps $\Delta_j := t_j - t_{j-1} \geq 0$ to an effective event-domain log-multiplier $\bar{s}_k := \log \lambda_k$.

**Setup and standing assumptions.** As established in Appendix B, each oscillatory mode satisfies $\rho_k > 0$, hence $\Re(s_k) = -\rho_k < 0$. We consider event times $\{t_j\}_{j\geq 0}$ with gaps $\{\Delta_j\}_{j\geq 1}$. In the stationary renewal idealization used for analysis, $\Delta_j$ are i.i.d. and independent of the latent state (so in particular $\Delta_{j+1} \perp \zeta_j^{(k)}$ and $\Delta_{j+1} \perp \boldsymbol{\xi}_j^{(k)}$ below). Because $\Delta \geq 0$ and $\Re(s_k) < 0$,

$$\left|\mathrm{e}^{s_k \Delta}\right| = \mathrm{e}^{\Re(s_k)\Delta} = \mathrm{e}^{-\rho_k \Delta} \leq 1,$$

so $\lambda_k := \mathbb{E}[\mathrm{e}^{s_k \Delta}]$ exists (no additional moment assumptions are required for existence). For complex logarithms, we use the principal branch unless stated otherwise; see the discussion around Eq. (76) for branch-cut and phase-unwrapping edge cases.

### C.1. Complex scalar mode

Consider one eigen-coordinate $\zeta^{(k)}(t) \in \mathbb{C}$ evolving as

$$\zeta^{(k)}(t) = \mathrm{e}^{s_k(t-t_0)}\zeta^{(k)}(t_0), \qquad s_k = -\rho_k + i\omega_k. \tag{65}$$

Sampling at event times gives the exact event recursion

$$\zeta_{j+1}^{(k)} := \zeta^{(k)}(t_{j+1}) = \mathrm{e}^{s_k(t_{j+1}-t_j)}\zeta^{(k)}(t_j) = \mathrm{e}^{s_k \Delta_{j+1}}\zeta_j^{(k)}. \tag{66}$$

Taking expectations and using $\Delta_{j+1} \perp \zeta_j^{(k)}$,

$$\mathbb{E}[\zeta_{j+1}^{(k)}] = \mathbb{E}\left[\mathrm{e}^{s_k \Delta_{j+1}}\zeta_j^{(k)}\right] = \mathbb{E}\left[\mathrm{e}^{s_k \Delta}\right]\mathbb{E}[\zeta_j^{(k)}] := \lambda_k \,\mathbb{E}[\zeta_j^{(k)}]. \tag{67}$$

Iterating yields

$$\mathbb{E}[\zeta_j^{(k)}] = \lambda_k^j \,\mathbb{E}[\zeta_0^{(k)}]. \tag{68}$$

Define the effective event-domain log-pole

$$\bar{s}_k := \log(\lambda_k) = -\bar{\rho}_k + i\bar{\omega}_k, \tag{69}$$

so $\mathbb{E}[\zeta_j^{(k)}] = \mathrm{e}^{\bar{s}_k j}\,\mathbb{E}[\zeta_0^{(k)}]$, which is Eq. (12) in Sec. 4.4. If $\lambda_k = 0$, then $\mathbb{E}[\zeta_j^{(k)}] = 0$ for all $j \geq 1$ and we omit this mode; equivalently one may view $\Re(\bar{s}_k) = -\infty$. Note that $\bar{s}_k$ is per event index $j$, not per unit time.

**Conditional/nonstationary gaps.** If the gaps are not i.i.d. but are conditionally distributed given the past event history, let $\mathcal{F}_j^{\mathrm{evt}}$ denote the event-history filtration. Then from Eq. (66),

$$\mathbb{E}[\zeta_{j+1}^{(k)}|\mathcal{F}_j^{\mathrm{evt}}] = \mathbb{E}\left[\mathrm{e}^{s_k \Delta_{j+1}}|\mathcal{F}_j^{\mathrm{evt}}\right]\zeta_j^{(k)}, \tag{70}$$

### C.2. Real $2 \times 2$ block form

In implementation we represent the conjugate pair $-\rho_k \pm i\omega_k$ using the real block (Appendix B)

$$\boldsymbol{A}_k = \begin{bmatrix} -\rho_k & -\omega_k \\ \omega_k & -\rho_k \end{bmatrix} = -\rho_k \boldsymbol{I}_2 + \omega_k \boldsymbol{J}_2, \qquad \boldsymbol{J}_2 := \begin{bmatrix} 0 & -1 \\ 1 & 0 \end{bmatrix}, \qquad \boldsymbol{J}_2^2 = -\boldsymbol{I}_2, \tag{71}$$

where $\boldsymbol{J}_2$ is a $2 \times 2$ rotation generator. Let $\boldsymbol{\xi}^{(k)}(t) := [\Re(\zeta^{(k)}(t)), \Im(\zeta^{(k)}(t))]^\top \in \mathbb{R}^2$. Then $\boldsymbol{\xi}^{(k)}(t) = \mathrm{e}^{\boldsymbol{A}_k(t-t_0)}\boldsymbol{\xi}^{(k)}(t_0)$, and at events,

$$\boldsymbol{\xi}_{j+1}^{(k)} = \mathrm{e}^{\boldsymbol{A}_k \Delta_{j+1}}\boldsymbol{\xi}_j^{(k)}. \tag{72}$$

Under the same independence assumption $\Delta_{j+1} \perp \boldsymbol{\xi}_j^{(k)}$,

$$\mathbb{E}[\boldsymbol{\xi}_{j+1}^{(k)}] = \mathbb{E}[\mathrm{e}^{\boldsymbol{A}_k \Delta_{j+1}}]\mathbb{E}[\boldsymbol{\xi}_j^{(k)}] := \Phi_k \, \mathbb{E}[\boldsymbol{\xi}_j^{(k)}], \qquad \Phi_k := \mathbb{E}[\mathrm{e}^{\boldsymbol{A}_k \Delta}], \tag{73}$$

and thus $\mathbb{E}[\boldsymbol{\xi}_j^{(k)}] = \Phi_k^j \mathbb{E}[\boldsymbol{\xi}_0^{(k)}]$, i.e., Eq. (13) in Sec. 4.4.

**Closed form of $\mathrm{e}^{\boldsymbol{A}_k t}$ and $\Phi_k$.** Using Eq. (71) and $\boldsymbol{J}_2^2 = -\boldsymbol{I}_2$,

$$\mathrm{e}^{\boldsymbol{A}_k t} = \mathrm{e}^{-\rho_k t}\mathrm{e}^{\omega_k \boldsymbol{J}_2 t} = \mathrm{e}^{-\rho_k t}\Big(\cos(\omega_k t)\boldsymbol{I}_2 + \sin(\omega_k t)\boldsymbol{J}_2\Big) := \mathrm{e}^{-\rho_k t}\,\mathrm{Rot}(\omega_k t),$$

where $\mathrm{Rot}(\theta) := \begin{bmatrix} \cos\theta & -\sin\theta \\ \sin\theta & \cos\theta \end{bmatrix}$. Taking expectations gives

$$\Phi_k = \mathbb{E}[\mathrm{e}^{\boldsymbol{A}_k \Delta}] = \begin{bmatrix} \lambda_k^{\mathrm{R}} & -\lambda_k^{\mathrm{I}} \\ \lambda_k^{\mathrm{I}} & \lambda_k^{\mathrm{R}} \end{bmatrix}, \qquad \lambda_k^{\mathrm{R}} := \mathbb{E}\big[\mathrm{e}^{-\rho_k \Delta}\cos(\omega_k \Delta)\big], \qquad \lambda_k^{\mathrm{I}} := \mathbb{E}\big[\mathrm{e}^{-\rho_k \Delta}\sin(\omega_k \Delta)\big]. \tag{74}$$

The eigenvalues of $\Phi_k$ are $\lambda_k^{\mathrm{R}} \pm i\lambda_k^{\mathrm{I}}$. Moreover,

$$\lambda_k^{\mathrm{R}} + i\lambda_k^{\mathrm{I}} = \mathbb{E}\big[\mathrm{e}^{-\rho_k \Delta}\big(\cos(\omega_k \Delta) + i\sin(\omega_k \Delta)\big)\big] = \mathbb{E}[\mathrm{e}^{(-\rho_k + i\omega_k)\Delta}] = \lambda_k, \tag{75}$$

so the $2 \times 2$ real-block picture is exactly consistent with the complex scalar multiplier in Appendix C.1.

**Effective generator via logarithms.** If $\Phi_k$ has no eigenvalues on $(-\infty, 0]$ (equivalently $\lambda_k \notin (-\infty, 0]$), the principal matrix logarithm exists and satisfies $\log(\Phi_k) = \boldsymbol{V}\log(\boldsymbol{\Lambda})\boldsymbol{V}^{-1}$ for a diagonalization $\Phi_k = \boldsymbol{V}\boldsymbol{\Lambda}\boldsymbol{V}^{-1}$. If $\lambda_k$ lies on (or numerically close to) the branch cut $(-\infty, 0]$, one may instead use a continuous choice of $\mathrm{Arg}(\lambda_k)$ (phase unwrapping) to define a consistent logarithm, or treat it as a limiting case. Define $\bar{\boldsymbol{A}}_k := \log(\Phi_k)$ so that $\Phi_k = \mathrm{e}^{\bar{\boldsymbol{A}}_k}$ and $\mathbb{E}[\boldsymbol{\xi}_j^{(k)}] = \mathrm{e}^{\bar{\boldsymbol{A}}_k j}\mathbb{E}[\boldsymbol{\xi}_0^{(k)}]$. Write $\lambda_k = |\lambda_k|\mathrm{e}^{i\,\mathrm{Arg}(\lambda_k)}$ with $\mathrm{Arg}(\lambda_k) \in (-\pi, \pi]$. Then

$$\bar{\boldsymbol{A}}_k = \begin{bmatrix} -\bar{\rho}_k & -\bar{\omega}_k \\ \bar{\omega}_k & -\bar{\rho}_k \end{bmatrix}, \qquad \bar{\rho}_k := -\log|\lambda_k|, \qquad \bar{\omega}_k := \mathrm{Arg}(\lambda_k), \tag{76}$$

with phase unwrapping if a continuous branch is selected. Note that $\bar{\omega}_k$ is an event-domain phase increment (radians per event), not an angular frequency per unit time.

### C.3. Stability under renewal averaging

With $\Re(s_k) < 0$ and $\Delta \geq 0$, by Jensen's inequality (equivalently, $|\mathbb{E}(\cdot)| \leq \mathbb{E}(|\cdot|)$),

$$|\lambda_k| = \big|\mathbb{E}[\mathrm{e}^{s_k \Delta}]\big| \leq \mathbb{E}\big[|\mathrm{e}^{s_k \Delta}|\big] = \mathbb{E}\big[\mathrm{e}^{\Re(s_k)\Delta}\big] = \mathbb{E}\big[\mathrm{e}^{-\rho_k \Delta}\big] \leq 1, \tag{77}$$

where we used the triangle inequality and $\mathrm{e}^{-\rho_k \Delta} \leq 1$. If $\mathbb{P}(\Delta > 0) > 0$ and $\rho_k > 0$, then $\mathrm{e}^{-\rho_k \Delta} < 1$ on a set of positive probability, implying $\mathbb{E}[\mathrm{e}^{-\rho_k \Delta}] < 1$, hence $|\lambda_k| < 1$. Because $\Re(\log \lambda_k) = \log|\lambda_k|$, we obtain

$$\Re(\bar{s}_k) = \log|\lambda_k| \leq 0 \quad (\text{strict } < 0 \text{ when } \mathbb{P}(\Delta > 0) > 0),$$

i.e., renewal averaging preserves (and typically strengthens) mean stability.

**Why random gaps can be more contractive for oscillatory modes.** Decompose

$$\lambda_k = \mathbb{E}\left[\mathrm{e}^{(-\rho_k + i\omega_k)\Delta}\right] = \mathbb{E}\left[\mathrm{e}^{-\rho_k \Delta}\mathrm{e}^{i\omega_k \Delta}\right]. \tag{78}$$

Define the exponentially tilted law $Q_k$ on $\Delta$ by the Radon–Nikodym derivative

$$\frac{dQ_k}{d\mathbb{P}}(\Delta) := \frac{\mathrm{e}^{-\rho_k \Delta}}{\mathbb{E}[\mathrm{e}^{-\rho_k \Delta}]}, \tag{79}$$

which is a valid density since $\mathrm{e}^{-\rho_k\Delta} \geq 0$ and $\mathbb{E}[\mathrm{e}^{-\rho_k\Delta}] > 0$. Then Eq. (78) factorizes as

$$\lambda_k = \mathbb{E}[\mathrm{e}^{-\rho_k\Delta}] \cdot \mathbb{E}_{Q_k}[\mathrm{e}^{i\omega_k\Delta}] =: \mathbb{E}[\mathrm{e}^{-\rho_k\Delta}] \cdot \varphi_{Q_k}(\omega_k), \tag{80}$$

where $\varphi_{Q_k}$ is the characteristic function under $Q_k$. Since $|\mathrm{e}^{i\omega\Delta}| = 1$,

$$|\varphi_{Q_k}(\omega)| = |\mathbb{E}_{Q_k}[\mathrm{e}^{i\omega\Delta}]| \leq \mathbb{E}_{Q_k}[|\mathrm{e}^{i\omega\Delta}|] = 1, \tag{81}$$

with equality only when $\omega\Delta$ is almost surely constant modulo $2\pi$. Thus,

$$|\lambda_k| = \mathbb{E}[\mathrm{e}^{-\rho_k\Delta}] |\varphi_{Q_k}(\omega_k)| \leq \mathbb{E}[\mathrm{e}^{-\rho_k\Delta}],$$

showing an additional attenuation term from phase cancellation when $\omega_k \neq 0$ and $\Delta$ is non-degenerate.

### C.4. Second-order approximation

Let $M_\Delta(s) := \mathbb{E}[\mathrm{e}^{s\Delta}]$, which is finite for all $s$ with $\Re(s) \leq 0$ and in particular at $s = s_k$. Then

$$\bar{s}_k = \log M_\Delta(s_k). \tag{82}$$

Assuming $\mathbb{E}[\Delta^2] < \infty$, the cumulant expansion of $\log M_\Delta(s)$ around $s = 0$ gives

$$\log M_\Delta(s) = \kappa_1 s + \frac{\kappa_2}{2}s^2 + O(s^3) = s\,\mathbb{E}[\Delta] + \frac{s^2}{2}\mathrm{Var}(\Delta) + O(s^3), \tag{83}$$

where $\kappa_1 = \mathbb{E}[\Delta]$ and $\kappa_2 = \mathrm{Var}(\Delta)$ are the first two cumulants. Substituting $s = s_k$ yields Eq. (15):

$$\bar{s}_k \approx s_k\,\mathbb{E}[\Delta] + \frac{s_k^2}{2}\mathrm{Var}(\Delta). \tag{84}$$

Writing $s_k = -\rho_k + i\omega_k$ and $s_k^2 = (\rho_k^2 - \omega_k^2) - 2i\rho_k\omega_k$ makes explicit that variability perturbs both event-domain decay and phase:

$$\Re(\bar{s}_k) \approx -\rho_k\mathbb{E}[\Delta] + \frac{\rho_k^2 - \omega_k^2}{2}\mathrm{Var}(\Delta), \qquad \Im(\bar{s}_k) \approx \omega_k\mathbb{E}[\Delta] - \rho_k\omega_k\,\mathrm{Var}(\Delta). \tag{85}$$

The approximation is intended for intuition (moderate variability / small higher-order cumulants), complementing the exact bounds in Appendix C.3.

## D. Preprocessing & Baseline Handling

**Unified cache and windowing.** All datasets are converted into a shared `ratio-index` cache format that stores per-entity feature matrices and targets on a shared timestamp grid, alongside a global window index of (`entity_id`, `start_idx`) pairs and the corresponding context end-times. We set the shared grid resolution to the dataset's native sampling interval (reported per dataset in Tab. 5). We use the dataset's native interval as an indexing grid for windowing; observations are assigned to the nearest native bin for consistent batching (no interpolation), and we preserve missingness masks and time metadata (timestamps or per-step deltas) for irregularity-aware models. This allows every method to train/evaluate on identical context-horizon slices $(\ell, h)$ without dataset-specific glue. For each entity, we keep only windows whose forecast horizon contains at least one observed target value, ensuring well-defined evaluation under sparse observations. *Coverage* denotes the percentage of observed (non-missing) values per timestamp.

**Cleaning, filling, and missingness.** Raw missing values are preserved as `NaN` during cleaning at their original timestamps. When forward-filling is used during preprocessing (dataset-dependent), we also retain binary masks that distinguish originally observed entries from filled ones. Dataset statistics, including average per-timestep coverage, are reported in Tab. 9.

**Normalization and model inputs.** We standardize inputs (per-entity when applicable) using training statistics. To ensure all backbones receive dense tensors, remaining `NaN` entries are replaced with zero after normalization (i.e., zero corresponds to the training-set mean in the original scale under standardization), and masks are passed alongside the inputs. Concretely, each batch provides (i) inputs and first differences (with masks), and (ii) metadata containing observation/fill masks and per-step time deltas (or timestamps) for irregularity-aware baselines.

**Baselines and irregularity handling.** All baselines consume the same normalized windows and targets. Methods with native irregular-time support (e.g., NeuralCDE/mTAN/ContiFormer/T-PATCHGNN) additionally receive time metadata (timestamps or time deltas) and masks when required by their standard implementations. mTAN is evaluated as a probabilistic model (Gaussian likelihood), and we compute CRPS from its predictive distribution (estimated with the same sampling protocol as other probabilistic models). Regular-grid deterministic baselines (e.g., DLinear/PatchTST) operate on the dense sequences (with temporal encoding (timestamps and masks) as additional features for conditioning, according to our renewal-averaging view); masking is applied in the loss to evaluate only on observed targets. Diffusion-based baselines (e.g., CSDI/mr-Diff/TimeGrad) operate on masked raw sequences with temporal encoding (timestamps) as additional conditioning features. We follow the standard implementations for baseline model configurations, **except for mr-Diff**, as the paper does not have code to replicate; we implement with our version based on the paper.

**Meteorology and finance dataset construction.** We construct the meteorology datasets (NOAA-UK and NOAA-US) from the **NOAA/NCEI Integrated Surface Database (Global Hourly)**. NOAA-UK uses 7 stations with hourly data from 2021-10-05T14:00:00 to 2024-12-31T23:00:00 (UTC), and NOAA-US uses 45 stations from 2021-01-01T01:00:00 to 2024-12-31T23:00:00 (UTC). Meteorology features include temperature, dew point, sea-level pressure, wind speed, and precipitation. We construct the finance datasets (Crypto and US Equity) from **Google Finance** daily market data. Crypto spans 2019-01-01 to 2024-12-31 with the top 118 (volume and market cap) most actively traded currencies selected; US Equity spans 2017-01-01 to 2024-12-31 with the top 222 (volume and market cap) most actively traded equities selected from NYSE and NASDAQ. Finance features include open, high, low, close, volume, daily returns (percent; adjacent trading days), gapped returns, a market index feature, 20-day rolling close, high–low ratio, dividends, and sinusoidal calendar features (day-of-week, day-of-month, month-of-year).

*Table 9.* Dataset statistics.

| Dataset | # Input channels ($d_x$) | # Entities ($N$) | # Tran+Val+Test Samples | Coverage (min / mean) | Valid target channels ($d_y$) | Irregularity type |
|---|---|---|---|---|---|---|
| BMS Air Quality | 12 | 12 | 414,710 | 83.3% / 100.0% | 11 | missingness + gaps |
| UCI Air Quality | 12 | 1 | 8,854 | 100.0% / 100.0% | 12 | N/A |
| PhysioNet CinC | 4 | 1022 | 12,497 | 7.4% / 87.3% | 1 | time jitter (raw; aligned to shared timestamps) + missingness + gaps |
| NOAA US | 5 | 45 | 1,541,698 | 37.8% / 99.1% | 4 | missingness + gaps |
| NOAA UK | 5 | 7 | 194,509 | 14.3% / 99.6% | 4 | missingness + gaps |
| US Equity | 16 | 222 | 397,586 | 99.5% / 99.9% | 8 | missingness + gaps |
| Crypto | 16 | 118 | 91,809 | 59.3% / 88.1% | 8 | missingness + gaps |

# E. Pre-training

## E.1. Latent Target Pre-training

We pre-train a Transformer VAE to construct latent trajectories over the target horizon. The encoder/decoder architecture follows the VAE hyperparameters in Tab. 10. Let $\mathcal{Y} \in \mathbb{R}^{h \times N \times d_y}$ denote the target window over the $h$ query times $\{t_r^q\}_{r=1}^h$, and let $\mathcal{M}^y \in \{0,1\}^{h \times N \times d_y}$ be the corresponding stacked query mask (for $\{t_r^q\}_{r=1}^h$ in Sec. 3). We avoid temporal downsampling: the latent sequence retains length $h$ to preserve temporal fidelity. Instead, the encoder compresses across entity/variable dimensions at each time step via set encoding over entities.

**Entity-set encoder/decoder.** At each query index $r$, we treat the $N$ entities as a set of tokens. For each entity $n$, we zero-fill missing entries $\tilde{\mathcal{Y}}_{r,n} = \mathcal{M}_{r,n}^y \odot \mathcal{Y}_{r,n}$ and form an entity token by concatenating the mask, $[\tilde{\mathcal{Y}}_{r,n}; \mathcal{M}_{r,n}^y] \in \mathbb{R}^{2d_y}$, followed by a linear projection to width $d_{\text{vae}}$. We then apply $L$ layers of self-attention across the $N$ entity tokens (optionally using key-padding masks when entity tokens are padded), and mean-pool over entities to obtain a single per-query representation. This representation is mapped to posterior parameters $(\boldsymbol{\mu}_r, \log \boldsymbol{\sigma}_r) \in \mathbb{R}^{d_z}$, yielding a factorized approximate posterior

$$q_{\phi_{\text{VAE}}}(\boldsymbol{z} \mid \mathcal{Y}) = \prod_{r=1}^h \mathcal{N}(\boldsymbol{z}_r; \boldsymbol{\mu}_r, \text{diag}(\boldsymbol{\sigma}_r^2)).$$

The decoder mirrors this by broadcasting each $\boldsymbol{z}_r$ to $N$ entity tokens, applying the same entity-attention blocks, and mapping tokens back to $\hat{\mathcal{Y}}_{r,n} \in \mathbb{R}^{d_y}$.

**Window-relative positional encoding.** To make decoding invariant to the absolute placement of a window in continuous time, we use learnable window-relative positional embeddings: indices $1, \ldots, h$ are reused for every window (equivalently, the positional embedding is "reset" per window). Note that while the latent diffusion process operates in continuous time, the VAE decoder utilizes this discrete indexing corresponding to the dataset's native resolution. This encourages the decoder

to act as a time-invariant reconstructor over relative offsets; query times are handled by aligning the decoded latent sequence to the window's chosen query set.

**Objective.** We optimize a masked Gaussian ELBO with an isotropic Gaussian prior,

$$\mathcal{L}_{\text{VAE}} = \underbrace{\frac{1}{\sum_{r,n,d} \mathcal{M}^y_{r,n,d}} \sum_{r,n,d} \mathcal{M}^y_{r,n,d} \left(\mathcal{Y}_{r,n,d} - \hat{\mathcal{Y}}_{r,n,d}\right)^2}_{\mathcal{L}_{\text{recon}}} + \beta_{\text{KL}} \cdot D_{\text{KL}}\big(q_{\phi_{\text{VAE}}}(\boldsymbol{z} \mid \mathcal{Y}) \,\|\, \mathcal{N}(\boldsymbol{0}, \boldsymbol{I})\big). \tag{86}$$

We mitigate posterior collapse by annealing $\beta_{\text{KL}}$ from $0$ to a final value $\beta_{\text{VAE}}$ during early epochs. We use the posterior mean $\boldsymbol{z}_r = \boldsymbol{\mu}_r$ as the latent trajectory.

## E.2. Summarizer Pre-training

The summarizer produces a history summary token sequence from an input history tensor $\mathcal{X} \in \mathbb{R}^{\ell \times N \times d_x}$ with missingness mask $\mathcal{M}^x \in \{0,1\}^{\ell \times N \times d_x}$ and shared window timestamps $\boldsymbol{t} \in \mathbb{R}^\ell$. It consists of three lightweight components: (i) feature mixing over time, (ii) proxy dynamics signals, and (iii) timestamp encoding, followed by attention-based pooling to obtain history summary token sequence.

- **Feature mixing (port/forcing token).** The masked input history is first processed by a 1D convolution (kernel $k = 3$, stride 1) along time. This soft patching mixes local temporal features while preserving the sequence length $\ell$, projecting $d_x$ features to a mixing dimension $d_{\text{mix}}$.

- **Proxy dynamics signals.** We compute two auxiliary scalar proxies per time/entity. The first proxy $\text{V}_{\text{sig}}$ (potential-like) is derived directly from the raw input via a lightweight MLP: $\text{V}_{\text{sig}} = \text{MLP}_\text{V}(\mathcal{M}^x \odot \mathcal{X}) \in \mathbb{R}^{\ell \times N \times 1}$. The second proxy $\text{T}_{\text{sig}}$ (kinetic-like) is derived from masked discrete differences of the input. Let

$$\Delta \mathcal{X}_{j,n} := (\mathcal{X}_{j,n} - \mathcal{X}_{j-1,n}) \odot \mathcal{M}^\Delta_{j,n}, \qquad \mathcal{M}^\Delta_{j,n} := \mathcal{M}^x_{j,n} \odot \mathcal{M}^x_{j-1,n},$$

with $\Delta \mathcal{X}_{1,n} := \boldsymbol{0}$. We then define $\text{T}_{\text{sig}} = \text{MLP}_\text{T}(\Delta \mathcal{X}) \in \mathbb{R}^{\ell \times N \times 1}$. We form content tokens by concatenating mixed features with two proxies and a missingness proxy, yielding a content width $d_{\text{mix}} + 3$.

- **Timestamp encoding (Time2Vec).** We embed each window timestamp $t_j$ using a Time2Vec map $\text{T2V}(\cdot)$. We use relative time within the window, $\tilde{t}_j = t_j - t_1$, and compute $\boldsymbol{e}_j = \text{T2V}(\tilde{t}_j) \in \mathbb{R}^{d_t}$. We then concatenate content and time for every entity-time token:

$$\mathcal{X}^t_{j,n} = \text{concat}\big(\mathcal{X}^{\text{fuse}}_{j,n}, \boldsymbol{e}_j\big) \in \mathbb{R}^{d_{\text{enc}}}, \qquad d_{\text{enc}} = d_{\text{mix}} + 3 + d_t.$$

With the default $d_{\text{mix}} = 64$ and $d_t = 9$, $d_{\text{enc}} = 76$, so no padding projection is required before four-head attention.

- **Summary token sequence via self-attention.** We apply a Transformer encoder over time to each entity timeline, then mean pool the encoded timelines over valid entities:

$$\boldsymbol{H}_{:,n} = \text{TransformerEnc}\big(\mathcal{X}^t_{:,n} + \boldsymbol{P}_{\text{win}}\big) \in \mathbb{R}^{\ell \times d_{\text{enc}}}, \qquad \tilde{\boldsymbol{H}}_j = \text{pool}_n(\boldsymbol{H}_{j,n}) \in \mathbb{R}^{d_{\text{enc}}}.$$

A linear projection maps $\tilde{\boldsymbol{H}}_j$ from $d_{\text{enc}} = 76$ to $d_{\text{ctx}} = 256$. Learned query pooling over the $\ell$ projected time tokens produces the summary token sequence $\boldsymbol{E}_{t_i} \in \mathbb{R}^{S \times d_{\text{ctx}}}$, where $S$ is the configured summary length. Here $\boldsymbol{P}_{\text{win}} \in \mathbb{R}^{\ell \times d_{\text{enc}}}$ is the learnable window-relative positional embedding shared across windows.

- **Training objective.** The summarizer is pre-trained to reconstruct the masked input history and the auxiliary proxies from the summary sequence $\boldsymbol{E}_{t_i}$ using lightweight decoder heads:

$$\mathcal{L}_{\text{sum}} = w_\text{X}\mathcal{L}^\text{X}_{\text{rec}} + w_\text{V}\mathcal{L}^\text{V}_{\text{rec}} + w_\text{T}\mathcal{L}^\text{T}_{\text{rec}} + w_{\Delta t}\mathcal{L}^{\Delta t}_{\text{rec}} + w_{\text{obs}}\mathcal{L}^{\text{obs}}_{\text{rec}}.$$

where $\mathcal{L}^\text{X}_{\text{rec}}$ reconstructs $\mathcal{X}$ on valid entries, while $\mathcal{L}^\text{V}_{\text{rec}}$ and $\mathcal{L}^\text{T}_{\text{rec}}$ reconstruct $\text{V}_{\text{sig}}$ and $\text{T}_{\text{sig}}$, respectively; the last two terms reconstruct relative timestamp and observation-mask auxiliaries:

$$\mathcal{L}^\text{X}_{\text{rec}} = \frac{\sum_{j,n,d} \mathcal{M}^x_{j,n,d} \left\|\mathcal{X}_{j,n,d} - \hat{\mathcal{X}}_{j,n,d}\right\|^2}{\sum_{j,n,d} \mathcal{M}^x_{j,n,d}}, \ \mathcal{L}^\text{V}_{\text{rec}} = \frac{\sum_{j,n} \left\|\text{V}_{\text{sig},j,n} - \hat{\text{V}}_{\text{sig},j,n}\right\|^2_2}{\ell N}, \ \mathcal{L}^\text{T}_{\text{rec}} = \frac{\sum_{j,n} \left\|\text{T}_{\text{sig},j,n} - \hat{\text{T}}_{\text{sig},j,n}\right\|^2_2}{\ell N}.$$

We use $(w_\text{X}, w_\text{V}, w_\text{T}, w_{\Delta t}, w_{\text{obs}}) = (1, 0.1, 0.1, 0.05, 0.05)$ by default.

- **Lightweight decoder heads.** The summarizer is pretrained with auxiliary reconstruction heads applied to the pooled context representation. After query pooling, the context tokens are averaged,

$$\bar{\boldsymbol{E}} = \frac{1}{S} \sum_{s=1}^{S} \boldsymbol{E}_s \in \mathbb{R}^{d_{\mathrm{ctx}}}.$$

A compact MLP reconstructs the input panel, $\hat{\mathcal{X}} = \mathrm{reshape}\big(\boldsymbol{W}_2\,\mathrm{GeLU}(\boldsymbol{W}_1\bar{\boldsymbol{E}})\big) \in \mathbb{R}^{\ell \times N \times d_x}$, with hidden width $2d_{\mathrm{ctx}}$. Scalar proxy heads are direct linear projections,

$$\hat{\mathrm{V}}_{\mathrm{sig}} = \mathrm{reshape}(\boldsymbol{W}_{\mathrm{V}}\bar{\boldsymbol{E}}) \in \mathbb{R}^{\ell \times N}, \qquad \hat{\mathrm{T}}_{\mathrm{sig}} = \mathrm{reshape}(\boldsymbol{W}_{\mathrm{T}}\bar{\boldsymbol{E}}) \in \mathbb{R}^{\ell \times N},$$

and the same pattern is used for the timestamp and observation-mask auxiliaries. These heads are used only as lightweight pretraining losses; the diffusion model conditions on the summary tokens, not on the decoder outputs.

## F. Realizations of Modal Predictor & Modal Synthesizer

We implement the denoiser using modal predictor and modal synthesizer blocks, which operate in the latent space. Given a noisy latent trajectory $\boldsymbol{z}_\tau \in \mathbb{R}^{B \times h \times d_z}$ at diffusion step $\tau$ and a history summary token sequence $\boldsymbol{E}_{t_i}$, the model (i) predicts history-conditioned continuous-time poles, (ii) computes an initial set of modal residues (cosine/sine components) from $\boldsymbol{z}_\tau$, (iii) refines these residues through a stack of lightweight Transformer blocks in modal-token space (optionally attending to summary tokens), and (iv) synthesizes the denoised latent trajectory $\hat{\boldsymbol{z}}_0$ in one parallelizable pass.

### F.1. Modal Predictor $\mathcal{L}_\theta$

The modal predictor produces history-conditioned continuous-time modal parameters $\hat{\vartheta} := \mathcal{L}_\theta(\boldsymbol{z}_\tau, \tau, \boldsymbol{E}_{t_i})$, $\hat{\vartheta} = \{(\hat{\rho}_k, \hat{\omega}_k, \hat{\boldsymbol{c}}_k, \hat{\boldsymbol{b}}_k)\}_{k=1}^{K}$, where $\hat{\rho}_k > 0$ and $\hat{\omega}_k > 0$ denote decay/frequency parameters (stable conjugate poles), and $\hat{\boldsymbol{c}}_k, \hat{\boldsymbol{b}}_k \in \mathbb{R}^{d_z}$ are cosine/sine latent residues. We parameterize the base frequency as $\omega_k = \omega_{\max}\,\mathrm{Sigmoid}(\varphi_k)$ so that $\mathrm{logit}(\omega_k/\omega_{\max}) = \varphi_k$. For implementation, we enforce $\hat{\rho}_k > 0$ via Softplus and bound $\hat{\omega}_k \in (0, \omega_{\max})$ via Sigmoid, allowing small $\hat{\omega}_k$ (near-real poles). Let $\{t_r^q\}_{r=1}^{h}$ denote the queried timestamps of the latent trajectory (forecast horizon at test time, or queried times for imputation). We form relative times $\tilde{t}_r := t_r^q - t_1^q$ (equivalently via cumulative $\Delta t$), so that $\tilde{t}_1 = 0$ and $\tilde{t} = 0$ refers to the earliest query time. In forecasting, $t_1^q$ is typically the first query time after the context end $t_i$, so $\tilde{t}$ measures offsets from forecast start; for imputation, queries may satisfy $t_r^q \leq t_i$. In both cases, the absolute positioning relative to the context (including any gap between $t_i$ and $t_1^q$) is provided via the conditioning tokens.

**History-conditioned continuous-time poles (computed once).** Let $\mathbf{e}_\tau$ be the diffusion-step embedding and let $\mathrm{pool}(\boldsymbol{E}_{t_i})$ denote mean pooling over summary tokens. We form a conditioning vector $\mathbf{h}_i = \mathrm{concat}(\mathbf{e}_\tau, \mathrm{pool}(\boldsymbol{E}_{t_i})) \in \mathbb{R}^{2d_{\mathrm{model}}}$. We maintain global base poles $(\rho_k, \omega_k)$ and predict bounded perturbations from $\mathbf{h}_i$ with an MLP:

$$(\Delta\rho_k, \Delta\omega_k) = \mathrm{MLP}(\mathbf{h}_i),\ \Delta\rho_k, \Delta\omega_k\ \mathrm{bounded\ via}\ \tanh.$$

The continuous-time poles are obtained with stability constraints enforced by construction:

$$\hat{\rho}_k = \mathrm{Softplus}(\rho_k + \Delta\rho_k) + \rho_{\min},\ \hat{\omega}_k = \omega_{\max}\,\mathrm{Sigmoid}\big(\mathrm{logit}(\omega_k/\omega_{\max}) + \Delta\omega_k\big),$$

where $\omega_{\max}$ is an optional upper bound for numerical stability. In implementation, $(\hat{\rho}, \hat{\omega})$ are computed once per denoising forward pass and reused across the stacked modal refinement blocks.

**Stable damped-sinusoid basis.** For each mode $k \in \{1, \ldots, K\}$, we define $\chi_k^{(c)}(\tilde{t}) = \mathrm{e}^{-\hat{\rho}_k\tilde{t}}\cos(\hat{\omega}_k\tilde{t})$, $\chi_k^{(s)}(\tilde{t}) = \mathrm{e}^{-\hat{\rho}_k\tilde{t}}\sin(\hat{\omega}_k\tilde{t})$. We stack these into a design matrix $\mathbf{A}_{\mathrm{lap}}(\tilde{t}; \hat{\rho}, \hat{\omega}) \in \mathbb{R}^{B \times h \times 2K}$ by concatenating $\{\chi_k^{(c)}\}_{k=1}^{K}$ and $\{\chi_k^{(s)}\}_{k=1}^{K}$ along the last dimension.

**Initial residues from noisy latent (computed once).** Given $\mathbf{A}_{\mathrm{lap}}$, we compute initial residues $\hat{\mathbf{R}} \in \mathbb{R}^{B \times 2K \times d_z}$ from $\boldsymbol{z}_\tau$. Unless stated otherwise, we use time cross-attention for residue initialization.

**Learned time-to-mode projection (time cross-attention).** We embed each pole pair $(\hat{\rho}_k, \hat{\omega}_k)$ into a mode query and embed each timestamp $\tilde{t}_j$ into a time key. Values are obtained by projecting $\boldsymbol{z}_\tau(\tilde{t}_j)$. Cross-attention produces residues as weighted sums over time. We view the first $K$ rows of $\hat{\mathbf{R}}$ as cosine residues $\{\hat{\boldsymbol{c}}_k\}_{k=1}^{K}$ and remaining $K$ rows as sine residues $\{\hat{\boldsymbol{b}}_k\}_{k=1}^{K}$.

**Stacked modal refinement blocks (update residues).** We treat the $2K$ residue vectors in $\hat{\mathbf{R}} \in \mathbb{R}^{B \times 2K \times d_z}$ as modal tokens. Each block first projects residues to hidden tokens $\hat{\mathbf{R}}_h \in \mathbb{R}^{B \times 2K \times d_{\mathrm{model}}}$, adds (i) the diffusion-step embedding $\mathbf{e}_\tau$ and (ii) a learnable modal positional embedding, and then applies: (i) cross-attention from modal tokens to the summary token sequence $\boldsymbol{E}_{t_i}$ and (ii) self-attention mixing among modal tokens. The block outputs an additive residue update $\Delta\mathbf{R} \in \mathbb{R}^{B \times 2K \times d_z}$. $\hat{\mathbf{R}} \leftarrow \hat{\mathbf{R}} + \Delta\mathbf{R}$. Repeating $L$ blocks yields updated residues $\hat{\mathbf{R}}_{\mathrm{upd}}$, which correspond to updated $(\hat{\boldsymbol{c}}_k, \hat{\boldsymbol{b}}_k)$ while learned continuous-time poles $(\hat{\rho}_k, \hat{\omega}_k)$ remain the history-conditioned poles computed above.

### F.2. Modal Synthesizer $\mathcal{L}_\theta^+$

The modal synthesizer generates the denoised latent trajectory $\hat{\boldsymbol{z}}_0$ by explicit synthesis using the updated residues and the learned continuous-time poles. **Synthesis (computed once).** Using the same design matrix $\mathbf{A}_{\mathrm{lap}}(\tilde{t}; \hat{\rho}, \hat{\omega})$, we synthesize in parallel:

$$\hat{\boldsymbol{z}}_0 = \mathcal{L}_\theta^+(\hat{\vartheta}, \tau, \boldsymbol{E}_{t_i}) = \mathbf{A}_{\mathrm{lap}}(\tilde{t}; \hat{\rho}, \hat{\omega})\, \hat{\mathbf{R}}_{\mathrm{upd}} \in \mathbb{R}^{B \times h \times d_z}.$$

Equivalently, for each query time $\tilde{t}_j$, $\hat{\boldsymbol{z}}_0(\tilde{t}_j) = \sum_{k=1}^K e^{-\hat{\rho}_k \tilde{t}_j}\left(\hat{\boldsymbol{c}}_k \cos(\hat{\omega}_k \tilde{t}_j) + \hat{\boldsymbol{b}}_k \sin(\hat{\omega}_k \tilde{t}_j)\right)$.

**Residual refinement.** We apply a small residual correction (via a lightweight MLP) so the stable modal component dominates long-range behavior, while learning any deviations that are not fully captured by the modal basis.

## G. Training & Inference

---
**Algorithm 1** Training (latent diffusion)
---
1: **for** each training step **do**
2:    Sample history/target windows $(\mathcal{H}_{t_i}, \mathcal{Y}_{t_i}) \sim$ Trainset
3:    Encode target latent trajectory: $\boldsymbol{z}_0 \leftarrow \mathrm{VAE}_{\mathrm{enc}}(\mathcal{Y}_{t_i})$
4:    Build conditioning: $\boldsymbol{E}_{t_i} \leftarrow \mathcal{S}_\phi(\mathcal{H}_{t_i})$
5:    With probability $p_{\mathrm{uncond}}$, set $\boldsymbol{E}_{t_i} \leftarrow \varnothing$, (classifier-free)
6:    Sample diffusion step and noise: $\tau \sim \mathcal{U}(\{1, \ldots, T\})$, $\boldsymbol{\epsilon} \sim \mathcal{N}(\mathbf{0}, \boldsymbol{I})$
7:    Forward diffuse latent: $\boldsymbol{z}_\tau \leftarrow \sqrt{\bar{\alpha}_\tau}\, \boldsymbol{z}_0 + \sqrt{1 - \bar{\alpha}_\tau}\, \boldsymbol{\epsilon}$
8:    Predict synthesis parameters: $\hat{\vartheta} \leftarrow \mathcal{L}_\theta(\boldsymbol{z}_\tau, \tau, \boldsymbol{E}_{t_i})$
9:    Synthesize clean latent: $\hat{\boldsymbol{z}}_0 \leftarrow \mathcal{L}_\theta^+(\hat{\vartheta}, \tau, \boldsymbol{E}_{t_i})$
10:    Optionally refine: $\hat{\boldsymbol{z}}_0 \leftarrow \hat{\boldsymbol{z}}_0 + \mathrm{MLP}(\hat{\boldsymbol{z}}_0)$, (residual)
11:    Update $\theta$ to minimize $\left\| \boldsymbol{z}_0 - \hat{\boldsymbol{z}}_0 \right\|_2^2$
12: **end for**

---
**Algorithm 2** Inference (deterministic DDIM with guidance)
---
1: Build conditioning: $\boldsymbol{E}_{t_i} \leftarrow \mathcal{S}_\phi(\mathcal{H}_{t_i})$
2: Initialize latent: $\boldsymbol{z}_T \sim \mathcal{N}(\mathbf{0}, \boldsymbol{I})$
3: **for** $\tau = T, \ldots, 1$ **do**
4:    Conditional params: $\hat{\vartheta}^c \leftarrow \mathcal{L}_\theta(\boldsymbol{z}_\tau, \tau, \boldsymbol{E}_{t_i})$
5:    Unconditional params: $\hat{\vartheta}^u \leftarrow \mathcal{L}_\theta(\boldsymbol{z}_\tau, \tau, \varnothing)$
6:    $\hat{\boldsymbol{z}}_0^c \leftarrow \mathcal{L}_\theta^+(\hat{\vartheta}^c, \tau, \boldsymbol{E}_{t_i})$, $\hat{\boldsymbol{z}}_0^u \leftarrow \mathcal{L}_\theta^+(\hat{\vartheta}^u, \tau, \varnothing)$
7:    Guided estimate: $\hat{\boldsymbol{z}}_0 \leftarrow \hat{\boldsymbol{z}}_0^u + w(\hat{\boldsymbol{z}}_0^c - \hat{\boldsymbol{z}}_0^u)$
8:    Recover noise: $\hat{\boldsymbol{\epsilon}} \leftarrow \dfrac{\boldsymbol{z}_\tau - \sqrt{\bar{\alpha}_\tau}\, \hat{\boldsymbol{z}}_0}{\sqrt{1 - \bar{\alpha}_\tau}}$
9:    DDIM step: $\boldsymbol{z}_{\tau-1} \leftarrow \sqrt{\bar{\alpha}_{\tau-1}}\, \hat{\boldsymbol{z}}_0 + \sqrt{1 - \bar{\alpha}_{\tau-1}}\, \hat{\boldsymbol{\epsilon}}$
10:    Set $\boldsymbol{z}_\tau \leftarrow \boldsymbol{z}_{\tau-1}$
11: **end for**
12: Decode: $\hat{\mathcal{Y}}_{t_i} \leftarrow \mathrm{VAE}_{\mathrm{dec}}(\hat{\boldsymbol{z}}_0)$
13: **return** $\hat{\mathcal{Y}}_{t_i}$

# H. Diffusion Setting

*Table 10.* **Hyperparameter settings**

| Hyperparameter | BMS / UCI / PhysioNet / NOAA US / NOAA UK / US Equity / Crypto |
|---|---|
| Prediction horizon | 168 / 168 / 12 / 168 / 168 / 100 / 100 |
| Context length $\ell$ | 336 / 336 / 24 / 336 / 336 / 200 / 200 |
| Batch size | 10 / 10 / 5 / 15 / 15 / 5 / 5 |
| **VAE** | |
| Latent channels ($d_z$) | 24 / 16 / 16 / 24 / 16 / 12 / 16 |
| Transformer width ($d_{\mathrm{vae}}$) | 128 |
| FFN dim | 256 |
| Encoder & decoder layers | 3 |
| Heads | 4 |
| Dropout | 0.1 |
| LR / weight decay | $1\times10^{-4}$ / $1\times10^{-4}$ |
| KL weight ($\beta_{\mathrm{VAE}}$) | $1\times10^{-3}$ |
| KL warmup / anneal / min epochs | 5 / 25 / 40 |
| Early-stop patience | 20 |
| **History summarizer** | |
| Summary length | 336 / 336 / 24 / 336 / 336 / 200 / 200 |
| Summary dim ($d_{\mathrm{ctx}}$) | 256 |
| Feature mixing dim ($d_{\mathrm{mix}}$) | 64 |
| Time2Vec dim ($d_t$) | 9 |
| Fused encoder dim ($d_{\mathrm{enc}}$) | 76 |
| Temporal encoder layers / heads | 2 / 4 |
| Patch kernel | 3 |
| Dropout | 0.1 |
| Proxy scalar hidden dim ($d_{\mathrm{V}}/d_{\mathrm{T}}$) | 32 |
| LR | $1\times10^{-4}$ / $5\times10^{-4}$ / $5\times10^{-4}$ / $5\times10^{-4}$ / $5\times10^{-4}$ / $5\times10^{-4}$ / $5\times10^{-4}$ |
| Weight decay / grad clip | $1\times10^{-4}$ / 1.0 |
| Max epochs / early-stop patience | 200 / 10 |
| **Diffusion** | |
| Diffusion steps | 1000 |
| Noise schedule | cosine |
| Prediction type | `x0` or `v` |
| Loss weighting | `weighted_min_snr` |
| Weighted_min_snr $\gamma$ | 5.0 / 4.5 / 5.0 / 4.5 / 4.5 / 5.0 / 5.0 |
| Model width ($d_{\mathrm{model}}$) | 256 |
| Layers ($L$) / heads | 5 / 4 |
| Number of poles $K$ | 256 |
| Dropout / attn dropout | 0.0 / 0.0 |
| Drop conditioning probability ($p_{\mathrm{uncond}}$) | 0.18 |
| $\rho_{\min}$ / $\omega_{\max}$ | $1\times10^{-6}$ / $\pi$ |
| Pole perturbation tanh scale ($\Delta\rho/\Delta\omega$) | 0.5 / 0.5 |
| **Optimization & evaluation** | |
| Optimizer | AdamW |
| Max epochs | 600 |
| Base LR / min LR | $1.5\times10^{-4}$ / $3\times10^{-6}$ |
| LR schedule / warmup fraction | `warmup_constant` / 0.095 |
| Weight decay / grad clip | $5\times10^{-4}$ / 1.0 |
| Early-stop patience | 50 |
| EMA eval / decay | `True` / 0.999 |
| **Sampling / generation** | |
| Sampling steps | 64 |
| Sampler / default schedule | DDIM / `karras` |
| DDIM $\eta$ | 0.0 |
| Guidance strength $w$ | (1.0, 2.0) |
| Guidance power | 1.0 |
| Dynamic threshold $p$ / max | 0.995 / 1.0 |
| Karras exponent ($\rho_{\mathrm{Karras}}$) | 7.5 |

# I. Complexity Analysis & Additional Results

| Method | Category | Compute to generate $h$ points | Seq. ops | Main bottleneck |
|---|---|---|---|---|
| *Discrete deterministic* | | | | |
| DLinear | LTSF-Linear | $\mathcal{O}(C\,\ell\,h)$ | $\mathcal{O}(1)$ | One-shot linear map from lookback $\ell$ to horizon $h$ (per-channel). |
| PatchTST | Patch Transformer | $\mathcal{O}(L(P^2d + Pd^2))$ $+ \mathcal{O}(Chd)$ (head) | $\mathcal{O}(1)$ | Patch-level self-attention; quadratic in #patches $P$. |
| *Diffusion-based forecasting / imputation* | | | | |
| mr-Diff | Multi-resolution diffusion | $\mathcal{O}(T_{\text{inf}} \cdot \text{Cost}_{\text{denoise}}(n))$ | $\mathcal{O}(T_{\text{inf}})$ | Iterative reverse diffusion; denoiser dominates per step (non-autoregressive across time). |
| CSDI | Conditional diffusion | $\mathcal{O}\big(T_{\text{inf}}\,L_r\big(n^2d + C^2d +$ $(n{+}C)d^2\big)\big)$ | $\mathcal{O}(T_{\text{inf}})$ | Reverse diffusion with temporal & feature Transformer blocks in each residual layer. |
| TimeGrad | AR diffusion | $\mathcal{O}\big(h \cdot T_{\text{inf}} \cdot \text{Cost}_{\text{score}}\big)$ | $\mathcal{O}(h \cdot T_{\text{inf}})$ | Autoregressive sampling: diffusion chain per forecast time step. |
| *IMTS (irregular-time) models* | | | | |
| mTAN | Time-attention | $\mathcal{O}(h\,\ell\,d)$ | $\mathcal{O}(1)$ | Attention from $h$ query times to $\ell$ irregular observations. |
| T-PATCHGNN | Patch GNN | $\mathcal{O}(L_g \cdot |E| \cdot d)$ | $\mathcal{O}(1)$ | Graph message passing on patch graphs; cost depends on edge density $|E|$. |
| *Continuous-time baselines* | | | | |
| Neural ODE | ODE-Net | $\mathcal{O}(\text{NFE}\,d^2)$ | $\mathcal{O}(\text{NFE})$ | Sequential solver calls; NFE depends on tolerance/dynamics. |
| Latent ODE | ODE-RNN | $\mathcal{O}(\text{NFE}_{\text{tot}}\,d^2)$ | $\mathcal{O}(\text{NFE}_{\text{tot}})$ | Solver-dominated hidden evolution between observations (plus encoder/decoder). |
| GRU-ODE-Bayes | CT-RNN | $\mathcal{O}(\text{NFE}_{\text{tot}}\,d^2)$ $(+ \text{updates})$ | $\mathcal{O}(\text{NFE}_{\text{tot}})$ | ODE evolution + sporadic observation updates. |
| Neural CDE | CDE model | $\mathcal{O}(\text{NFE}\,d^2)$ $(+ \text{interpolation})$ | $\mathcal{O}(\text{NFE})$ | Differential-equation solve driven by interpolated control. |
| Latent / Neural SDE | SDE model | $\mathcal{O}(\text{NFE}\,d^2)$ $(\text{larger const.})$ | $\mathcal{O}(\text{NFE})$ | Stochastic solver + Brownian/noise handling. |
| ContiFormer | CT-Transformer | $\mathcal{O}\big(S\,L\,(n^2d + nd^2)\big)$ | $\mathcal{O}(S)$ | Solver-style sequential steps; each step runs attention over length $n$. |
| **LLapDiff** | **Latent diffusion** | $\mathcal{O}\big(T_{\text{inf}}Kd(K + \ell + h)\big)$ | $\mathcal{O}(T_{\text{inf}})$ | Diffusion loop; per-step modal attention $\sim K^2d$; synthesis is linear $\sim hKd$. |

*Table 11.* **Inference-time complexity** to generate a horizon of length $h$. $C$: #channels; $p$: patch size; $P = \lceil \ell/p \rceil$: #patches; $d$: hidden width; $L$: backbone depth; $L_r$: residual depth (e.g., diffusion block stack); $K$: Laplace modes; $T_{\text{inf}}$: diffusion sampling steps; $n := \ell + h$ (when the model processes past+future jointly); NFE: solver function evaluations; $|E|$: graph edges; $S$: solver-style sequential steps.

*Table 12.* All entries mean (±std) over 10 runs. Tab. 1 in the main body reports means only for readability.

| Dataset | $h$ | DLinear MAE | DLinear MSE | PatchTST MAE | PatchTST MSE | mr-Diff CRPS | mr-Diff MSE | TimeGrad CRPS | TimeGrad MSE | mTAN CRPS | mTAN MSE | T-PATCHGNN MAE | T-PATCHGNN MSE | ContiFormer MAE | ContiFormer MSE | NeuralCDE MAE | NeuralCDE MSE | LLapDiff CRPS | LLapDiff MSE |
|---|---|---|---|---|---|---|---|---|---|---|---|---|---|---|---|---|---|---|---|
| BMS Air | 24 | 1.12±0.04 | 2.70±0.08 | 0.94±0.02 | 1.31±0.05 | 0.58±0.02 | 1.38±0.05 | **0.46**±0.01 | **0.88**±0.02 | 0.48±0.02 | 1.00±0.04 | 0.78±0.02 | 1.14±0.04 | 0.92±0.03 | 1.76±0.06 | 0.88±0.03 | 1.63±0.04 | 0.49±0.02 | 1.18±0.03 |
| BMS Air | 48 | 1.31±0.04 | 3.58±0.12 | 0.94±0.02 | 1.30±0.04 | 0.58±0.02 | 1.38±0.04 | **0.46**±0.01 | **0.91**±0.02 | 0.52±0.02 | 1.19±0.02 | 0.73±0.02 | 1.03±0.03 | 0.89±0.03 | 1.70±0.04 | 0.90±0.04 | 1.72±0.04 | 0.50±0.02 | 1.18±0.02 |
| BMS Air | 96 | 1.43±0.04 | 4.13±0.12 | 0.94±0.02 | 1.30±0.04 | 0.58±0.02 | 1.38±0.04 | 0.51±0.03 | 1.08±0.03 | 0.54±0.02 | 1.29±0.04 | 0.68±0.02 | 0.98±0.03 | 0.95±0.03 | 1.90±0.06 | 1.01±0.03 | 2.22±0.06 | **0.50**±0.01 | 1.20±0.03 |
| BMS Air | 168 | 1.45±0.00 | 4.21±0.00 | 0.93±0.00 | 1.30±0.00 | 0.56±0.02 | 1.33±0.04 | 0.54±0.00 | 1.17±0.00 | 0.55±0.00 | 1.30±0.00 | 0.76±0.00 | 1.10±0.00 | 0.98±0.06 | 2.03±0.06 | 1.02±0.03 | 2.20±0.06 | 0.52±0.00 | 1.25±0.00 |
| UCI Air | 24 | 2.68±0.08 | 9.12±0.28 | 1.10±0.04 | **1.92**±0.05 | 1.09±0.04 | 3.36±0.11 | 1.22±0.02 | 3.71±0.12 | **0.91**±0.02 | 2.55±0.09 | 1.11±0.03 | 1.99±0.06 | 2.08±0.06 | 7.06±0.24 | 1.86±0.06 | 5.67±0.18 | 0.94±0.03 | 2.48±0.07 |
| UCI Air | 48 | 2.64±0.08 | 9.00±0.28 | 1.09±0.04 | **1.98**±0.05 | 0.99±0.03 | 2.81±0.08 | 1.24±0.04 | 3.81±0.12 | **0.83**±0.02 | 2.17±0.06 | 1.13±0.03 | 2.05±0.06 | 2.12±0.06 | 7.34±0.24 | 1.82±0.05 | 5.42±0.16 | 0.94±0.02 | 2.38±0.07 |
| UCI Air | 96 | 2.69±0.08 | 9.51±0.28 | 1.06±0.04 | **1.79**±0.05 | 1.10±0.03 | 3.37±0.11 | 1.20±0.04 | 3.58±0.12 | **0.97**±0.02 | 2.59±0.09 | 1.23±0.04 | 2.23±0.07 | 2.09±0.06 | 7.14±0.21 | 2.05±0.06 | 6.81±0.27 | 0.94±0.02 | 2.33±0.07 |
| UCI Air | 168 | 2.75±0.00 | 9.90±0.00 | 1.15±0.00 | 2.20±0.00 | 1.17±0.03 | 3.81±0.11 | 1.12±0.00 | 3.12±0.01 | **0.84**±0.00 | 2.39±0.00 | 1.12±0.00 | 2.00±0.00 | 2.14±0.06 | 7.45±0.24 | 1.99±0.06 | 6.33±0.18 | 1.00±0.00 | 2.87±0.01 |
| PhysioNet | 4 | 0.46±0.01 | 0.69±0.02 | 0.47±0.01 | 0.72±0.02 | 0.42±0.01 | 0.81±0.02 | 0.43±0.01 | 0.85±0.02 | 0.44±0.01 | 0.84±0.01 | 0.37±0.01 | 0.56±0.02 | 0.41±0.01 | 0.63±0.02 | 0.42±0.01 | 0.64±0.02 | **0.34**±0.01 | 0.65±0.02 |
| PhysioNet | 8 | 0.47±0.01 | 0.70±0.01 | 0.48±0.01 | 0.73±0.02 | 0.43±0.01 | 0.82±0.02 | 0.44±0.01 | 0.86±0.02 | 0.44±0.01 | 0.85±0.01 | 0.40±0.01 | 0.58±0.02 | 0.40±0.01 | 0.63±0.02 | 0.41±0.01 | 0.64±0.02 | **0.34**±0.01 | 0.64±0.02 |
| PhysioNet | 10 | 0.47±0.01 | 0.71±0.02 | 0.48±0.01 | 0.73±0.02 | 0.39±0.01 | 0.78±0.02 | 0.44±0.01 | 0.87±0.02 | 0.45±0.01 | 0.86±0.02 | 0.40±0.01 | 0.59±0.02 | 0.42±0.01 | 0.65±0.02 | 0.45±0.01 | 0.68±0.02 | **0.33**±0.01 | 0.67±0.02 |
| PhysioNet | 12 | 0.48±0.00 | 0.72±0.00 | 0.49±0.00 | 0.74±0.00 | 0.43±0.01 | 0.83±0.02 | 0.45±0.01 | 0.87±0.00 | 0.45±0.00 | 0.87±0.00 | 0.38±0.00 | 0.58±0.00 | 0.42±0.01 | 0.65±0.02 | 0.43±0.01 | 0.66±0.02 | **0.32**±0.00 | 0.64±0.00 |
| NOAA US | 24 | 0.35±0.01 | 0.21±0.01 | 0.32±0.01 | **0.20**±0.01 | 0.37±0.01 | 0.36±0.01 | 0.45±0.01 | 0.54±0.02 | 0.25±0.01 | 0.24±0.01 | 0.36±0.01 | 0.23±0.01 | 0.48±0.02 | 0.35±0.01 | 0.45±0.01 | 0.32±0.01 | 0.43±0.01 | 0.46±0.01 |
| NOAA US | 48 | 0.35±0.01 | 0.21±0.01 | 0.31±0.01 | **0.19**±0.01 | 0.37±0.01 | 0.36±0.01 | 0.45±0.01 | 0.54±0.02 | 0.25±0.01 | 0.24±0.01 | 0.36±0.01 | 0.23±0.01 | 0.50±0.02 | 0.37±0.01 | 0.47±0.01 | 0.34±0.01 | 0.43±0.01 | 0.47±0.01 |
| NOAA US | 96 | 0.35±0.01 | 0.21±0.01 | 0.33±0.01 | **0.21**±0.01 | 0.37±0.01 | 0.36±0.01 | 0.45±0.01 | 0.54±0.01 | 0.24±0.01 | 0.22±0.01 | 0.36±0.01 | 0.23±0.01 | 0.49±0.01 | 0.36±0.01 | 0.51±0.01 | 0.38±0.01 | 0.43±0.01 | 0.46±0.01 |
| NOAA US | 168 | 0.35±0.00 | 0.21±0.00 | 0.33±0.00 | 0.21±0.00 | 0.37±0.01 | 0.36±0.01 | 0.45±0.00 | 0.55±0.00 | 0.25±0.00 | 0.24±0.00 | 0.33±0.00 | 0.20±0.00 | 0.47±0.01 | 0.35±0.01 | 0.51±0.01 | 0.39±0.01 | 0.44±0.00 | 0.45±0.00 |
| NOAA UK | 24 | 1.46±0.04 | 3.22±0.08 | 0.74±0.02 | **0.71**±0.02 | 0.86±0.02 | 1.80±0.05 | 0.65±0.02 | 0.95±0.02 | 0.90±0.03 | 1.89±0.05 | 0.78±0.02 | 0.79±0.02 | 1.33±0.04 | 2.54±0.07 | 1.08±0.03 | 1.81±0.05 | **0.56**±0.01 | 1.06±0.03 |
| NOAA UK | 48 | 1.65±0.05 | 3.98±0.10 | 0.75±0.02 | **0.74**±0.02 | 0.84±0.02 | 1.71±0.04 | 0.63±0.02 | 0.92±0.02 | 0.94±0.03 | 2.01±0.04 | 0.82±0.02 | 0.88±0.02 | 1.28±0.04 | 2.40±0.06 | 1.10±0.03 | 1.86±0.05 | **0.56**±0.01 | 1.08±0.03 |
| NOAA UK | 96 | 1.58±0.02 | 3.65±0.02 | 0.75±0.02 | **0.74**±0.02 | 0.86±0.02 | 1.80±0.05 | 0.64±0.02 | 0.92±0.02 | 0.94±0.03 | 2.02±0.05 | 0.84±0.02 | 0.82±0.02 | 1.33±0.04 | 2.59±0.07 | 1.09±0.03 | 1.83±0.05 | **0.56**±0.01 | 1.10±0.03 |
| NOAA UK | 168 | 1.55±0.00 | 3.51±0.00 | 0.75±0.00 | **0.74**±0.00 | 0.88±0.02 | 1.86±0.05 | 0.64±0.00 | 0.93±0.00 | 0.87±0.00 | 1.80±0.00 | 0.82±0.00 | 0.91±0.00 | 1.35±0.04 | 2.66±0.07 | 1.11±0.03 | 1.89±0.05 | **0.56**±0.00 | 1.09±0.00 |
| US Equity | 5 | 0.57±0.02 | 0.67±0.02 | 0.57±0.02 | 0.66±0.02 | 0.42±0.01 | 0.77±0.02 | 0.43±0.01 | 0.82±0.02 | 0.42±0.01 | 0.76±0.02 | 0.58±0.02 | 0.65±0.02 | 0.57±0.02 | 0.66±0.02 | 0.57±0.02 | 0.66±0.02 | **0.42**±0.01 | 0.71±0.02 |
| US Equity | 20 | 0.57±0.02 | 0.67±0.02 | 0.57±0.02 | 0.66±0.02 | 0.42±0.01 | 0.76±0.02 | 0.42±0.01 | 0.81±0.02 | 0.42±0.01 | 0.76±0.02 | 0.59±0.02 | 0.67±0.02 | 0.56±0.01 | 0.65±0.02 | 0.57±0.02 | 0.66±0.02 | **0.42**±0.01 | 0.71±0.02 |
| US Equity | 60 | 0.57±0.02 | 0.66±0.02 | 0.57±0.01 | 0.64±0.02 | 0.42±0.01 | 0.74±0.02 | 0.42±0.01 | 0.79±0.02 | 0.41±0.01 | 0.73±0.02 | 0.56±0.02 | 0.64±0.02 | 0.56±0.01 | 0.64±0.02 | 0.56±0.01 | 0.66±0.02 | **0.40**±0.01 | 0.68±0.02 |
| US Equity | 100 | 0.57±0.00 | 0.65±0.00 | 0.56±0.00 | 0.64±0.00 | 0.42±0.01 | 0.75±0.02 | 0.42±0.00 | 0.81±0.02 | 0.42±0.00 | 0.74±0.00 | 0.56±0.00 | 0.64±0.00 | 0.56±0.00 | 0.65±0.02 | 0.56±0.01 | 0.67±0.02 | **0.41**±0.00 | 0.69±0.00 |
| Cryptos | 5 | 0.50±0.01 | 0.60±0.02 | 0.48±0.01 | 0.57±0.02 | 0.37±0.01 | 0.66±0.02 | **0.35**±0.01 | 0.65±0.02 | 0.38±0.01 | 0.70±0.03 | 0.42±0.01 | 0.49±0.02 | 0.46±0.01 | 0.54±0.02 | 0.46±0.01 | 0.54±0.02 | 0.37±0.01 | 0.64±0.01 |
| Cryptos | 20 | 0.49±0.01 | 0.58±0.02 | 0.47±0.01 | 0.54±0.01 | 0.37±0.01 | 0.63±0.01 | **0.34**±0.01 | 0.62±0.01 | 0.36±0.01 | 0.66±0.01 | 0.44±0.01 | 0.51±0.01 | 0.45±0.01 | 0.51±0.01 | 0.45±0.01 | 0.51±0.01 | **0.36**±0.01 | 0.62±0.01 |
| Cryptos | 60 | 0.47±0.01 | 0.54±0.01 | 0.47±0.01 | 0.51±0.01 | 0.38±0.01 | 0.71±0.02 | **0.33**±0.01 | 0.58±0.01 | 0.35±0.01 | 0.60±0.01 | 0.39±0.01 | 0.41±0.01 | 0.44±0.01 | 0.47±0.01 | 0.46±0.01 | 0.47±0.01 | **0.35**±0.01 | 0.58±0.01 |
| Cryptos | 100 | 0.47±0.00 | 0.53±0.00 | 0.47±0.00 | 0.52±0.00 | 0.36±0.01 | 0.61±0.01 | 0.36±0.00 | 0.62±0.00 | 0.36±0.00 | 0.63±0.00 | 0.46±0.00 | 0.52±0.00 | 0.44±0.01 | 0.47±0.01 | 0.45±0.01 | 0.49±0.01 | **0.35**±0.00 | 0.56±0.00 |
| Avg. rank | | 7.89±2.01 | 6.25±2.54 | 5.82±2.54 | 2.96±1.94 | 4.07±1.13 | 6.54±1.15 | 3.82±2.16 | 6.29±2.86 | 3.32±2.07 | 5.86±2.15 | 4.68±1.79 | 1.93±0.96 | 6.57±1.76 | 5.04±2.72 | 6.75±1.33 | 5.21±2.02 | 2.07±1.71 | 4.93±1.81 |

