# OpenReview forum: "Latent Laplace Diffusion for Irregular Multivariate Time Series"
_ICML.cc/2026/Conference — ICML 2026 spotlight_

### Official Review · Reviewer_d2ek · 2026-03-05

**Soundness:** 2
**Presentation:** 3
**Significance:** 2
**Originality:** 3
**Overall Recommendation:** 4
**Confidence:** 3

**Summary:**

This paper proposes LLapDiff, a diffusion-based framework for irregular multivariate time series forecasting. LLapDiff performs conditional diffusion in a low-dimensional latent trajectory space. The denoiser is parameterized by (1) a modal predictor that outputs history-conditioned continuous-time modal parameters and (2) a modal synthesizer that reconstructs the whole horizon at arbitrary query timestamps, avoiding time-stepping over physical time. The design is motivated by a stochastic port-Hamiltonian energy-balance argument for stability bias and by a renewal-averaging analysis linking sampling gaps to effective event-domain poles. Experiments on seven real-world datasets show improvements over baseline methods.

**Compliance With Llm Reviewing Policy:**

Affirmed.

**Key Questions For Authors:**

1. How sensitive are results to the VAE and summarizer pretraining quality?

2. How does performance vs. speed trade off as you vary complex-conjugate poles $K$ and the number of sampling steps?

**Limitations:**

None.

**Strengths And Weaknesses:**

Strengths:

- This paper introduces a new latent diffusion framework for irregular multivariate time series, and establishes a Laplace-domain modal parameterization inspired by stochastic port-Hamiltonian dynamics.
- The effectiveness of the proposed method is evaluated on real-world benchmarks.

Weaknesses:

- It is unclear how much improvement comes from the modal/Laplace bias versus the pretrained VAE.
- The baseline methods, with the latest being from 2024, are not the most recent. It would be beneficial to include more up-to-date SOTA baselines.
- There is no comparison with the runtime of the baseline methods, which is important in the field of time series forecasting.

---

> ### Author Rebuttal · Authors · 2026-03-27
>
> We thank the reviewer for the constructive feedback. Below, we directly address the questions raised.
>
> **1. Sensitivity to VAE and summarizer pretraining quality**
>
> **Direct answer.** Pretraining helps, but gains are not mainly due to a pretrained VAE/summarizer alone, and the model performance is not brittle to one specific latent width.
>
> **Evidence.** The VAE and history summarizer are pretrained on the training split only, then frozen for diffusion training. Table 2 separates pretraining from the latent modal dynamics: under the same pretrained pipeline, removing learned poles degrades NOAA-US by **+0.87 CRPS**, removing the latent trajectory by **+0.59**, and jointly training the summarizer is also worse (**+0.92**). Table 3 further shows that summarizer is not just a generic encoder: among its tokens, the **temporal token** is the most important, consistent with our claim that gap-aware conditioning matters most under irregular sampling. Therefore, the gains are not explained by pretraining alone; the modal parameterization and conditioning each contribute substantially.
>
> We also include additional latent-width sweeps on two representative nonstationary datasets ($h=168, K=256$): **NOAA-US** (irregular) and **BMS Air** (near-regular).
>
> **Latent-width sensitivity (CRPS)**
>
> **NOAA-US:** ch=8: 0.9990; 16: 0.9693; 24: 0.9699; 32: 0.9847; 64: 0.9753
>
> **BMS Air:** ch=8: 0.6507; 16: 0.6366; 24: 0.6386; 32: 0.6602; 64: 0.6967
>
> So a learned latent space is beneficial, but the performance is stable across a moderate width range rather than depending on one fragile bottleneck. More broadly, as a latent model, LLapDiff would naturally suffer if the latent manifold were badly captured, but results above, together with Table 2 and Table 3, show that the observed gains are not simply inherited from pretraining quality.
>
> **2. Performance-speed tradeoff as K and sampling steps vary**
>
> **Direct answer.** LLapDiff is no one-shot forecaster, and its inference time is nearly flat as $K$ increases; the dominant cost is iterative DDIM denoising, whose runtime scales approximately linearly with the number of DDIM steps.
>
> **Evidence.** When sweeping K from 16 to 512, inference stays around **452–457 ms** on both NOAA-US and BMS Air across all tested horizons, so increasing modal capacity adds very little overhead. Representative **inference runtime** is **449.16–451.17 ms** for LLapDiff on NOAA-US and **449.25–450.93 ms** on BMS Air, versus **553.41–560.05 / 554.52–558.88 ms** for NeuralCDE, **885.52–890.10 / 879.95–882.18 ms** for MRDiff, **5107.12–35773.63 / 5043.82–35170.37 ms** for TimeGrad. Thus, it is materially faster than heavier generative/solver-style baselines while avoiding physical-time numerical integration. This is roughly **1.2x faster than NeuralCDE**, **about 2x faster than MRDiff**, and **substantially faster** than TimeGrad / **ContiFormer (4x slower than TimeGrad)**, especially long horizons. Below, we include the tradeoff for pole number and DDIM steps.
>
> **Varying pole inference time (ms; columns = horizon 24/48/96/168)**
>
> **NOAA-US**
> K=16: 453.31 / 454.49 / 454.45 / 454.30
> K=32: 451.85 / 452.49 / 451.86 / 454.04
> K=64: 456.59 / 453.14 / 454.41 / 454.79
> K=128: 456.10 / 455.42 / 454.88 / 455.13
> K=256: 454.74 / 453.80 / 453.95 / 454.66
> K=512: 456.55 / 454.28 / 454.18 / 454.97
>
> **BMS Air**
> K=16: 452.16 / 452.79 / 452.58 / 452.82
> K=32: 453.63 / 452.49 / 453.09 / 454.11
> K=64: 454.84 / 454.74 / 455.66 / 456.85
> K=128: 455.40 / 454.82 / 455.16 / 455.83
> K=256: 455.14 / 454.38 / 454.08 / 454.70
> K=512: 456.21 / 456.07 / 454.41 / 455.96
>
> **Longest-horizon CRPS vs. K**
>
> **NOAA-US:** 1.1890 / 0.9971 / 0.9947 / 0.9847 / 0.9685 / 0.9747 for $K=16/32/64/128/256/512$
>
> **BMS Air:** 0.7299 / 0.6952 / 0.6596 / 0.6518 / 0.6368 / 0.6594 for $K=16/32/64/128/256/512$
>
> So there is a clear sweet spot around $K \approx 256$: performance improves substantially from small $K$ to moderate $K$, then saturates or slightly regresses, while runtime stays almost unchanged.
>
> **DDIM step sweep (CRPS / median ms)**
>
> **NOAA-US:** 16: 0.9839 / 151.69; 32: 0.9765 / 257.54; 64: 0.9685 / 456.32; 128: 0.9697 / 802.21
>
> **BMS Air:** 16: 0.6539 / 156.86; 32: 0.6461 / 261.86; 64: 0.6361 / 446.21; 128: 0.6374 / 793.88
>
> The pattern is clean: runtime scales almost linearly with the number of DDIM steps, while downstream quality changes only modestly. This directly answers the efficiency question: the main runtime bottleneck is **step count, not pole count**, and the best operating point in our setup is around $K$=256, 64 DDIM steps as provided in Appx. H.
>
> Overall, these results indicate: (i) LLapDiff’s gains are not mainly due to pretrained components alone, and (ii) its inference cost is driven mainly by DDIM step count rather than modal size $K$. We appreciate this suggestion and agree that including newer baselines would strengthen the paper; in the current version, we focused on the key methodological families for this setting.

---

> > ### Author Rebuttal · Reviewer_d2ek · 2026-04-01
> >
> > Thank you for your reply. I have raised the score to 4.

---

> > > ### Author Response · Authors · 2026-04-06
> > >
> > > Thank you for the acknowledgement and for raising your score. We appreciate your thoughtful comments and are glad our response addressed your concerns.

---

### Official Review · Reviewer_emF6 · 2026-03-06

**Soundness:** 3
**Presentation:** 3
**Significance:** 3
**Originality:** 4
**Overall Recommendation:** 4
**Confidence:** 4

**Summary:**

Focusing on irregular time series forecasting, the authors propose to model the target as a low-dimensional latent trajectory, enabling horizon-wide generation without step-by-step integration over physical time. A stable modal parameterization, inspired by stochastic port-Hamiltonian dynamics, is used to guide the reverse process. The theoretical evidence and empirical validation seem to be solid.

**Compliance With Llm Reviewing Policy:**

Affirmed.

**Final Justification:**

At this stage, I will keep the original score for now.

**Key Questions For Authors:**

(1) The notation used in the equations is too complex. More importantly, how is the mapping from $z_0$ to $x_0$ performed in the reference stage?

(2) In Table 1, why are different metrics used across different datasets?

(3) The following statement is not objective: "However, most time-series diffusion methods denoise directly in the observation space and handle irregularity mainly through masks and time embeddings." The related work and experimental evaluation should be improved accordingly.

(4) A more precise definition of irregular time series is expected, including an analysis of the datasets used.

(5) The generalization of the proposed approach needs to be analyzed more deeply. The method models mean evolution via learnable complex-conjugate poles—are these poles optimized during training and then held constant at test time, or do they adapt in some way to different input samples? If they are indeed fixed after training, this design would implicitly assume that any test-time dynamics can be expressed as a linear combination of these pre-learned, globally shared modes. How would the model then cope with real-world irregular time series that often exhibit dynamic regime shifts, introducing novel spectral patterns not encountered during training? Since the history summarizer can only re-weight existing modes rather than generate new poles, can the model reliably extrapolate to out-of-distribution dynamics? I would invite the authors to clarify the nature of these poles and, if they are indeed fixed, provide a deeper discussion on this potential limitation. Additional experiments on tasks with clear regime changes—such as synthetic data with controlled frequency/decay shifts or real-world non-stationary datasets—would help empirically assess the model's robustness and clarify the conditions under which the fixed-pole design remains sufficient or becomes a generalization bottleneck.

**Limitations:**

yes

**Strengths And Weaknesses:**

Strengths:
(1) The idea addresses a key gap in irregular time series forecasting: discrete methods can distort temporal structure via re-gridding, while continuous-time models often rely on sequential numerical solvers that are prone to drift.
(2) The submission is technically sound and provides theoretical evidence.
(3) The submission is well-written and organized.

Weaknesses:
(1) The generalization of the proposed approach needs to be analyzed more deeply.
(2) Existing Laplace-based or latent space diffusion-based approaches should be either analyzed or compared with the proposed framework.
(3) The source code is not publicly available, and there is no claim to open it in the future.

---

> ### Author Rebuttal · Authors · 2026-03-27
>
> We thank the reviewer for the careful reading and constructive questions. An anonymized code repository is linked in the submission (line 312, above Sec. 6). We address the 5 points below.
>
> **1. Notation complexity / mapping from $z_0$ to $x_0$**
>
> **Direct answer.** Here, $x_t$ is an auxiliary state used only in the Sec. 4 derivation. In that derivation, we identify $x_t \equiv z_0(t)$ conceptually, but operationally, inference predicts $\hat z_0$ directly and decodes it; there is no separate $z_0 \to x_0$ stage.
>
> **Evidence.** In Sec. 5.2, the modal synthesizer reconstructs $\hat z_0(\tilde t)$ from the predicted modal parameters, and the final output is $\hat{\mathcal Y}=\text{VAE}_{dec}(\hat z_0)$. Thus, the practical path is $z_T \rightarrow \cdots \rightarrow \hat z_0 \rightarrow \hat{\mathcal Y}$, whereas $x_t$ is introduced only to motivate the stable modal parameterization.
>
> **2. Why are different metrics used in Table 1**
>
> **Direct answer.** The metrics differ across model types, not datasets. The baselines include both probabilistic and deterministic methods, so we report CRPS for probabilistic models, MAE for deterministic models (equivalently, the CRPS of a Dirac predictor), and MSE as a shared point metric for all methods.
>
> **Evidence.** The evaluation target is unchanged across datasets; only the prediction type differs. Table 1 therefore uses CRPS for predictive distributions, MAE for point forecasts, and MSE as a common point metric. We will make this distinction explicit in the final paper.
>
> **3. Prior diffusion methods / broad wording**
>
> **Direct answer.** We agree that the original wording is quite general. Our intended claim is concentrated: in our forecasting/imputation setting, the compared diffusion baselines mainly denoise in observation space and handle irregularity through masks, timestamps, or similar conditioning, rather than through an explicit continuous-time modal parameterization.
>
> **Evidence.** The intended distinction in Sec. 2 is among observation-space diffusion, latent-space diffusion, and continuous-time / Laplace-parameterized approaches. Our contribution is to combine latent trajectory diffusion with stable Laplace-domain modal synthesis that can be queried at irregular timestamps without physical-time numerical integration. We will revise Sec. 2 and clarify this positioning relative to prior latent-space / Laplace-based work.
>
> **4. More precise definition of irregular time series and dataset analysis**
>
> **Direct answer.** In our setting, irregularity has two forms: I) **temporal irregularity**: nonuniform or gapped observation/query times; II) **value irregularity**: asynchronous channel/entity missingness even on a native timestamp grid.
>
> **Evidence.** Our dataset audit shows these are distinct axes. For example, BMS Air has genuine temporal gaps yet is labeled stationary; US Equity has the largest gap fraction and is also stationary; Crypto has almost no temporal gaps but noticeable value missingness and is also stationary. By contrast, NOAA-US, NOAA-UK, UCI Air, and PhysioNet are comparatively cleaner in the cached benchmark gap/missingness statistics but are labeled non-stationary. Therefore, some datasets are irregular yet stationary, while others are relatively clean but non-stationary. Extra gap and missingness can also be induced by minimum-coverage filtering, as in the stress tests in Table 4. We will make this distinction more explicit in the main text and Appx. D.
>
> **5. Are the poles fixed after training, and what does this imply for generalization**
>
> **Direct answer.** The poles are not fixed globally at test time. LLapDiff uses learned base poles plus history-conditioned bounded perturbations, so the realized poles adapt to each input window and each denoising step while remaining stable by construction.
>
> **Evidence.** In Appx. F.1, the model maintains learned base poles and predicts bounded perturbations from a conditioning vector formed by diffusion-step embedding and pooled history summary. The poles are then
> $\hat \rho_k=\text{softplus}(\rho_k+\Delta \rho_k)+\rho_{min}$ and
> $\hat \omega_k=\omega_{max}\text{sigmoid}(\mathrm{logit}(\omega_k/\omega_{max})+\Delta \omega_k)$.
> Therefore, the history summarizer does more than re-weight fixed modes; through bounded perturbations, it also shifts pole locations per sample and per denoising step, though within a finite $K$-slot parameterization.
>
> To probe regime shifts, we test synthetic damped sinusoids with an abrupt change point at $t=216$. In a regime-crossing setting (context ends in $[204,215]$), frequency doubling changes CRPS from 0.3065 to 0.3376 and MAE from 0.3879 to 0.4217; increasing the decay rate by $2.5\times$ changes CRPS from 0.2482 to 0.2890 and MAE from 0.3166 to 0.3621. This suggests that the adaptive pole parameterization remains reasonably robust under abrupt spectral shifts, while degrading mildly under harder out-of-distribution regime changes. We will include these results in Appx. I.

---

> > ### Author Rebuttal · Reviewer_emF6 · 2026-04-03
> >
> > The authors have addressed most of my concerns. Regarding point 5 in the rebuttal, could the authors provide a more thorough experimental analysis?

---

> > > ### Author Response · Authors · 2026-04-05
> > >
> > > Thank you for the follow-up. We appreciate that you found our rebuttal had addressed most of your concerns, and we agree that point 5 deserved a more thorough experimental analysis. We therefore ran a stricter set of regime-shift experiments targeted at this question.
> > >
> > > **Direct answer.**
> > >
> > > Under our original synthetic setup, the regime-shift experiment should be interpreted as a **boundary-crossing robustness** test rather than a strict unseen-regime generalization test, since some training windows already cross the change point. To address your question directly, we then introduced a **strict unseen-regime protocol** in which no training or validation window crosses the change point and the shifted regime is encountered at test time. We also compared full LLapDiff against an otherwise identical **fixed-pole ablation** that disables only the history-conditioned pole perturbation. This isolates the mechanism in question: whether adaptive perturbations to the learned base poles materially help under regime shift.
> > >
> > > **Evidence and analysis.**
> > >
> > > (1) **Original setup = boundary-crossing robustness, not strict OOD.** Even in that setting, the degradation is only moderate under regime changes: for frequency doubling, CRPS/MAE increase from `0.3065/0.3879` to `0.3376/0.4217` (`+10.1%/+8.7%`), and for a `2.5x` increase in decay, CRPS/MAE increase from `0.2482/0.3166` to `0.2890/0.3621` (`+16.4%/+14.4%`).
> > >
> > > (2) **Stricter unseen-regime protocol.** We kept the LLapDiff recipe unchanged (same architecture/training setup, no retuning; context window `96`, horizon `48`, latent channels `24`, Laplace basis size `256`, over three seeds) and changed only the regime protocol: `series_length=432`, `change_point=373`, with `0/0/48` train/val/test crossing windows per asset and `11` post-shift-context test windows per asset. In the fixed-pole ablation, only the pole perturbation update is disabled; the summarizer, backbone, VAE, and sampling settings remain the same.
> > >
> > > (3) **Representative strict-OOD results at cited severities.**
> > >
> > > | Task (shift) | Model | Crossing CRPS | Post-shift CRPS | Crossing MAE | Post-shift MAE |
> > > |---|---:|---:|---:|---:|---:|
> > > | freq shift (`2.0x`) | adaptive | `0.2014` | `0.1812` | `0.2589` | `0.2366` |
> > > | freq shift (`2.0x`) | fixed | `0.2025` | `0.1831` | `0.2593` | `0.2371` |
> > > | decay shift (`2.5x`) | adaptive | `0.1940` | `0.1548` | `0.2505` | `0.2071` |
> > > | decay shift (`2.5x`) | fixed | `0.1937` | `0.1558` | `0.2524` | `0.2096` |
> > >
> > > These results indicate that adaptive poles **modestly but consistently** match or outperform fixed poles at the originally cited severities. For frequency shift, adaptive is better on all reported metrics. For decay shift, crossing CRPS is essentially tied (`0.1937` fixed vs. `0.1940` adaptive), but adaptive improves crossing MAE and both post-shift metrics.
> > >
> > > (4) **Additional analysis beyond the representative table.** We also ran a severity sweep under the same strict protocol. For frequency shift, we evaluated multipliers `{1.25x, 1.5x, 2.0x, 2.5x}`; for decay shift, `{1.5x, 2.0x, 2.5x, 3.0x}`. The results show no catastrophic deterioration across the tested severities: for example, frequency-shift crossing CRPS changes only from `0.1991` at `1.25x` to `0.2024` at `2.5x`. We further examined performance as a function of distance to the change point and did not observe a sharp error spike at the boundary; for example, for frequency shift, CRPS changes from `0.2164` well before the change point to `0.1816` after the shift, and for decay shift from `0.2151` to `0.1558`. Finally, pole diagnostics show nonzero test-time pole motion in the adaptive model, whereas the fixed-pole ablation has effectively no pole movement. This provides mechanism evidence: LLapDiff does not rely on globally fixed poles at inference, but combines learned base poles with bounded history-conditioned adjustments.
> > >
> > > **Key takeaways.**
> > > - The earlier synthetic result is still useful, but it should be framed as **boundary-crossing robustness**, not the main unseen-regime claim.
> > > - Under a stricter test-only regime-shift protocol, the stronger evidence favors **adaptive poles over fixed poles**.
> > > - The gain is **small but consistent**, especially on post-shift metrics, so the right statement is improved robustness rather than a large absolute margin.
> > > - The distance-to-change and severity-sweep analyses show **graceful behavior near and after the change point**, not catastrophic failure.
> > > - The pole-diagnostic analysis is important because it shows that the benefit is tied to the intended mechanism: **history-conditioned perturbations to base poles**.
> > >
> > > Overall, these additional experiments support a more precise conclusion: LLapDiff does not use globally fixed poles at test time; instead, bounded history-conditioned perturbations of learned base poles provide a **small but consistent robustness benefit** under unseen regime changes, while not implying unlimited extrapolation to arbitrary OOD dynamics.

---

### Official Review · Reviewer_k8ZN · 2026-03-11

**Soundness:** 3
**Presentation:** 3
**Significance:** 2
**Originality:** 2
**Overall Recommendation:** 5
**Confidence:** 4

**Summary:**

The paper presents Latent Laplace Diffusion (LLapDiff), a generative framework designed to address the challenges of forecasting and imputing irregular multivariate time series. The authors propose modeling target sequences as low-dimensional latent trajectories to avoid the temporal distortions inherent in re-gridding or the error accumulation common in continuous-time models that require sequential numerical integration. The methodology incorporates a stable modal parameterization derived from stochastic port-Hamiltonian dynamics, representing the mean evolution of the latent state in the Laplace domain using learnable complex-conjugate poles. This approach allows for direct evaluation at any irregular timestamp without step-by-step integration. Furthermore, the framework introduces a gap-aware history summarizer, theoretically motivated by renewal-averaging analysis, to condition the diffusion process on sampling irregularities. The performance of LLapDiff is evaluated against various deterministic, diffusion-based, and continuous-time baselines across several datasets, demonstrating improved accuracy in long-horizon forecasting and effective zero-shot imputation.

**Compliance With Llm Reviewing Policy:**

Affirmed.

**Final Justification:**

Thank you for your reply. I have raised the score to 5.

**Key Questions For Authors:**

First remark I want to emphasis on the basics of the methodology: Regarding the VAE pretraining, could the authors clarify the impact of latent space dimensionality on the stability of the diffusion process, and whether an end-to-end training approach was considered to minimize information loss? The local linearization of the Hamiltonian dynamics assumes a stable equilibrium point; how does the model respond to highly non-stationary data where the Jacobian-equivalent representation might shift rapidly within a single context window? Furthermore, could the authors provide a more detailed comparison of the total training time and inference latency for LLapDiff relative to single-pass Transformer models, given the iterative nature of the reverse diffusion process? Finally, is there a specific mechanism to ensure that the learned complex-conjugate poles remain within the stable region of the complex plane during the entire training process?

**Limitations:**

The primary technical limitation of LLapDiff is its reliance on a linear modal parameterization to represent mean latent dynamics, which may prove insufficient for systems with strong non-linear coupling. The framework’s performance is also highly sensitive to the pretraining of the latent manifold, and failures in the VAE stage can propagate directly to the diffusion model. Computationally, although the model avoids numerical ODE integration, it still requires multiple denoising steps during inference, which maintains a high computational cost compared to deterministic one-shot forecasters. The theoretical motivation for the gap-aware summarizer assumes a certain degree of statistical regularity in the sampling process (renewal process), which may not hold for all types of informatively missing data. Lastly, the current evaluation demonstrates that the specialized continuous-time architecture provides marginal benefits in near-regular datasets, limiting its general-purpose applicability.

**Strengths And Weaknesses:**

The first originality of this work is the paper provides a theoretically grounded approach to irregular time series by integrating principles from physics-informed modeling and generative diffusion. The use of port-Hamiltonian dynamics to justify a stable, dissipative mean trajectory offers a clear inductive bias that distinguishes this work from standard diffusion models that operate directly in the observation space (I found this very interesting). By performing denoising on a latent manifold, the framework effectively decouples the complexity of high-dimensional irregular observations from the underlying temporal dynamics. The methodological rigor is further supported by the renewal-averaging analysis, which provides a mathematical link between random sampling gaps and the effective poles of the system. This theoretical contribution offers a rational basis for the design of the conditioning mechanism, moving beyond heuristic time-embeddings typical in the literature.

However, the architecture's reliance on a two-stage process—pretraining a VAE followed by training a diffusion model on the latent space—introduces significant dependencies that are not fully explored (I think we should see more in depth-treatment). The quality of the latent representation acts as a bottleneck for the entire system, and the paper lacks an extensive analysis of how variations in the VAE’s reconstruction fidelity or latent dimensionality impact the final forecasting accuracy. While the "solver-free" evaluation is a technical advantage, it is important to note that the mean dynamics are essentially restricted to a linear modal parameterization through local linearization. This assumption may limit the model's ability to capture complex, non-linear transitions or structural breaks that cannot be represented by a sum of damped sinusoids in the Laplace domain. Additionally, the empirical gains are noticeably concentrated in highly irregular or sparse regimes, with the authors acknowledging that the benefits of the continuous-time structure diminish as the data density increases toward a regular grid. This suggests that the framework’s complexity may not be justified for a large subset of practical time-series applications.

---

> ### Author Rebuttal · Authors · 2026-03-27
>
> We thank the reviewer for the thoughtful review. We appreciate your interest in the stochastic port-Hamiltonian and renewal-averaging view. We address your questions directly.
>
> **1. Latent dimensionality / end-to-end training**
>
> **Direct answer.** LLapDiff benefits from pretraining, but is not brittle to the exact latent width tested. End-to-end training was considered, and we intentionally freeze the pretrained VAE/summarizer to stabilize the latent geometry used by diffusion under sparse, irregular supervision.
>
> **Evidence.** The VAE and history summarizer are pretrained on the training split and then frozen during diffusion training. This deliberately avoids coupling representation learning with diffusion denoising under sparse, irregular supervision. Table 2 separates pretraining from our modal bias and conditioning: on NOAA-US, removing the latent space degrades CRPS from 0.953 to 1.539, while removing learned poles or conditioning degrades further to 1.818 and 1.961, respectively; jointly training the summarizer also performs worse (1.868). We also tested latent-width sweeps on representative cases (NOAA-US and BMS Air), showing modest sensitivity rather than brittleness to one exact dimension. Empirically, training remained stable across the tested widths.
>
> **Latent-channel sensitivity, $h=168$, $K=256$**
>
> **NOAA-US**
> ch=8 CRPS=0.9990034
> ch=16 CRPS=0.9693359
> ch=24 CRPS=0.9698674
> ch=32 CRPS=0.9846532
> ch=64 CRPS=0.9752838
>
> **BMS Air**
> ch=8 CRPS=0.650650
> ch=16 CRPS=0.636598
> ch=24 CRPS=0.638643
> ch=32 CRPS=0.660181
> ch=64 CRPS=0.696674
>
> **2. Highly non-stationary data / rapidly shifting local Jacobian**
>
> **Direct answer.** The model does not use one fixed global linearization; it predicts a history-conditioned, window-specific Jacobian-equivalent modal representation. This matters empirically because the benchmark includes several non-stationary datasets (e.g., NOAA-US/UK, UCI Air, PhysioNet).
>
> **Evidence.** In realization, poles $(\rho,\omega)$ are conditioned on the history summary via global base poles plus history-conditioned perturbations. Concretely, perturbations are predicted from a conditioning vector formed by the diffusion-step embedding and pooled history summary, so the realized poles are recomputed for each input window and denoising step rather than fixed globally. Thus, the shared part is only parameterization and finite number $K$ of modal slots; the realized modal dynamics vary with the observed history. Extremely abrupt within-window regime changes can still be challenging, but the model is designed to adapt across contexts rather than reuse one fixed spectrum.
>
> **3. Runtime vs. single-pass Transformers**
>
> **Direct answer.** LLapDiff is not a one-shot forecaster, so it is not faster than single-pass models such as DLinear or PatchTST on raw latency. In practice, its main tradeoff is not versus deterministic one-pass predictors, but versus heavier continuous-time / generative methods. Our efficiency claim is narrower: LLapDiff avoids physical-time numerical integration.
>
> **Evidence.** In wall-clock inference, LLapDiff is around 449–451 ms, whereas DLinear and PatchTST are much faster (roughly 0.3–2.3 ms), as expected for single-pass predictors. However, LLapDiff is faster than heavier baselines such as NeuralCDE (553–560 ms), MRDiff (880–890 ms), and substantially faster than TimeGrad / ContiFormer in the same setup. Representative inference time (ms, NOAA-US; horizons 24/48/96/168): LLapDiff 449/449/451/451; NeuralCDE 553/560/560/554; MRDiff 890/888/886/886; TimeGrad 5107/10226/20452/35774. Training is also heavier than single-pass baselines because LLapDiff uses a two-stage protocol (~0.6–1.7 s per epoch for the diffusion stage in our setup). Accordingly, our claim is not lower total cost than single-pass Transformers, but a favorable tradeoff relative to stronger continuous-time / diffusion baselines when irregular long-horizon generation is required. The complexity analysis in Appx. I reflects this: single-pass models have $\mathcal{O}(1)$ sequential generation, while LLapDiff has $\mathcal{O}(T_{\text{inf}})$ due to diffusion loop. We also note in Sec. 6.2, relative gains are smaller in dense, near-regular regimes, while they grow higher in irregular and long-horizon scenarios. We will include these wall-clock times in the Appx. I.
>
> **4. Stable-region guarantee for learned poles**
>
> **Direct answer.** Yes, stability is enforced by construction via constrained positive damping.
>
> **Evidence.** The modal parameterization uses complex-conjugate poles of the form $-\rho_k \pm i\omega_k$ with $\hat \rho_k=\text{softplus}(\rho_k+\Delta \rho_k)+\rho_{min}$. Therefore, $\hat \rho_k>0$ throughout training and inference. The bounded-perturbation parameterization allows sample-adaptive pole movement while keeping all realized poles in the stable region. Empirically, Fig. 4 shows that all learned poles satisfy $\rho>0$, with substantial mass away from the instability boundary.

---

> > ### Author Rebuttal · Reviewer_k8ZN · 2026-04-02
> >
> > Thank you for your reply. I have raised the score to 5.

---

> > > ### Author Response · Authors · 2026-04-06
> > >
> > > Thank you for the acknowledgement and for raising the score. We appreciate your careful reading and are glad our response addressed your concerns.

---

### Official Review · Reviewer_umN8 · 2026-03-11

**Soundness:** 4
**Presentation:** 3
**Significance:** 3
**Originality:** 4
**Overall Recommendation:** 6
**Confidence:** 3

**Summary:**

This paper proposes Latent Laplace Diffusion, a generative framework for modeling irregular multivariate time series. The method performs diffusion in a latent trajectory space obtained from a pretrained VAE and parameterizes denoising dynamics through Laplace-domain modal components motivated by stochastic port-Hamiltonian systems. This formulation allows trajectory generation over irregular timestamps without numerical integration in physical time. A gap-aware history summarizer is introduced to capture irregular sampling patterns. Experiments on several real-world datasets demonstrate improved long-horizon forecasting performance compared with multiple baselines.

Overall, the paper presents an interesting and technically sophisticated approach to irregular time-series modeling. The integration of latent diffusion with Laplace-domain modal dynamics is novel and empirically promising, particularly as an alternative to solver-based continuous-time models. However, the framework introduces considerable complexity and relies on several modeling assumptions whose practical robustness is not fully explored.

**Compliance With Llm Reviewing Policy:**

Affirmed.

**Key Questions For Authors:**

1 How sensitive is the model performance to the quality and dimensionality of the pretrained VAE latent space?
2 How does inference time compare with NeuralODE / CDE models and other diffusion-based time-series models?
3 Since the method relies on local linear modal dynamics, how well does it perform on strongly nonlinear or chaotic systems?
4 The renewal averaging derivation assumes i.i.d. sampling gaps. How robust is the method when sampling patterns are strongly nonstationary or state-dependent?

**Limitations:**

yes

**Strengths And Weaknesses:**

Strengths
Soundness: The paper presents a technically motivated framework that integrates latent diffusion modeling with structured continuous-time dynamics. The stability constraint via damped modal poles is well motivated, and the renewal-averaging perspective provides theoretical intuition for handling irregular sampling gaps. The experimental evaluation is fairly comprehensive, including multiple datasets, ablation studies, and robustness tests under missingness.
Presentation :The paper is generally well structured and clearly explains the motivation for avoiding numerical integration used in continuous-time models such as Neural ODE/CDE approaches. The architecture and experimental design are described in reasonable detail, and the inclusion of visualizations and ablation results helps interpret the contributions of different components.
Significance: Modeling irregular multivariate time series is an important problem in many domains. The proposed approach provides a potentially useful alternative to solver-based continuous-time models by enabling horizon-wide trajectory generation without time stepping. The empirical improvements on several datasets suggest the approach may be beneficial in long-horizon and highly irregular settings.
Originality: The work combines several ideas in a novel way for irregular time-series forecasting. The modal reconstruction of latent trajectories within a diffusion framework appears to be a distinctive contribution relative to existing diffusion or neural ODE approaches.
Weaknesses
Soundness: The theoretical framework relies on several modeling assumptions, including local linearization of the dynamics and the use of pretrained latent representations. The authors acknowledge these limitations, noting that the approach may be less effective when the latent manifold is poorly captured or when system dynamics are strongly nonlinear. While this transparency is appreciated, it would be helpful to better understand empirically how sensitive the method is to these assumptions.
Presentation: Some technical sections, particularly the derivations involving stochastic port-Hamiltonian dynamics and Laplace-domain parameterization, are quite dense and may be difficult to follow for readers without background in dynamical systems. Additional intuition or simplified explanations could improve accessibility.
Significance: Although the proposed framework avoids numerical integration required by some continuous-time models, inference still requires iterative diffusion sampling. The paper does not provide a clear comparison of computational cost or runtime with existing approaches, making it difficult to assess the practical efficiency benefits.
Originality: The combination of latent diffusion, modal parameterization, and physics-inspired stability constraints is interesting and novel in this context. However, several individual components build on existing ideas, and the paper could more clearly highlight which elements constitute the primary conceptual contribution.

---

> ### Author Rebuttal · Authors · 2026-03-27
>
> We thank the reviewer for the kind review and for finding the technical development of the paper solid. We address the 4 questions below.
>
> **1. Sensitivity to pretrained VAE latent quality / dimensionality**
>
> **Direct answer.** Pretraining helps, but not pretraining alone; the gains come from latent diffusion, stable modal parameterization, and gap-aware conditioning, and its performance is not brittle to the specific latent width.
>
> **Evidence.** The VAE and history summarizer are pretrained on the training split and then frozen. Table 2 separates the effects of latent space, learned poles, and conditioning: on NOAA-US, removing the latent space degrades CRPS from 0.953 to 1.539, while removing learned poles or conditioning degrades further to 1.818 and 1.961, respectively; jointly training the summarizer also performs worse (1.868). We also tested latent-width sweeps on NOAA-US and BMS Air, showing benefit from a learned latent space without brittleness to one exact dimensionality. We also noted in Sec.7 that the remaining limitation of these latent models is that performance can degrade if the latent manifold is poorly captured.
>
> **Latent-channel sensitivity, $h=168$, $K=256$**
>
> **NOAA-US**
> ch=8 CRPS=0.99900
> ch=16 CRPS=0.96933
> ch=24 CRPS=0.96986
> ch=32 CRPS=0.98465
> ch=64 CRPS=0.97528
>
> **BMS Air**
> ch=8 CRPS=0.65065
> ch=16 CRPS=0.63659
> ch=24 CRPS=0.63864
> ch=32 CRPS=0.66018
> ch=64 CRPS=0.69667
>
> **2. Inference time vs. NeuralODE/CDE and diffusion baselines**
>
> **Direct answer.** LLapDiff is not a one-shot forecaster: inference uses iterative DDIM denoising, so our efficiency claim is more specific: it avoids numerical integration over physical time.
>
> **Evidence.** In inference, LLapDiff is around **449–451 ms** on NOAA-US / BMS Air, versus **553–560** ms for NeuralCDE, **~880–890 ms** for MRDiff, and **5–35** s for TimeGrad (with ContiFormer much larger, roughly **4x times slower** than TimeGrad). Representative wall-clock inference time on NOAA-US (ms; horizons 24/48/96/168): **LLapDiff 449/449/451/451**; **NeuralCDE 553/560/560/554**; **MRDiff 890/888/886/886**; **TimeGrad 5107/10226/20452/35774**. Thus, while slower than one-pass models, LLapDiff is substantially faster than heavier solver-style / diffusion baselines. This matches the paper’s complexity discussion in Appx. I: solver methods scale with function evaluations, whereas LLapDiff’s cost is dominated by diffusion steps with closed-form trajectory synthesis per step. We also tested varying pole pairs $K$: runtime stays nearly flat as $K$ increases from 16 to 512 (about **452–457 ms** on NOAA-US / BMS Air), indicating that the **denoising loop**, not modal synthesis, dominates cost. We will include these inference wall-clock times in Appx. I.
>
> **3. Strongly nonlinear or chaotic systems**
>
> **Direct answer.** The latent modal dynamics are local, history-conditioned with window-specific adaptive poles, and remain reasonably robust on non-stationary real data and controlled regime shifts.
>
> **Evidence.** In the model, poles are conditioned on the history summary to produce a window-specific Jacobian-equivalent representation rather than one fixed global linearization. The learned poles (Fig. 4) remain in the stable region ($\rho>0$) while spanning a broad frequency range ($\omega\in[0,2.5]$). Empirically, we evaluate on challenging irregular real-world datasets, including **non-stationary** ones (NOAA-US, NOAA-UK, PhysioNet, UCI Air), rather than a dedicated chaotic benchmark. To probe sharper regime changes, we also test synthetic damped sinusoids with an **abrupt change point**: frequency doubling changes CRPS from 0.3065 to 0.3376 and MAE from 0.3879 to 0.4217, while increasing the decay rate by $2.5\times$ changes CRPS from 0.2482 to 0.2890 and MAE from 0.3166 to 0.3621. This suggests proper robustness to abrupt spectral shifts, while extreme unseen nonlinear regimes can still be challenging. We will include these results in Appx. I.
>
> **4. Robustness beyond the i.i.d. gap assumption**
>
> **Direct answer.** The i.i.d. renewal derivation is a tractable theoretical motivation, not an exact assumption of the implemented model.
>
> **Evidence.** The paper notes that for **nonstationary or history-dependent gaps**, the same effective-pole view can be written conditionally as $\lambda_k=\mathbb E[\mathrm e^{s_k\Delta}\mid\cdot]$, and in practice, LLapDiff uses renewal averaging only as a guiding inductive bias for the gap-aware summarizer. The implemented model addresses non-i.i.d. gaps by conditioning on a history summary whose temporal token encodes timestamps / $\Delta t$ and masks. Empirically, this conditioning matters (Tables 3 and 4): removing the temporal token yields the largest degradation in the token ablation, and induced-missingness tests show graceful degradation under increasingly sparse observations. Several evaluated datasets are also explicitly non-stationary, so the method is verified beyond an idealized i.i.d.-gap setting.

---

> > ### Author Rebuttal · Reviewer_umN8 · 2026-04-03
> >
> > The rebuttal strengthens confidence in the method, particularly regarding robustness and computational positioning. While some modeling assumptions remain, they are acknowledged and empirically mitigated. Overall, this is a solid and well-supported contribution.

---

> > > ### Author Response · Authors · 2026-04-06
> > >
> > > Thank you for the acknowledgement. We appreciate your careful reading and are glad our response was helpful.

---

### Decision · Program_Chairs · 2026-04-30

**Decision:**

Accept (spotlight)

**Comment:**

This paper makes a strong contribution to the problem of forecasting irregular multivariate time series, which arise in a number of scientific domains. The proposed approach is based on integrating latent diffusion with Laplace-domain modal dynamics, which reviewers found to be highly innovative, technically sophisticated, and well-justified. There were initial concerns about the approach's reliance on a pretrained VAE and local linearization as well as about its computational efficiency. The authors provided ablation and sensitivity analyses during the discussion period that assuaged concerns about the former, as well as a discussion of timing analyses that largely assuaged concerns about the latter. The authors should incorporate these results and runtime comparisons into a revised version of the paper, which is already strong and worthy of acceptance at ICML.